# The Total Carbon Column Observing Network's GGG2020 Data Version

Joshua L. Laughner[1], Geoffrey C. Toon[1], Joseph Mendonca[2], Christof Petri[3], Sébastien Roche[4,5], Debra Wunch[6], Jean-Francois Blavier[1], David W.T. Griffith[7], Pauli Heikkinen[8], Ralph F. Keeling[9], Matthäus Kiel[1], Rigel Kivi[8], Coleen M. Roehl[10], Britton B. Stephens[11], Bianca C. Baier[13], Huilin Chen[14,**], Yonghoon Choi[15,16], Nicholas M. Deutscher[7], Joshua P. DiGangi[15], Jochen Gross[17], Benedikt Herkommer[17], Pascal Jeseck[18], Thomas Laemmel[19,*], Xin Lan[12,13], Erin McGee[6], Kathryn McKain[13], John Miller[13], Isamu Morino[20], Justus Notholt[3], Hirofumi Ohyama[20], David F. Pollard[21], Markus Rettinger[22], Haris Riris[23], Constantina Rousogenous[24], Mahesh Kumar Sha[25], Kei Shiomi[26], Kimberly Strong[6], Ralf Sussmann[22], Yao Té[18], Voltaire A. Velazco[7,27], Steven C. Wofsy[4], Minqiang Zhou[25,***], and Paul O. Wennberg[10,28]

[1]Jet Propulsion Laboratory, California Institute of Technology, Pasadena, CA, USA
[2]Climate Research Division, Environment and Climate Change Canada, Toronto, ON, Canada
[3]Institute of Environmental Physics, University of Bremen, Otto-Hahn-Allee 1, 28359 Bremen, Germany
[4]John A. Paulson School of Engineering and Applied Sciences, Harvard University, Cambridge, MA, USA
[5]Center for Astrophysics, Harvard & Smithsonian, Cambridge, MA, USA
[6]Department of Physics, University of Toronto, Toronto, Canada
[7]Centre for Atmospheric Chemistry, School of Earth, Atmospheric and Life Sciences, University of Wollongong, Wollongong, New South Wales, Australia
[8]Space and Earth Observation Centre, Finnish Meteorological Institute, Sodankylä, Finland
[9]Scripps Institute of Oceanography, La Jolla, California, USA
[10]Division of Geological and Planetary Sciences, California Institute of Technology, Pasadena, CA, USA
[11]Earth Observing Laboratory, National Center for Atmospheric Research (NCAR), Boulder, CO, USA
[12]Cooperative Institute for Research in Environmental Sciences, University of Colorado Boulder, Boulder, USA
[13]NOAA Global Monitoring Laboratory, Boulder, USA
[14]Centre for Isotope Research (CIO), Energy and Sustainability Research Institute Groningen (ESRIG), University of Groningen, Groningen, Netherlands
[15]NASA Langley Research Center, Hampton, VA 23681
[16]Analytical Mechanics Associated, Hampton, VA 23666
[17]Karlsruhe Institute of Technology (KIT), Institute of Meteorology and Climate Research (IMK-ASF), Karlsruhe, Germany
[18]Laboratoire d'Études du Rayonnement et de la Matière en Astrophysique et Atmosphères (LERMA-IPSL), Sorbonne Université, CNRS, Observatoire de Paris, PSL Université, 75005 Paris, France
[19]Université Paris-Saclay, CEA, CNRS, UVSQ, Laboratoire des Sciences du Climat et de l'Environnement (LSCE/IPSL), Gif-sur-Yvette, France
[20]National Institute for Environmental Studies (NIES), Onogawa 16-2, Tsukuba, Ibaraki 305-8506, Japan
[21]National Institute of Water and Atmospheric Research Ltd (NIWA), Lauder, New Zealand
[22]Karlsruhe Institute of Technology (KIT), IMK-IFU, Garmisch-Partenkirchen, Germany
[23]NASA Earth Science Technology Office (ESTO), B22, 242, 8800 Greenbelt Rd., Greenbelt, MD. 20771
[24]Climate and Atmosphere Research Centre (CARE-C), The Cyprus Institute, Nicosia, Cyprus
[25]Royal Belgian Institute for Space Aeronomy, Brussels, Belgium
[26]Earth Observation Research Center (EORC), Japan Aerospace Exploration Agency (JAXA), Tsukuba, Japan
[27]Deutscher Wetterdienst, Meteorological Observatory Hohenpeissenberg, 82383 Hohenpeissenberg, Germany
[28]Division of Engineering and Applied Science, California Institute of Technology, Pasadena, CA, USA

*Now at: University of Bern, Department of Chemistry, Biochemistry and Pharmaceutical Sciences and Oeschger Center for Climate Change Research, Bern, Switzerland

**Now at: Joint International Research Laboratory of Atmospheric and Earth System Sciences, School of Atmospheric Sciences, Nanjing University, Nanjing, China

***Now at: Institute of Atmospheric Physics, Chinese Academy of Sciences, China

**Correspondence:** Joshua Laughner (josh.laughner@jpl.nasa.gov)

**Abstract.** The Total Carbon Column Observing Network (TCCON) measures column-average mole fractions of several green-house gases (GHGs) beginning in 2004 from over 30 current or past measurement sites around the world, using solar absorption spectroscopy in the near infrared region. TCCON GHG data have been used extensively for multiple purposes, including in studies of the carbon cycle and anthropogenic emissions as well as to validate and improve observations made from space-

5 based sensors. Here, we describe an update to the retrieval algorithm used to process the TCCON near IR solar spectra and generate the associated data products. This version, called GGG2020, was initially released in April 2022. It includes updates and improvements to all steps of the retrieval, including but not limited to: converting the original interferograms into spectra, the spectroscopic information used in the column retrieval, post hoc airmass dependence correction, and scaling to align with the calibration scales of in situ GHG measurements.

All TCCON data are available through tccondata.org and hosted on CaltechDATA (data.caltech.edu). Each TCCON site has a unique DOI for its data record. An archive of all sites' data is also available with the DOI 10.14291/TCCON.GGG2020 (Total Carbon Column Observing Network (TCCON) Team, 2022). The hosted files are updated approximately monthly, and TCCON sites are required to deliver data to the archive no later than one year after acquisition. Full details of data locations are provided in the data availability section.

# 1 Introduction

The Total Carbon Column Observing Network (TCCON) is a network of nearly 30 ground-based, solar-viewing, Fourier transform infrared (FTIR) spectrometers that report observations of column-average mole fractions of $CO_2$, $CH_4$, $N_2O$, CO, HF, $H_2O$, and HDO in the atmosphere. The first two TCCON stations were established in 2004, with additional stations joining over the following years. As of July 2023, 30 sites exist. In that time, TCCON data have been used to estimate or

20 evaluate carbon fluxes (e.g. Keppel-Aleks et al., 2012; Peiro et al., 2022), for satellite validation (e.g. Wunch et al., 2017; Chen et al., 2022; Lorente et al., 2022), for model verification (e.g. Byrne et al., 2023), and for other purposes.

The need for updates to the retrieval algorithm used by TCCON has been largely driven by the need for increasingly high accuracy and precision of total column greenhouse gas (GHG) data for carbon cycle science and satellite validation. GHG measurements require high precision to distinguish signals from anthropogenic, terrestrial, or oceanic processes from the

25 background mixing ratios. The 2018 National Academies decadal strategy recommends random and systematic errors for $CO_2$ be less than 1 and 0.2 ppm ($\sim 0.25\%$ and $\sim 0.05\%$), respectively and likewise less than 6 and 2.5 ppb ($\sim 0.3\%$ and $\sim 0.1\%$), respectively for $CH_4$ (National Academies of Sciences, Engineering, and Medicine, 2018, Table B.1, question C-3, p. 601). Future space-based $CO_2$ observing missions are striving for even greater precision; for example, CO2M has a stated goal of

0.7 ppm precision and $< 0.5$ ppm systematic error in $X_{CO_2}$ (ESA, 2020). The increasingly stringent precision requirements for carbon cycle science and satellite validation demands that ground based networks, such as TCCON, continue to refine their data to support these requirements.

A second factor driving improvements in the retrieval is the emergence of portable, low resolution solar-viewing FTIR instruments such as EM27/SUNs. These instruments can be deployed to areas that cannot support a full TCCON site, and are also affordable enough to be deployed in greater density around locations of interest (e.g. cities). This capability complements the higher precision and accuracy data produced by TCCON. To facilitate comparisons between TCCON and EM27/SUN data, it is beneficial to use the same retrieval for both. Improvements to handling of EM27/SUN interferograms (§4.3) have been added.

TCCON instruments record interferograms of direct-sun measurements in the near-infrared (NIR) wavelengths. These interferograms are transformed into spectra from which the final column-average mole fractions (henceforth denoted as "$X_{gas}$", e.g. "$X_{CO_2}$") are derived using the retrieval software GGG.[1] Major versions of GGG are identified by the year of development. The previous version used to generate public TCCON data was GGG2014 and is described in Wunch et al. (2015) (see also Wunch et al., 2011, 2010). GGG2020 is the first major update applied to TCCON public data since GGG2014. The primary goal of this paper is to describe the changes in GGG2020 compared to GGG2014.

GGG retrieves trace gas column amounts by iteratively scaling an a priori trace gas vertical profile until the best fit between a spectrum simulated from those trace gas profiles by the built-in forward model and the observed spectrum is found. (This differs slightly from the Bayesian framework described in Rodgers (2000). Please refer to section 3.4 of Roche (2021) for a discussion of specific differences.) A single gas may be fit in more than one spectral window; for example, GGG2020 produces the standard TCCON $CO_2$ product from two separate retrievals using two spectral windows (6220 to 6260 cm$^{-1}$ and 6297 to 6382 cm$^{-1}$). Each window is run separately and produces its own posterior scaled trace gas profile, which are separately integrated to generate a column density from each window. These column densities are combined and converted to the final $X_{gas}$ value. Retrieving each window separately, rather than concatenating the spectral information, makes it simpler to handle non-contiguous windows that need different state vector elements. It also allows biases that differ between these windows to be expressed separately in the resulting output data and, if necessary, corrected separately. The output values (column densities and profile scaling factors) from different windows with similar averaging kernels for the same target gas are combined in a weighted average during post processing.

The post processing step includes the conversion from column densities to column-average dry mole fractions, followed by the above window-to-window averaging, an empirical airmass-dependent correction, and a scaling correction to tie TCCON data to the relevant calibration scales. Airmass-dependent errors can arise from, for example, errors in the relative intensities of strong and weak absorption lines for a target gas. At large solar zenith angles (SZAs), the longer light paths through the atmosphere will cause strong absorption lines to completely absorb incoming light within their core wavelengths; such lines may be referred to as "blacked out". Blacked out lines cannot contribute information to the retrieval, so the retrieval must get a greater fraction of its information from weaker lines in the spectral window or the wings of saturated lines. If there is a different

---

[1]GGG is the proper name of the software, and is not an acronym.

bias in the forward model between the strong and weak lines, it will manifest as an error in the retrieved column amounts that varies with SZA and is symmetric about solar noon. Once the magnitude of this error is derived (§8.1), a post-processing correction is applied to mitigate it.

Apply a scaling factor to tie to the in situ calibration scales is necessary because the spectroscopic parameters used in the forward model are not in general known to the accuracy needed for greenhouse gas data. However, since all TCCON sites use the same retrieval (and thus the same forward model), we use a single mean scaling factor to remove the mean bias caused by errors in the spectroscopic parameters. It is not intended to correct biases from instrument artifacts, such as an imperfect instrument line shape (ILS), as such biases can change over time. The scaling factors for the various gases are derived from comparisons between TCCON data and in situ vertical profiles measured by aircraft- or balloon- borne instruments (§8.3).

Finally, the conversion from column densities to column-average dry mole fractions is done by dividing the target gas column density ($V_{\mathrm{gas}}$ in molec. cm$^{-2}$) by the $O_2$ column density ($V_{O_2}$ in molec. cm$^{-2}$), then multiplying by the mean $O_2$ dry mole fraction ($f_{O_2}$) in the atmosphere:

$$X_{\mathrm{gas}} = \frac{V_{\mathrm{gas}}}{V_{O_2}} \cdot f_{O_2} \tag{1}$$

GGG2020 assumes that $f_{O_2} = 0.2095$ for all $X_{\mathrm{gas}}$ products except those listed in §8.3.2 where a variable $O_2$ dry mole fraction has been implemented. The advantages of normalizing to the $O_2$ column are:

1. It normalizes for path length. Observations at higher surface elevations will have smaller column densities compared to those from lower altitudes, due to the shorter vertical extent. Normalizing to the $O_2$ column removes this effect.

2. Because $O_2$ and the primary TCCON gases are measured on the same detector, many biases related to the detector and pointing partially cancel out (e.g. ILS, mis-pointing, zero-level offsets, Wunch et al., 2011, Appendices A and B). Note that TCCON uses the $^1\Delta$ $O_2$ band around 7885 cm$^{-1}$, rather than the A-band (around 13080 cm$^{-1}$, commonly used by satellite missions to avoid interference from airglow). The $^1\Delta$ $O_2$ band is closer in frequency to the near-IR $CO_2$ and $CH_4$ bands than the $O_2$ A-band; this minimizes differences frequency-dependent effects (e.g. refraction) between the $O_2$ and $CO_2$ or $CH_4$ bands.

GGG is comprised of several sub-programs, which handle these various elements of the retrieval. The flow among these sub-programs is shown in Fig. 1. Each of these has been upgraded for GGG2020:

– **i2s**: converts interferograms to spectra. Updates include identifying detector nonlinearity and better phase correction (§4).

– **gsetup**: prepares the input files needed to run gfit (a priori meteorology and trace gas profiles, atmospheric path information, etc.) in the required formats. Updates include the source of a priori meteorology and trace gas profiles and the retrieval grid (§5).

– **gfit**: retrieves column densities from the spectra output by i2s. Updates include the forward model spectroscopy (§6) and continuum fitting (§7).

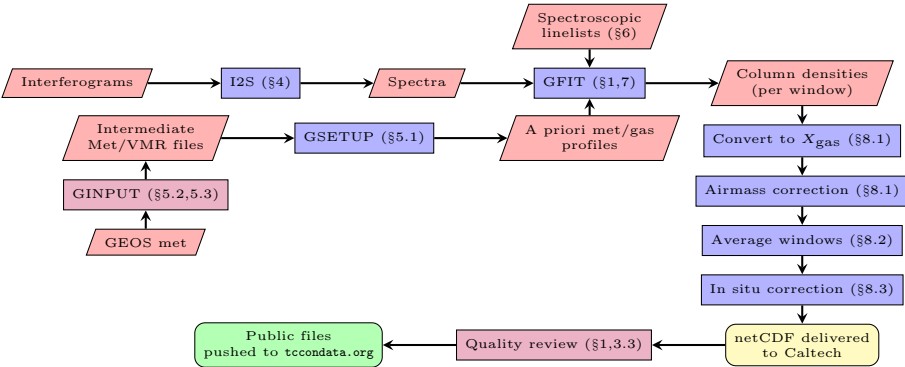

**Figure 1.** The flow among all the components of GGG and the TCCON data. Red trapezoids (e.g "Interferograms") represent input or intermediate data. Blue rectangles (e.g. "I2S") represent processing steps that are part of GGG. The yellow rounded rectangle ("netCDF...") represents a transfer step. The purple rectangles (e.g. "GINPUT") represent a centralized processing steps. The green rounded rectangle ("Public files") indicates public-facing data. Numbers prefixed with "§" refer to sections of this paper.

  – **Post processing**: a suite of programs that collates the output from gfit and applies post hoc corrections. Updates include the airmass correction (§8.1), window to window averaging (§8.2), and scaling to tie to in situ calibration scales (§8.3).

GGG2020 data is available through tccondata.org. A repository containing the full set of publicly available data is available through CaltechDATA (Total Carbon Column Observing Network (TCCON) Team, 2022). This data undergoes quality evaluation before release, with all data reviewed by experienced TCCON members from various sites. Each TCCON site's data record has its own unique DOI. On occasions that a site needs to reprocess and redeliver data already released to the public, the revised dataset receives a new DOI with the revision number incremented. TCCON sites are permitted to withhold data from the public archive for up to one year from acquisition. This public archive is updated approximately once per month with newly delivered or released data. The TCCON data product is documented extensively through the TCCON Wiki (https://tccon-wiki.caltech.edu/). Users are asked to familiarize themselves with the data use policy and license, which are available at https://tccon-wiki.caltech.edu/Main/DataUsePolicy.

As this paper is quite long, we provide a list of contents in Table 1 for readers to jump to sections of interest to them. We begin with a review of new $X_{\mathrm{gas}}$ products and changes to the data product most likely to be of interest to users. Next (starting with §4), for each step in the GGG processing chain, we describe the changes between GGG2014 and GGG2020. Finally, we present an uncertainty budget for GGG2020 (§9).

## 2 New $X_{\mathrm{gas}}$ products

GGG2020 introduced $X_{\mathrm{CO_2}}$ dry mole fractions retrieved in two new windows: band between 4809.74 and 4896.0 cm$^{-1}$ with higher intensity absorption than the 6330 band and a band with weaker absorption between 6041.8 and 6105.2 cm$^{-1}$. We refer to these as "lCO$_2$" (for "lower" CO$_2$) and "wCO$_2$" (for "weak" CO$_2$), respectively. Figure 2 shows how these two windows

**Table 1.** Contents of the paper with associated page numbers

(plus the windows for the standard TCCON $X_{CO_2}$ product) align with the strong and weak $CO_2$ windows used by OCO -2
and -3. These are reported as separate $CO_2$ products ($X_{lCO_2}$ and $X_{wCO_2}$) and are not averaged together with the standard
TCCON $X_{CO_2}$ product. Figure 3 shows the column averaging kernels (AKs) and $CO_2$ absorption lines in these two windows.
The $lCO_2$ AKs increase towards the surface, while, at small slant $X_{gas}$ amounts (i.e. small solar zenith angle) the $wCO_2$ AKs
are greater in the stratosphere than in the lower troposphere. This is because, as seen in Fig. 3b and d the $CO_2$ absorption lines
in the $lCO_2$ band are mostly saturated at the line center, while the $wCO_2$ lines are not. When used together with the standard
TCCON $X_{CO_2}$ product (which has an AK profile that is more constant with altitude than the $wCO_2$ or $lCO_2$ products, see
Fig. 5), this provides the potential to separate changes in $CO_2$ at the surface, from those in the free troposphere or stratosphere
(Parker et al., 2023).

For $wCO_2$, we chose not to use the second weak band around 6500 cm$^{-1}$ for reasons detailed in §8.1. For $lCO_2$, we did not
use the strong band around 4900 cm$^{-1}$ because the lines are so strong that the retrieval would be more sensitive to errors in the
line shape and zero level offsets in the interferograms.

Beginning with GGG2020, experimental mid-IR data products will be available from select TCCON sites equipped with an
InSb (indium antimonide) detector that enables measurements in the 1800 to 4000 cm$^{-1}$ frequency range. Gases observed in
this range include, but are not limited to, $O_3$, $N_2O$, CO, $CH_4$, NO, $NO_2$, carbonyl sulfide, formaldehyde, and ethane. These

**Table 2.** List of TCCON sites and their associated data citations as of 20 Dec 2022. Some sites (Lauder, JPL) have had different FTIR instruments operating over different periods, and so are listed multiple times.

| Site ID | Site Name | Location | Data Citation |
|---|---|---|---|
| ae | ascension01 | Ascension Island, Saint Helena | Feist et al. (2017) |
| an | anmeyondo01 | Anmyeondo, South Korea | Goo et al. (2017) |
| bi | bialystok01 | Bialystok, Poland | Petri et al. (2017) |
| br | bremen01 | Bremen, Germany | Notholt et al. (2022) |
| bu | burgos01 | Burgos, Philippines | Morino et al. (2022c) |
| ci | pasadena01 | Pasadena, California, USA | Wennberg et al. (2022c) |
| db | darwin01 | Darwin, Australia | Deutscher et al. (2023a) |
| df | edwards01 | AFRC, Edwards, CA, USA | Iraci et al. (2022b) |
| et | easttroutlake01 | East Trout Lake, Canada | Wunch et al. (2022) |
| eu | eureka01 | Eureka, Canada | Strong et al. (2022) |
| fc | fourcorners01 | Four Corners, NM, USA | Dubey et al. (2022b) |
| gm | garmisch01 | Garmisch, Germany | Sussmann and Rettinger (2017a) |
| hf | hefei01 | Hefei, China | Liu et al. (2022) |
| hw | harwell01 | Harwell, UK | Weidmann et al. (2023) |
| if | indianapolis01 | Indianapolis, Indiana, USA | Iraci et al. (2022a) |
| iz | izana01 | Izana, Tenerife, Spain | Blumenstock et al. (2017) |
| jc | jpl01 | JPL, Pasadena, California, USA | Wennberg et al. (2022e) |
| jf | jpl02 | JPL, Pasadena, California, USA | Wennberg et al. (2022a) |
| js | saga01 | Saga, Japan | Shiomi et al. (2022) |
| ka | karlsruhe01 | Karlsruhe, Germany | Hase et al. (2022) |
| lh | lauder01 | Lauder, New Zealand | Sherlock et al. (2022a) |
| ll | lauder02 | Lauder, New Zealand | Sherlock et al. (2022b) |
| lr | lauder03 | Lauder, New Zealand | Pollard et al. (2022) |
| ma | manaus01 | Manaus, Brazil | Dubey et al. (2022a) |
| ni | nicosia01 | Nicosia, Cyprus | Petri et al. (2023) |
| ny | nyalesund01 | Ny-Ålesund, Svalbard, Norway | Buschmann et al. (2022) |
| oc | lamont01 | Lamont, Oklahoma, USA | Wennberg et al. (2022d) |
| or | orleans01 | Orleans, France | Warneke et al. (2022) |
| pa | parkfalls01 | Park Falls, Wisconsin, USA | Wennberg et al. (2022b) |
| pr | paris01 | Sorbonne Université, Paris, FR | Te et al. (2022) |
| ra | reunion01 | Reunion Island, France | Maziere et al. (2022) |
| rj | rikubetsu01 | Rikubetsu, Hokkaido, Japan | Morino et al. (2022a) |
| so | sodankyla01 | Sodankylä, Finland | Kivi et al. (2022) |
| tk | tsukuba02 | Tsukuba, Ibaraki, Japan, 125HR | Morino et al. (2022b) |
| wg | wollongong01 | Wollongong, Australia | Deutscher et al. (2023b) |
| xh | xianghe01 | Xianghe, China | Zhou et al. (2022) |
| zs | zugspitze01 | Zugspitze, Germany | Sussmann and Rettinger (2017b) |

products offer the potential to extend the applications of TCCON data to new areas of research. However, currently these data
do not have any postprocessing corrections for airmass dependence (§8.1) or scaling to in situ data (§8.3) applied.

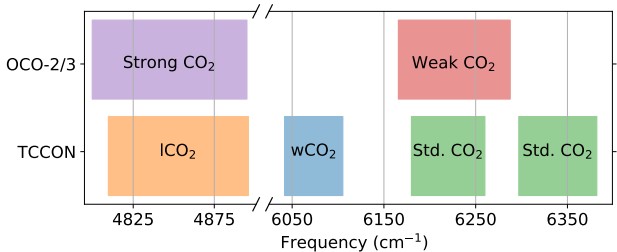

**Figure 2.** Frequency ranges of the TCCON $CO_2$ windows (those for the standard product as well as the two new products discussed in §2 compared to the frequency ranges of the OCO -2 and -3 $CO_2$ windows (Crisp et al., 2021).

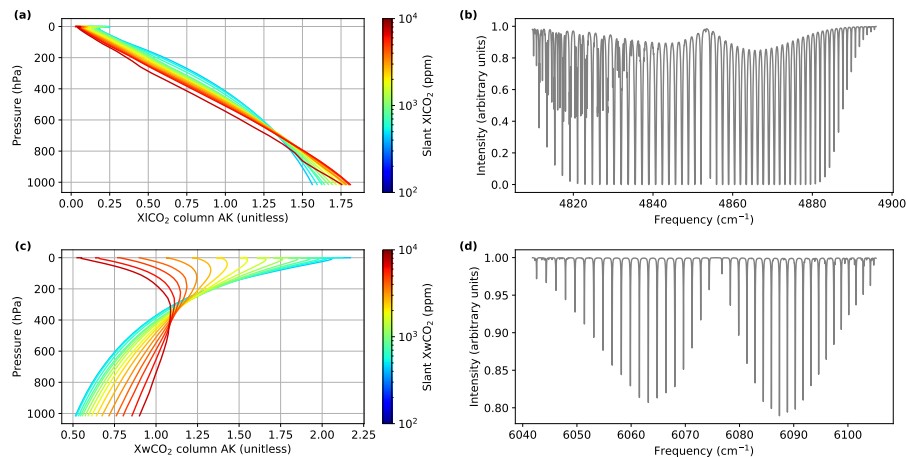

**Figure 3.** Column averaging kernels (panels **a**, **c**) and calculated $CO_2$ absorption lines (panels **b**, **d**) in the $lCO_2$ (panels **a**, **b**) and $wCO_2$ (panels **c**, **d**) windows, respectively. The absorption lines in panels (b) and (d) are for a TCCON spectrum measured at solar zenith angle = $39.684°$ in Jul 2004 at Park Falls, WI, USA. In panels (a) and (c), the different colors indicate AKs for different slant $X_{gas}$ amounts. "Slant $X_{gas}$" is a measure of total absorber column along the light path. See §3.1 for details.

## 3 Miscellaneous data format changes

### 3.1 AK binning

The publicly-available GGG2020 TCCON files now include one averaging kernel (AK) per observation. (For a description of how these column AKs are calculated by GGG, see section 3.5 of Roche (2021).) This is a change from GGG2014, where the
public files included a table of canonical AKs for a limited set of SZAs, and users were required to interpolate the AKs to the SZA of each spectrum. This was done in response to user requests to simplify the use of the averaging kernels. This does not mean that averaging kernels are computed by GGG for every TCCON observation (they are not). Internally, we still use a table

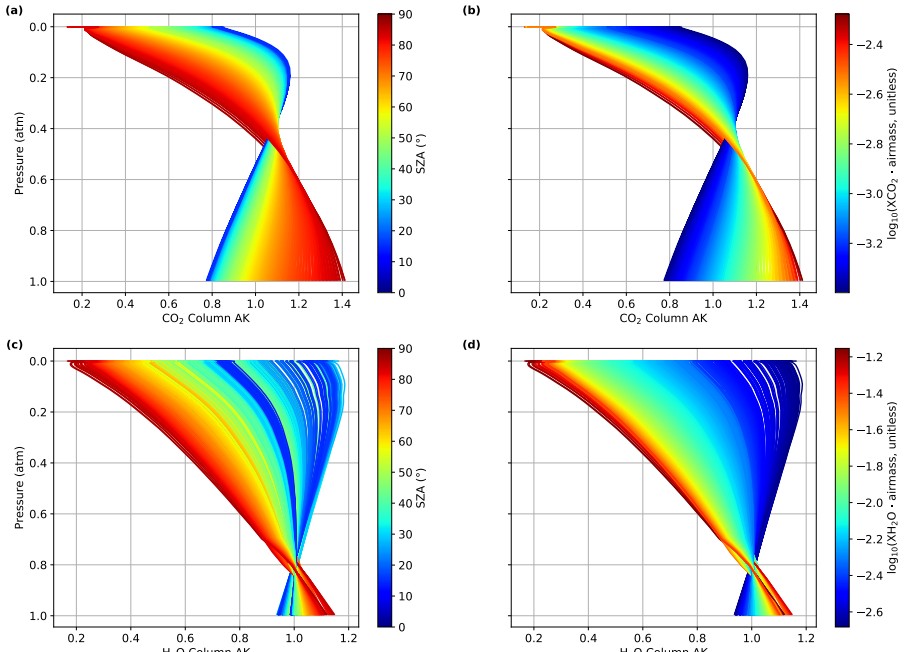

**Figure 4.** $CO_2$ and $H_2O$ AKs from four days' measurements at the TCCON site in Lamont, OK, USA. **(a)** $CO_2$ AKs binned by SZA. **(b)** $CO_2$ AKs binned by slant $X_{CO_2}$. **(c)** $H_2O$ AKs binned by SZA. **(d)** $H_2O$ AKs binned by slant $X_{H_2O}$.

of precomputed AKs, which are interpolated as needed to provide per-spectrum AKs in the public files. This affords significant saving in data storage, as the files GGG requires for to compute the AKs are very large.

Though users of public TCCON data no longer need to know how the AK tables are structured, there are two changes from GGG2014 that we wish to document here.

First, in GGG2020, the bin coordinate has changed from solar zenith angle (SZA) to "slant $X_{gas}$," which is defined as:

$$\text{Slant } X_{gas} = \text{airmass} \cdot X_{gas} \tag{2}$$

where "airmass" is the unitless ratio of slant to vertical column calculated by GGG in the $O_2$ window and "$X_{gas}$" is the
column-average dry mole fraction of the gas of interest. Using slant $X_{gas}$ as the bin coordinate correctly accounts for cases where the dynamic range of a gas's concentrations is large enough to change the AK at a single SZA. This can be seen in Fig. 4. For $CO_2$ (Fig. 4a,b), the AKs vary smoothly and monotonically with either SZA or slant $X_{CO_2}$. However, for $H_2O$, because the mixing ratios vary by orders of magnitude, the AKs do not vary simply with SZA (Fig. 4c) but do with slant $X_{H_2O}$ (Fig. 4d). Therefore, slant $X_{gas}$ was adopted as the binning coordinate for all AKs for consistency.
Second, in order to provide per-spectrum AKs in the public TCCON data files without significantly increasing the file size, it was necessary to ensure that observations with similar slant $X_{gas}$ values had identical AKs so that the netCDF compression

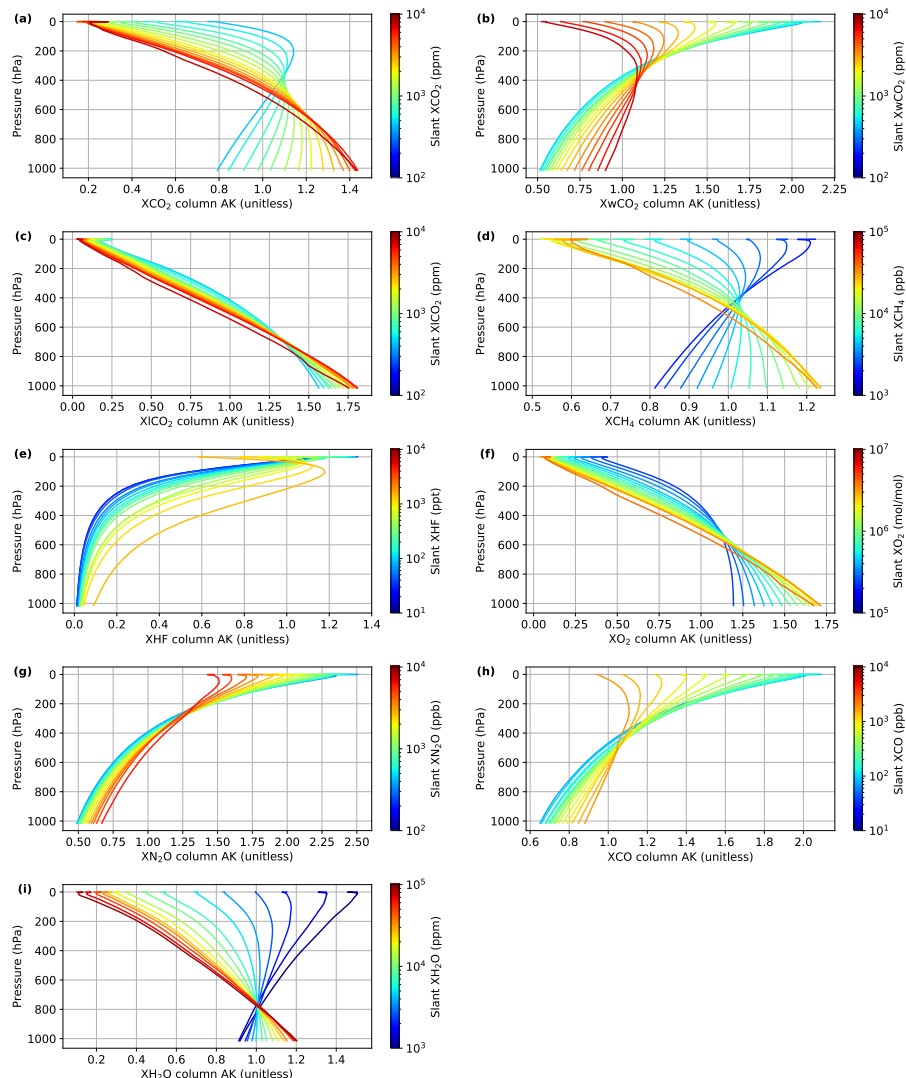

**Figure 5.** Precomputed column AKs for TCCON $X_{gas}$ products: **(a)** $X_{CO_2}$, **(b)** $X_{wCO_2}$, **(c)** $X_{lCO_2}$, **(d)** $X_{CH_4}$, **(e)** $X_{HF}$, **(f)** $X_{O_2}$, **(g)** $X_{N_2O}$, **(h)** $X_{CO}$, **(i)** $X_{H_2O}$, and **(j)** $X_{HDO}$.

algorithm could operate effectively. We achieved this by "quantizing" the slant $X_{gas}$ values that we interpolated the AKs to; that is, we select 500 slant $X_{gas}$ values that cover the expected range of slant $X_{gas}$, plus 50 additional points to cover extreme values. Each observation then uses the AK corresponding to the one of those 550 slant $X_{gas}$ values closest to its true slant $X_{gas}$ value. This scheme keeps the difference between the quantized and full resolution AKs to $< 1\%$ in 90% of observations while only increasing file size by $\sim 20\%$.

## 3.2 A priori profiles and AK corrections

As described in §5.3, the a priori profiles reported in the published GGG2020 netCDF files are in wet mole fraction. When applying an averaging kernel correction to calculate the $X_{gas}$ value that would be retrieved by TCCON for an arbitrary gas profile, that gas profile must be converted into wet mole fraction. This can be done using either the TCCON $H_2O$ a priori profile provided or an $H_2O$ profile measured or modeled coincidentally with the gas profile for which an $X_{gas}$ value is desired. Users who are unsure which is appropriate for their application are encouraged to reach out to the TCCON network chairs (listed at https://tccon-wiki.caltech.edu/Main/SteeringCommitteeMembership) for assistance.

## 3.3 Changes to quality flags

As in GGG2014, a retrieval is flagged as being poor quality if any of the retrieved $X_{gas}$ or $X_{gas}$ error values, or ancillary variables pertaining to instrument operation or local observation conditions are outside of expected ranges. Such spectra are not included in the publicly-available data files. In GGG2020, spectra may also be flagged as poor quality and withheld if:

- the staff at the TCCON site identify a hardware issue affecting that spectrum

- during pre-release data review, a time period containing that spectrum is identified as out-of-family for TCCON data.

The latter case focuses on a smoothed timeseries of $X_{luft}$ and DIP. DIP is a measure of nonlinearity in the detector or signal chain (§4.1). $X_{luft}$ is a diagnostic for retrieval biases (see §6.3 for a detailed definition). As shown in §8.3 and §9, deviation of $X_{luft}$ from the network median correlate with bias in the other $X_{gas}$ products. Therefore, when a 500-spectrum rolling median of $X_{luft}$ falls consistently outside the nominal range of 0.995 to 1.003, that time period is rejected, as the $X_{gas}$ products will likely have biases larger than the required TCCON accuracy. Likewise, testing has shown that increasing magnitude of DIP increases bias in $X_{CO_2}$ (Fig. 6). In most cases, data where DIP consistently exceeds $\pm 5 \times 10^{-4}$ during the initial quality assessment will be reprocessed with a nonlinearity correction (§4.1) applied to remove this bias. In very rare cases, if such reprocessing is not possible, the data are removed in order to keep the $X_{CO_2}$ bias less than 0.25 ppm.

## 4 Improved interferogram-to-spectrum conversion

There have been substantial code changes and streamlining of common code in i2s, the interferogram-to-spectrum conversion subroutine. The main substantive improvements to the code are in the handling of detector nonlinearity, the phase correction, and other changes.

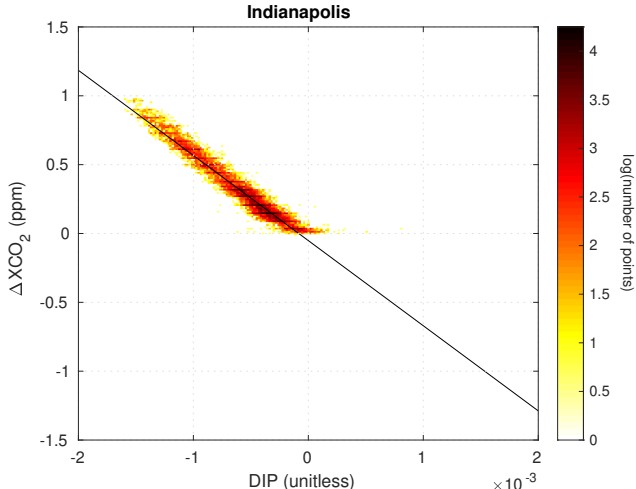

**Figure 6.** Detector nonlinearity can cause a bias in $X_{CO_2}$. This figure shows an example of the difference between the $X_{CO_2}$ retrieved after correcting the nonlinearity and prior to the nonlinearity correction as a function of the DIP parameter, that is a proxy for nonlinearity. Prior to correction, the Indianapolis data had DIP values that were almost exclusively negative. To limit the $X_{CO_2}$ bias caused by nonlinearity to less than 0.25 ppm, the absolute value of the DIP must be smaller than $0.5 \times 10^{-3}$.

## 4.1 Detector nonlinearity

The largest signals in an interferogram generated by a Fourier transform spectrometer are found near zero-path difference (ZPD), where light from all wavelengths constructively interfere. The modulated signal levels drop significantly away from ZPD. If the detector measuring the interferogram has a nonlinear response, the variations in the signal near ZPD will be more distorted than in the rest of the interferogram. This causes a discrepancy between the low-resolution spectral envelope (diagnosed near ZPD) and the high resolution spectral lines (diagnosed at larger path differences). Nonlinear detector responses can be strongly pronounced or subtle, and several improvements to i2s have been made to address these situations.

We have implemented a check early in i2s processing to remove interferograms affected by signal chain saturation, an extreme form of nonlinearity. If the signal intensity is too large, the ZPD signal will reach the maximum value permitted by the detector electronics. We call this "interferogram saturation" and this causes irreversible loss of information. Such saturation is rarely found in the TCCON spectra, and is straightforward to resolve once it is identified. To mitigate signal chain saturation, we carefully set the pre-amplifier gain such that, even under the most intense illumination, the signal chain does not saturate. To avoid detector saturation, we limit the number of photons incident on the detector through reducing the field stop or aperture stop diameter or by placing an optical filter in the beam. Because this effect depends on sunlight intensity, saturation is more likely to occur near noon than later or earlier in the day. It is also seasonally dependent, or dependent on the amount of water vapor in the atmosphere. In GGG2020, we have implemented a saturation check to discard any saturated interferograms based on the maximum and minimum values of their signal.

There are more subtle detector nonlinearity effects that do not necessarily result in interferogram saturation, but can adversely affect the retrievals. We now compute and store a detector nonlinearity diagnostic variable ("DIP") as part of the regular TCCON data processing. Keppel-Aleks et al. (2007) described the solar intensity variation correction applied to the TCCON interferograms that has been part of the TCCON processing software since 2007. In this correction, a low-pass filtered interferogram is used to re-weight the original AC interferogram, largely removing the impacts of solar intensity fluctuations during a measurement. As part of this work, Keppel-Aleks et al. realized that detector nonlinearity becomes observable in the low-pass filtered interferogram as a symmetrical reduction in intensity, that we term "DIP", near ZPD (see Fig. 6b in Keppel-Aleks et al., 2007). The magnitude of this DIP is a diagnostic of the severity of detector nonlinearity, and is now computed, stored, and reported as part of the routine TCCON processing.

Detector nonlinearity in the Sodankylä TCCON data persisted from early in their record until the problem was found in 2017. The problem in the early data was resolved by applying the nonlinearity correction developed by Hase (2000) directly to the interferogram before transforming it into a spectrum. This correction process and its results are described in detail in Appendices A and B of Sha et al. (2020). In that paper, the authors show that the nonlinearity caused a bias in $X_{\mathrm{CO_2}}$ of about 0.5 ppm in the 2017 Sodankylä data. After 2017, the problem was resolved by optically limiting the light entering the interferometer.

We now use the DIP diagnostic during the quality control step to identify all TCCON spectra affected similarly by nonlinearity. Once such data is identified, the correction process described in the previous paragraph is applied to the afflicted data. We are in the process of incorporating the correction process as a standardized part of the interferogram-to-spectrum processing to make this process easier to complete in the future.

At a few sites, DIP is consistently observed to be positive—that is, the detector appears to have a supralinear response, rather than traditional saturation response seen at, for example, Sodankylä. The procedure described in the last paragraph is not effective at correcting the supralinear behavior as it has a different physical cause than the sublinear behavior. Based on tests performed at the Garmisch TCCON site, our current hypothesis is that this behavior results from overfilling the detector element with the light beam (Corredera et al., 2003), and the magnitude of the effect varies from detector to detector. Another possible cause of supralinearity in detectors can come from absorptive layers on the InGaAs active region itself (Fox, 1993), but we do not yet have evidence that this is occurring in our instruments.

## 4.2 Phase correction

Sampled interferograms are always asymmetrical, either because the sampling grid does not include the ZPD position, or because the underlying continuous interferogram is already asymmetrical even before it is sampled. This asymmetry causes the resulting, post-FFT, complex spectrum to have substantial imaginary terms. A phase correction is necessary to resample the interferogram such that is it sampled symmetrically about ZPD, resulting in a computed spectrum that has the signals of interest in the real component and only the noise is divided between both the real and imaginary component.

If we used a power spectrum ($\sqrt{\Re^2 + \Im^2}$), avoiding phase correction, it would compute a spectrum that is entirely real, but would retain all of the noise in the real and imaginary component of the spectrum. Therefore the final noise level in a power

spectrum would be a factor of $\sqrt{2}$ greater than in a phase-corrected and Fourier transformed spectrum. Additionally, in a power spectrum, saturated (zero intensity) regions would no longer be centered at zero, as any noise present is rectified and so made all positive. For these reasons, we compute a phase correction.

We use the phase correction method described by Forman et al. (1966), with a spectral domain convolution as described by Mertz (1965, 1967). The phase correction is performed using a low resolution double-sided interferogram, apodized with a $\cos^2$ function, to compute the angle between the real and imaginary components of the spectrum. This angle is a smoothly varying function of wavenumber, and is called the phase curve. Its counterpart in interferogram space is called the phase correction operator. In regions of the spectrum with sufficient signal, the phase curve is well defined, but where the spectrum is blacked out by water vapor, another strong absorber, or an optical component, it can become undefined. Therefore, to compute the phase correction operator, we need to set a signal threshold so that we can compute a well-behaved phase curve across the spectral region of interest. We interpolate the phase curve linearly across the blacked-out regions of the spectrum where the phase curve is below the signal threshold. The phase curve is interpolated to 0 at both 0 $\mathrm{cm}^{-1}$ and the Nyquist frequency (15798 $\mathrm{cm}^{-1}$).

In GGG2014, several TCCON stations showed retrievals of $X_{\mathrm{gas}}$ with systematic differences between spectra generated from interferograms collected while the scanning mirror moves away from zero path difference ("forward" scans) and while moving toward zero path difference ("reverse" scans). These differences are typically less than 0.5 ppm in $X_{\mathrm{CO_2}}$, but with larger differences observed at the Ny Ålesund, Eureka, Paris, and Zugspitze TCCON stations. This forward-reverse bias was tracked down to the phase correction operator, and, more specifically, the minimum signal level threshold for which the phase operator is calculated.

To address this issue, we lowered the phase curve threshold from 0.02 (2%, in GGG2014) to 0.001 (0.1%, in GGG2020) of the peak spectral signal which improves the consistency between forward and reverse scans. This eliminates the observed bias in $X_{\mathrm{CO_2}}$ between forward and reverse scans, but is not a fully general solution to the underlying problem. In a future version of i2s, we hope to develop a phase correction scheme that is independent of the signal level.

### 4.3 Improved EM27/SUN support

We now make better use of the entire interferogram collected by the spectrometer in i2s. In typical linear single-passed Fourier transform spectrometers (such as those used by TCCON), we collect most of our interferometric data between zero path difference (ZPD) and the maximum optical path difference (MOPD) positions of the scanning mirror. However, in order to perform a phase correction, a small amount of data must be collected on the other side of ZPD, which we call the "short arm" of the interferometer. The "long arm" is the section from ZPD to MOPD.

I2S now has the capability to process interferograms as single sided (using data only from one side of ZPD, usually the long arm) or double sided (using data from both sides of ZPD, the long and short arms). When processing an interferogram as double sided, the optical path difference (OPD) on either side of ZPD must be the same. This means that for standard TCCON processing, I2S will always choose to process the interferogram as single sided, because the long arm is much longer ($\geq 45$ cm) than the short arm (typically 0.2 to 5.0 cm). However, for spectrometers such as the EM27/SUNs where the OPD is more

symmetrical about ZPD, I2S can process the interferogram as double sided, which avoids discarding useful data from the other side of ZPD.

## 5 Improved a priori profiles

### 5.1 Modified retrieval grid

In GGG, the retrieval is done on a fixed altitude grid. In GGG2014 the altitude grid had a constant spacing of 1 km with 71 levels between 0-70 km above sea level. In GGG2020 the grid was updated to 51 levels between 0-70 km above sea level with spacing increasing away from the surface following the expression:

$$z_i = i \cdot (0.4 + 0.02 \cdot i) \tag{3}$$

where $z_i$ is the altitude of the $i^{th}$ level in kilometers. As the altitude grids are fixed to sea level, this does mean that some
sites have some levels below the terrain which are not included in the integration.

### 5.2 Meteorological updates

In GGG2014 the a priori $H_2O$, pressure, density, and temperature profiles were derived from NCEP 6-hourly reanalyses. In GGG2020, these profiles are now derived from GEOS 5 FP-IT 3-hourly product in addition to potential temperature, potential vorticity, $O_3$, and CO profiles. GGG2020 uses the nearest profile in time, changing every three hours, to better capture changes
throughout the day. The potential vorticity profiles are used to derive equivalent latitude profiles based on the equation in Allen and Nakamura (2003). Equivalent latitude is used in deriving the stratospheric part of the a priori trace gas concentration profiles (Laughner et al., 2023). GGG2020 will transition to the GEOS IT product when it replaces GEOS FP-IT; an analysis to quantify the impact of that change on TCCON $X_{\mathrm{gas}}$ products is planned.

### 5.3 Trace gas profile updates

GGG2020 includes a substantial redesign of the algorithm that generates the $CO_2$, $CH_4$, $N_2O$, HF, CO, and $O_3$ a priori profiles. Generating these profiles is now handled by ginput, a separate program from gsetup. The ginput algorithm is described in detail in Laughner et al. (2023). Briefly, the $CO_2$, $CH_4$, and $N_2O$ profiles are tied to the long term records from the NOAA observatories in Mauna Loa, Hawaii and American Samoa (Lan et al., 2022b, a, c), in order to ensure the growth rates of these gases are correctly accounted for. Individual profiles are produced based on the mean transport time between the profile location
and the Mauna Loa/American Samoa observatories and (in the stratosphere) chemical loss. HF profiles are derived from $CH_4$ profiles using the HF-$CH_4$ relationships previously identified by Washenfelder et al. (2003) and Saad et al. (2014, 2016). CO and $O_3$ profiles are drawn from the GEOS FP-IT chemical product[2] (Lucchesi, 2015) with adjustments in the stratosphere to better match observations. (See Laughner et al. (2023) for details on these adjustments.)

---

[2]We expect to transition to the GEOS IT product when it supersedes GEOS FP-IT. However, that had not yet occurred at time of writing.

One additional change compared to GGG2014 is that the a priori profiles are now given in wet, rather than dry, mole fraction. This is necessary as GGG calculates absorber number densities as the prior wet mole fractions times the number density of air, which is assumed to include water. The a priori profiles provided in the published data files are also in wet mole fraction. Thus, whenever comparing GGG2020 a priori profiles in the published netCDF files with other sources, care must be taken to ensure that the comparisons convert both profiles to the same (wet or dry) mole fractions. Note that the column-average $X_{\mathrm{gas}}$ values are always reported in dry mole fraction.

## 6    Updated spectroscopy

### 6.1    Telluric & Solar line lists

As described in Toon et al. (2016), the telluric linelist (atm.161, Toon, 2022c) is a "greatest hits" compilation based heavily on HITRAN predecessor lists, but not necessarily the latest HITRAN version for all bands and gases. As new linelists become available, they are evaluated using laboratory and atmospheric spectra, and compared with earlier HITRAN linelists and the current atm.161 linelist, which is updated if the new linelist represents an improvement in any spectral regions, as determined by 1) improved fitting residuals, 2) better consistency of retrieved gas amounts from different windows and bands, and 3) reduced airmass-dependence of the retrieved gas amounts. Additionally, ad hoc empirical corrections are performed to some lines, bands, and gases to fix obvious errors. Since the GGG2014 version of the linelist, there have been many improvements to the $H_2O$ and HDO spectroscopy throughout the main TCCON region (4000 to 8000 cm$^{-1}$). Water vapor is an important interferer in almost all windows, as is $CH_4$ which has also undergone substantial ad hoc correction, but not in the $2v_3$ band (5800 to 6200 cm$^{-1}$) where $CH_4$ itself is retrieved.

Table 3 shows how the spectral residuals (i.e. the difference between the observed and simulated spectra for the retrieved state) and VMR scale factors (VSFs, the ratio of the retrieved to a priori gas column) have progressed between the GGG2014 and GGG2020 linelists. These results are for spectra of gas cells with a known amount of $CO_2$, so are restricted to $CO_2$. For all the $CO_2$ bands used by GGG2020, the spectral residuals show clear reductions for the GGG2020 linelist combining Voigt and non-Voigt lines (see §6.2 for details of the non-Voigt lineshapes) compared to GGG2014. The mean bias in line strengths, as indicated by the VSF values, was more varied: two windows had less bias (with VSFs closer to 1) but the other two had slightly larger bias. However, such biases are removed by scaling to match in situ data (§8.3), so while removing such biases with improved spectroscopy is desirable, their presence has little impact on the TCCON data.

Improvements to the telluric linelists are communicated to the HITRAN group through spectroscopic evaluations, posted to https://mark4sun.jpl.nasa.gov/presentation.html (last access 31 Jan 2024). Such evaluations are also performed on candidate linelists developed by the HITRAN group to provide feedback on the performance of those linelists before they are adopted.

The solar linelist (Toon, 2022b) is completely empirical, based on high-resolution solar spectra measured by various instruments from the ground, balloon, and space. In the 4000 to 8000 cm$^{-1}$ spectral region covered by TCCON, the linelist is based primarily on ground-based Kitt Peak and TCCON spectra, with additional balloon-borne MKIV spectra from 40 km altitude

| Gas product | Freq. (cm$^{-1}$) | RMS (14) | RMS (20, V) | RMS (20, V + NV) | VSF (14) | VSF (20, V) | VSF (20, V+NV) |
|---|---|---|---|---|---|---|---|
| $X_{1CO_2}$ | 4830.3 to 4874.1 | 0.1785 | 0.1753 | **0.1496** | **1.0089** | 1.0109 | 1.0170 |
| $X_{wCO_2}$ | 6047.5 to 6096.5 | 0.0937 | **0.0918** | **0.0918** | 1.0179 | **1.0083** | **1.0083** |
| $X_{CO_2}$ | 6200.0 to 6240.0 | 0.1279 | 0.1248 | **0.1110** | 1.0194 | **1.0180** | **1.0180** |
| $X_{CO_2}$ | 6318.3 to 6360.7 | 0.1363 | 0.1325 | **0.1215** | 1.0203 | **1.0184** | 1.0212 |

**Table 3.** Results of test retrievals on known amounts of $CO_2$ in a cell with three different linelists. "(14)" indicates the GGG2014 linelist, "(20, V)" indicates the GGG2020 linelists *without* the non-Voigt lines discussed in §6.2, and "(20, V+NV)" indicates the full GGG2020 linelist (with non-Voigt lines included). The $X_{wCO_2}$ window does not have non-Voigt $CO_2$ lines, so its (20, V) and (20, V+NV) results are the same. "Gas product" indicates which of the TCCON products is retrieved in each frequency window, and "Freq." gives the span of that window. Note that these windows are used to fit laboratory cell spectra, and differ slightly from those used operationally by TCCON (given in Tables A2 and A3). The "RMS" columns list the root mean squared difference between observed and simulated spectra normalized by the continuum level. The "VSF" columns list the ratio of the retrieved $CO_2$ amount to the prior amount. Since these measure known $CO_2$ amounts in laboratory cells, VSF $\neq$ 1 indicates a systematic bias in the $CO_2$ line strengths. For both the RMS and VSF columns, the best values (closest to 0 for RMS and closest to 1 for VSF) is in bold.

up to 5600 cm$^{-1}$. To deduce which absorption features are solar, rather than telluric, we fit out the telluric spectrum as best we can. Remaining dips in the residuals are solar, unless they grow with airmass, in which case they are missing tellurics.

The solar linelist is not the same format as the HITRAN linelist. In addition to the line position, there are parameters representing the line center absorption depth, a doppler width and a Lorentz width, each for disk-center and disk-integrated cases, so seven parameters in total. A simple subroutine computes a solar pseudo-transmittance spectrum from these seven parameters, providing flexibility to model disk-center, disk-integrated, or intermediate cases. Since GGG2014 the improvements in the main TCCON region have been modest, adding new weak lines (< 0.1% depth).

The solar continuum is handled separately from the linelist in GGG. This is discussed in §7.

## 6.2 Non-Voigt lineshapes for O$_2$, CO$_2$, and CH$_4$

Absorption coefficients calculations were improved in GGG2020. In previous versions of GGG absorption coefficients were calculated using a Voigt spectral line shape. Numerous spectroscopic studies (e.g. Tran et al., 2013; Hartmann et al., 2009; Gordon et al., 2017) have shown that the Voigt line shape is insufficient for use with $CO_2$ and other molecules, so a more sophisticated line shape is required to improve the accuracy of the retrieval. So the quadratic speed-dependent Voigt (qSDV) with line mixing (LM) code from Tran et al. (2013) was implemented into the forward model of GGG (Toon, 2022a). Tables A2 and A3 list the frequency windows used in GGG2020, and contain columns identifying which windows include speed-dependent and line mixing lineshape information.

It was shown in Mendonca et al. (2016) that using the qSDV with first order LM and adopting the spectroscopic parameters from Devi et al. (2007b) for the $CO_2$ lines in the $CO_2$ window centered at 6220 cm$^{-1}$ and Devi et al. (2007a) for the window

centered at 6339 cm$^{-1}$ resulted in an up to 40% improvement to both spectral fit RMS and a reduction in the airmass depen-
dence of the retrieved XCO$_2$. For the CO$_2$ band lines in the window centered at 4850 cm$^{-1}$, the spectroscopic parameters
from Benner et al. (2016) are used with the qSDV and first order LM to calculate absorption coefficients. This resulted in
improving the quality of XCO$_2$ retrievals (i.e. reducing the spectral fit RMS) from this spectral region. New spectroscopic
studies aimed at improving CO$_2$ absorption coefficient calculations are ongoing. Recent studies like Hashemi et al. (2020) that
provide spectroscopic parameters for CO$_2$ can be tested with TCCON spectra to see if the retrievals can be improved.

TCCON CH$_4$ is retrieved from three windows that are composed of the P, Q, and R branches of the $2\nu_3$ CH$_4$ band. To
improve the forward model of GGG the spectroscopic parameters from Devi et al. (2015, 2016) are used to calculate the
absorption coefficients with the qSDV with full line mixing. Unlike CO$_2$ that uses first order line mixing requiring one extra
parameter to be added to the linelist per spectral line, CH$_4$ requires full line mixing. This requires spectroscopic parameters
from all coupled lines (i.e. a relaxation matrix) be used to calculate the effective spectral line parameters for each spectral
line. In previous versions of GGG, absorption coefficients could only be calculated by reading in spectroscopic parameters
line by line making it awkward to take into account full line mixing. GGG2020 has been updated to read in spectroscopic
parameters and the relaxation matrix (supplied with Devi et al. (2015, 2016)) at the same time for spectral lines that require
full line mixing. More details on how this is done are provided in Mendonca et al. (2017). The improved absorption coefficient
calculations for CH$_4$ lines for the $2\nu_3$ CH$_4$ band has improved the quality of the spectral fits and airmass dependence of the
retrieved XCH$_4$. The addition of full line mixing can be extended to other molecules to improve retrievals.

To improve the retrievals of O$_2$ columns, which are required to calculate $X_{\text{gas}}$, spectroscopic parameters for the O$_2$ singlet
delta band were retrieved by fitting cavity ring down spectra as detailed in Mendonca et al. (2019). The spectroscopic parame-
ters derived from the cavity ring down spectra were tested on TCCON spectra where they were shown to slightly improve the
quality of the spectral fit as well as greatly decrease the airmass dependence of the retrieved O$_2$ column. The study by Men-
donca et al. (2019) is the first to show the need for a spectral line shape that takes into account speed-dependence. Since then,
newer spectroscopic studies such as Tran et al. (2020) and Fleurbaey et al. (2021) have shown the need to take into account
Dicke narrowing and line mixing in order to fit new cavity ring down spectra in the O$_2$ singlet delta band. The spectroscopic
parameters of Mendonca et al. (2019), Tran et al. (2020), and Fleurbaey et al. (2021) were used to fit TCCON O$_2$ spectra in
Tran et al. (2021). The study showed that the newer spectroscopic parameters slightly improved the quality of the spectral fit
but they should also be assessed on how they impact the airmass dependence of retrieved O$_2$ columns.

This does mean that the standard 160-character wide HITRAN linelist product does not include all of the parameters re-
quired for these gases. GGG has always used a customized version of the HITRAN linelist. Therefore, this need for additional
parameters represents an increase in the complexity of our linelist strategy, but a continuation of the same approach to use the
best spectroscopic information from various sources, rather than a wholly new approach.

## 6.3 Empirical optimization of $O_2$ line widths

During pre-release testing, we found that a diagnostic quantity we call $X_{\mathrm{luft}}$ had a noticeable temperature dependence (Fig. 7a). $X_{\mathrm{luft}}$ is defined as:

$$X_{\mathrm{luft}} = \frac{\sum_k (1 - x_{\mathrm{H_2O},k}) \cdot n_{\mathrm{air},k} \cdot \Delta z_{\mathrm{eff},k}}{V_{\mathrm{O_2}}/f_{\mathrm{O_2}}} \tag{4}$$

where:

- $x_{\mathrm{H_2O}_k}$ is the water vapor wet mixing ratio for level $k$,

- $n_{\mathrm{air},k}$ is the ideal gas number density of air at level $k$ (calculated from temperature and pressure),

- $\Delta z_{\mathrm{eff},k}$ is an effective path length for level $k$ that accounts for the pressure-weighted contribution of that level and the surface pressure,

- $V_{\mathrm{O_2}}$ is the retrieved $O_2$ column (with the same integration as the numerator), and

- $f_{\mathrm{O_2}}$ is the mean dry mole fraction of $O_2$ in air, fixed at 0.2095 for GGG2020 (see §8.3.2 for discussion of accounting for the trend in $O_2$ dry mole fraction).

Conceptually, $X_{\mathrm{luft}}$ is a ratio of distinct two ways of calculating the column of dry air (one from surface pressure and the a priori $H_2O$ profile, and one from the column of $O_2$ retrieved in the singlet delta band—or put another way, it is the column-average dry mole fraction of dry air), and thus should not have a temperature dependence. Since dry mole fractions of $O_2$ in the atmosphere are highly constant over space and time, this implied that either temperature-dependence or the water broadening of the $O_2$ line widths in the forward model was incorrect, as the concentration of water in the atmosphere is generally correlated with temperature.

To disentangle the effect of temperature and water, we first examined data from the Darwin, Australia TCCON station. Darwin is located in the tropics, and so experiences greater water columns and a narrower range of temperatures than other TCCON sites (Fig. 8a,b). We chose approximately 14 months of data from Darwin when the instrument was performing well, and processed that year three times, with water broadening set to 1.0, 1.4, and 1.8 times that of the air broadening half width.

To identify the optimal strength for water broadening, we examined the slope of $X_{\mathrm{luft}}$ vs. water column in 10° SZA bins for each of these tests. Binning the data by SZA helps to separate the water dependence from airmass dependence. Figure 8c shows that a water broadening of 1.4 times that of air minimized the dependence of $X_{\mathrm{luft}}$ on water.

With the water broadening optimized, we turned to the temperature dependence of the $O_2$ line widths. Reducing the dependence of $X_{\mathrm{luft}}$ on temperature was the primary goal; however, we had to account for the interplay between the temperature and pressure dependence. In particular, our concern was that changing the temperature dependence of the $O_2$ line widths would introduce or increase an SZA dependence by changing the average line widths.

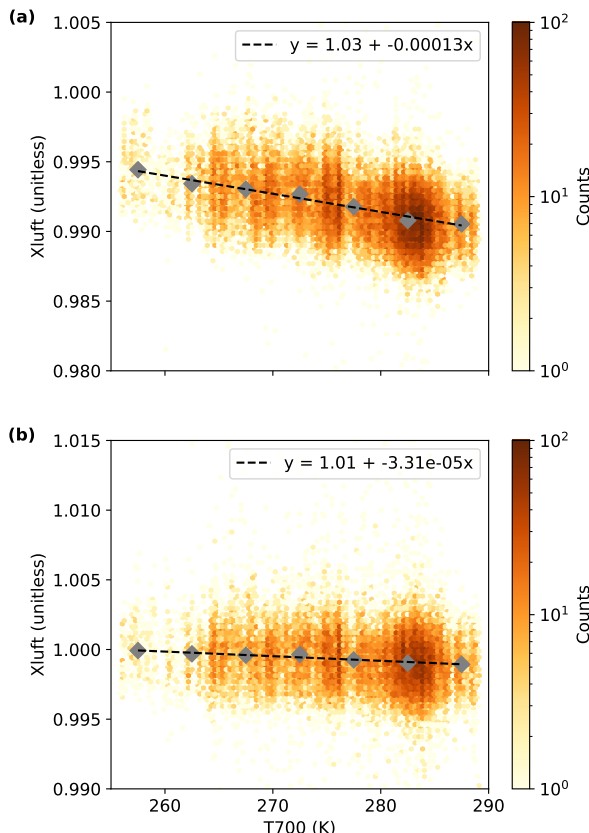

**Figure 7.** Correlation between $X_{\text{luft}}$ and temperature at 700 hPa **(a)** before and **(b)** after optimizing the $O_2$ line broadening in terms of its water, pressure, and temperature dependencies. Note that (a) is *not* from the previous TCCON data version (GGG2014), it is from a preliminary beta test of GGG2020. In both panels, the colored background is a 2D histogram, the gray diamonds mark the mean $X_{\text{luft}}$ in 5 K bins, and the black line is a linear fit to the gray diamonds. The data shown here is from the Lamont TCCON site between 2 Sep 2017 to 30 Sep 2018. Note that the $y$-axis limits shift between the panels; this is because the mean magnitude of $X_{\text{luft}}$ changed with the increase of $O_2$ line intensities (see text) between the tests plotted in the two panels. The slope is visually comparable between the panels, since the span of $X_{\text{luft}}$ is the same (0.025) in both panels.

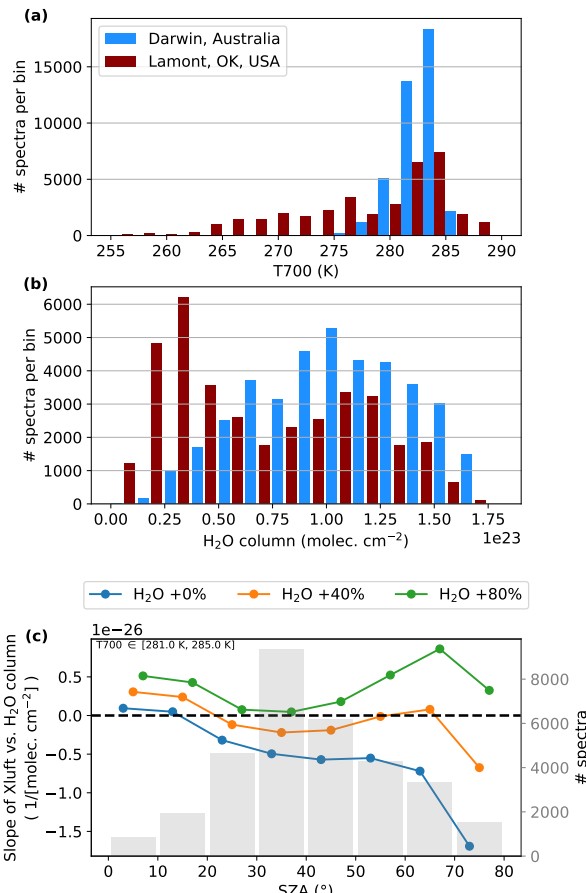

**Figure 8. (a)** Histogram of temperatures at 700 hPa at the Darwin (located at 12.5° S) and Lamont (at 36.6° N) TCCON sites. **(b)** Histogram of water column amounts at the same sites. **(c)** Slopes of $X_{\mathrm{luft}}$ vs. water column in 10° SZA bins at Darwin with water broadening of $O_2$ set as equal to, 40% greater, and 80% greater than air. The grey bars give the number of spectra in each bin. The Lamont data in (a) and (b) is from the period 2 Sep 2017 to 30 Sep 2018, and the Darwin data in all bins is from 21 Jul 2015 to 30 Sep 2016.

Our solution was to simultaneously adjust both the temperature and pressure dependence of the $O_2$ line widths. To find the optimal combination of these coefficients, we minimized a cost function of three quantities. For each quantity, we tested how the results changed using a different collection of TCCON sites:

1. The average magnitude of the $X_{luft}$ vs. temperature at 700 hPa (T700) slope across various combinations of 1–3 of the East Trout Lake, Lamont, and Park Falls sites.

2. The variance of the $X_{luft}$ vs. SZA slopes across the Darwin, East Trout Lake, Lamont, and Park Falls sites.

3. The variance of the magnitude of $X_{luft}$ across the same sites as #2.

Our rationale was that the temperature dependence of $X_{luft}$ was the most important error to eliminate, thus minimizing its magnitude took priority. T700 is taken from the a priori meteorology data, and was chosen on the assumption that this is a reasonable metric for temperature variations in the free troposphere ($\sim 800$ to 200 hPa) containing the majority ($\sim 60\%$) of the $O_2$ column. We then minimized the variance in slopes of $X_{luft}$ vs. SZA across different TCCON sites because GGG already has a well-tested program to remove spurious SZA dependencies in the output $X_{gas}$ products, so long as those dependencies are the same across sites. While minimizing the magnitude of the SZA dependence itself would have been preferable, we were not certain there would be enough flexibility in the $X_{luft}$-$O_2$ spectroscopy relationship to simultaneously minimize the temperature and SZA dependencies. Similarly, we minimized the variance in $X_{luft}$ itself because the average magnitude of $X_{luft}$ depends on the strengths of the $O_2$ lines, rather than the pressure and temperature effects on line width adjusted in this initial experiment. Therefore, while we ideally want $X_{luft} = 1$, this first step was not optimizing the spectroscopic parameters that can achieve that. We do adjust the $O_2$ line strengths separately, as noted at the end of this section.

To carry out this optimization, we ran approximately one year of data from four TCCON sites (Darwin, Australia; East Trout Lake, Canada; Lamont, OK, USA; Park Falls, WI, USA) multiple times. In each test, we scaled the temperature dependence, pressure dependence, or both of all lines in the $O_2$ band, covering a reasonable range of estimates from the literature. We could then interpolate between these test runs to estimate the three cost function quantities for any pressure/temperature broadening coefficients, and from that find the combination of coefficients that minimized the overall cost function. Note that we did not use Darwin data to calculate the $X_{luft}$ versus T700 slopes for the cost function, as the small range of temperatures that Darwin experiences (Fig. 8a) make it difficult to get reliable fits versus temperature.

The results of the optimization are shown in Fig. 9. Figure 9a shows how the three criterion described above (slope of $X_{luft}$ vs. T700, variance in slope of $X_{luft}$ vs. SZA, variance in $X_{luft}$) varied across the tests performed with different pressure and temperature broadening coefficients. The values are normalized to their respective pre-optimization values. We found that the best combination of coefficients reduced the slope of $X_{luft}$ vs. T700 by 82%, the variance in $X_{luft}$ vs. SZA slopes across TCCON sites by 89%, and the variance in $X_{luft}$ itself by 49%. The optimized air broadening half widths and temperature dependence coefficients for GGG2020 are shown in Fig. 9, panels b and c respectively, with GGG2014 values for comparison. The air broadening half widths were increased by 0.25% and the temperature dependence coefficients were decreased by 6.77%. The effect on the $X_{luft}$ vs. T700 relationship is shown in Fig. 7b, where although not reduced to zero, the slope is reduced by a factor of 4 compared to its pre-optimization value.

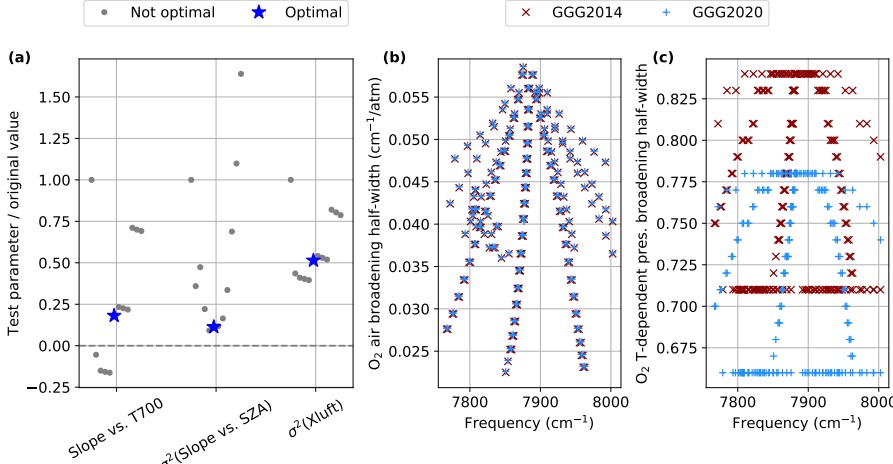

**Figure 9.** Result of the $O_2$ spectroscopy optimization. **(a)** The values of each criterion for each test using different values of pressure and temperature broadening coefficients. The values are normalized to their values in the baseline test (before optimizing the $O_2$ spectroscopy). The points within each parameter are spread horizontally for clarity. **(b)** The air broadening half widths used in GGG2020 (after optimization) compared with GGG2014. The mean GGG2020/GGG2014 ratio is 1.0025, so the points are barely different on this scale. **(c)** As (b), but for the temperature broadening coefficient. The mean GGG2020/GGG2014 ratio is 0.9323.

Finally, the $O_2$ line intensities were increased by $\sim 1\%$ to bring $X_{\text{luft}}$ closer to 1. This effect is apparent in Fig. 7, where the pre-optimization values are between 0.990 and 0.995, but the post-optimization $X_{\text{luft}}$ in panel b is near 1. Across the TCCON network, we determined that the median $X_{\text{luft}}$ is 0.999; therefore we use that as the benchmark for ideal $X_{\text{luft}}$.

## 7    Continuum fitting

TCCON spectra are a combination of narrow features due to solar and telluric absorptions superimposed on the much broader spectral responses of the instrument[3] and the solar Planck function (the continuum). To accurately fit the telluric features of interest, all other components of the spectrum must be accurately modelled simultaneously. Since TCCON spectra are not radiometrically calibrated, the continuum can vary from instrument to instrument or even from day to day (if optical components are inserted or replaced) and therefore a general approach was needed to model the continuum. Prior to GGG2014, the continuum was fitted with only two terms (mean and slope) over the $<100\,\text{cm}^{-1}$ wide windows used to retrieve atmospheric gases. To make use of wider spectral windows, it became necessary to include additional higher-order terms in the model of the continuum, to account for optical components within the instrument (e.g., detectors, optical filters, beamsplitter, etc.) that induce curvature in the spectral response (e.g., Kiel et al., 2016b). In GGG2014, we implemented the ability to fit higher order

---

[3]Here, by "spectral responses of the instrument," we mean an instrument-specific response which can be characterized as a frequency-dependent vector that multiplies the incoming solar spectra. This is distinct from the ILS, which is instead best considered as an instrument-specific vector that convolves the incoming solar spectra.

polynomials to the continuum level using discrete Legendre polynomials, although this capability was not uniformly used in the GGG2014 TCCON data processing (Wunch et al., 2015). (We use Legendre polynomials because they are orthogonal, whereas standard polynomials are not.) Higher order Legendre polynomials are now used widely in the GGG2020 spectral windows to better account for continuum shape changes between instruments and over time. The continuum curvature fitting option is not intended to fit out spectroscopic deficiencies; they will be airmass-dependent and so should be fixed separately. The default polynomial order in GGG2020 for each window has been chosen to capture the continuum shapes of all sites in GGG2020 and reduce the spectral residuals without over-fitting the spectrum. The default order for each window is listed in Tables A2 and A3.

## 7.1   Channel Fringe Fitting

Parallel optical surfaces delay a small fraction of the transmitted beam, which subsequently interferes with the main, un-delayed beam, resulting in a small periodic modulation of the spectral signal. This modulation has an amplitude of $R^2$ where R is the reflectivity of each surface, and a period of $(2 \cdot n \cdot d \cdot \cos\theta)^{-1}$ cm$^{-1}$, where $n$ in the refractive index of the optic, $d$ is its thickness (in cm) and $\theta$ is the angle to the normal.

For decades, GFIT has the capability to fit a channel fringe to determine its amplitude (as a fraction of the continuum), its period, and its phase, and then remove it from the measured spectrum during the spectral fitting. This capability was not used by TCCON until GGG2020, when spectral fits from some sites were noticed to exhibit the tell-tale periodicities in the residuals. Left untreated, channel fringes can seriously bias the retrieved gas amounts, by an amount that can vary from instrument to instrument and even over time for a single instrument, e.g., if its temperature changes.

An important code change for GGG2020 was to prevent channel fringes from being mistaken for higher-order continuum terms. This was much less of a problem for GGG2014 when we only ever fitted a straight line to represent the continuum. But now, if a particular wavelike feature in the continuum could be fitted by a higher order polynomial or by a channel fringe, this tends to slow down convergence as the continuum fitting and channel fringe fitting vie with each other. To prevent this, a lower limit was imposed on the channel fringe period that was fittable in a given window, such that it always was narrower than the periodicities in the continuum fitting polynomial. So if we are fitting an $N$-term polynomial to the spectrum (called the number of continuum basis functions, or $N_{\mathrm{CBF}}$), in a window of width $w$ cm$^{-1}$, then the period of the fitted channel fringes must be less than $w/(N_{\mathrm{CBF}} - 1)$.

Diagnostics to detect channel fringes are reviewed as part of the quality control process before TCCON data is made public. Any channel fringes detected will be removed by adjusting the fitting before the data is released to the public archive, though this is extremely uncommon.

## 8   Post-retrieval data processing

GGG incorporates several post-retrieval steps to (1) collate and average data (§8.2) from the individual retrieval windows into the final $X_{\mathrm{gas}}$ products and (2) correct post hoc for known errors in the forward model. There are two corrections. The first is

an airmass-dependent correction (§8.1), which aims to eliminate spurious dependence of $X_{\text{gas}}$ quantities on SZA. The second is an in situ-based, or airmass-independent correction (§8.3), which aims to eliminate the mean bias in $X_{\text{gas}}$ values arising from incorrect spectroscopic line strengths. These corrections are calculated from data that includes all improvements discussed in the preceding sections.

In the following sections, the post processing steps are presented in the order in which they are applied in GGG2020.

## 8.1 Updated airmass dependence correction

In the limit of no variation in trace gas dry air mole fraction, $X_{\text{gas}}$ quantities are independent of atmospheric path length, as the change in column density due to path length is multiplicative and so will cancel out between the target gas in the numerator and $O_2$ in the denominator. However, a spurious dependence of $X_{\text{gas}}$ on airmass can arise from errors in the spectroscopic forward model.

### 8.1.1 Changes to airmass correction approach

GGG2020, like GGG2014, applies a post hoc correction to the $X_{\text{gas}}$ values to remove airmass dependences. This correction is applied to each $X_{\text{gas}}$ value. It has a similar form to that in Appendix A of Wunch et al. (2011):

$$f_c = \left( \frac{\text{SZA} + g}{90 + g} \right)^p - \left( \frac{45 + g}{90 + g} \right)^p \tag{5}$$

and use this to correct the $X_{\text{gas}}$ value as

$$X_{\text{gas,corr}} = \frac{X_{\text{gas,raw}}}{1 + \text{ADCF} \cdot f_c} \tag{6}$$

In Eq. (6), ADCF (standing for airmass dependent correction factor) is a coefficient for each gas (in GGG2014) or each window (in GGG2020). In Eq. (5), SZA is the solar zenith angle in degrees and $g$ and $p$ are coefficients chosen to best represent the SZA-dependent behavior. This form was chosen to normalize to a 90° window centered on $(45 + g)°$. While the basic approach is the same in GGG2020 as it was in GGG2014, we made four changes to the implementation:

1. In GGG2014, column densities from different spectral windows used to retrieve a target gas were averaged first, then a single airmass correction applied to each gas. In GGG2020, each spectral window is airmass corrected first, then the resulting $X_{\text{gas}}$ values are averaged.

2. In GGG2014, $g = 13$ and $p = 3$ for all gases. In GGG2020, different values of $g$ and $p$ were selected for each window.

3. In GGG2014, only data from 3 TCCON sites (Park Falls, Lamont, and Darwin) were used to compute the ADCFs. For GGG2020, we use 18 sites' data.

4. In GGG2014, we did not examine the ADCF for temperature dependence. We do in GGG2020 and attempt to account for that in how we select the final ADCF values.

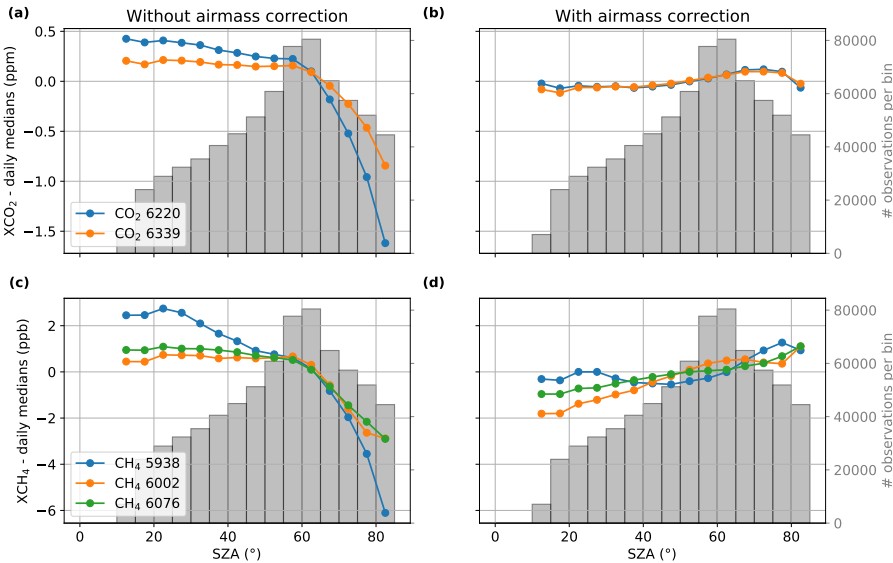

**Figure 10.** Variation of **(a,b)** the two $CO_2$ and **(c,d)** three $CH_4$ windows used by TCCON with SZA. **(a)** and **(c)** are *without* the airmass correction applied, **(b)** and **(d)** are with the correction applied. In all panels, the $y$-axis is column-average dry mole fraction of $CO_2$ or $CH_4$ derived from a single spectral window, with the central wavenumber given in the legend. The $y$ values have the daily median values subtracted (to remove day-to-day variability), and each point represents the median of all such values in a 5° SZA bin. The grey bars give the number of observations in each 5° SZA bin (this is the same in all panels).

The rationale for the first change is clear from Fig. 10. The standard TCCON $CO_2$ and $CH_4$ products are derived from two and three spectral windows, respectively. Although the overall SZA dependence has a similar shape for all windows of a given gas, there are clear differences in low and high SZA behavior. Thus, we decided to apply an SZA dependent correction to individual windows, rather than the average $X_{gas}$ value. The right panels of Fig. 10 show that applying the airmass correction significantly reduces the SZA dependence of the data.

The rationale for the second change is that we do not know a priori the best form to represent the airmass dependence in any given window. For GGG2020, we used data from the Darwin TCCON site for all of 2015 to choose the values of $g$ and $p$ for each window. We used Darwin because, as a tropical site, it sees a wide range of SZAs (useful for examining SZA dependence) and water columns (useful to check for water effects on the derived airmass dependence). We used 2015 data because the instrument at Darwin was well aligned during that year.

To understand how $g$ and $p$ were determined, we must first explain how the ADCF in Eq. 6) is calculated for a given $g$ and $p$. The ADCF is calculated by fitting the following function to each day's data:

$$f(t, \mathrm{SZA} | c_{\mathrm{mean}}, c_{\mathrm{asym}}, c_{\mathrm{ADCF}}) = c_{\mathrm{mean}} + c_{\mathrm{asym}} \cdot \sin\left(2\pi(t - t_{\mathrm{noon}})\right) + c_{\mathrm{ADCF}} f_c \tag{7}$$

where $t$ and $t_{\text{noon}}$ are the measurement time and solar noon time (in day of year), $f_c$ is the polynomial defined in Eq. (5), and $c_{\text{mean}}$, $c_{\text{asym}}$, and $c_{\text{ADCF}}$ are the fitted coefficients. This equation assumes that symmetrical variation of $X_{\text{gas}}$ values around noon (fit by $f_c$) are due to spectroscopic errors and real variations throughout the day are antisymmetrical and will be fit by the $c_{\text{asym}}$ term. The coefficients and their errors are calculated with a weighted least squares fit using the individual windows' $X_{\text{gas}}$ uncertainties (calculated from the spectral residuals of the target gas and $O_2$) as the weights. The ADCF for a given window is the error-weighted mean of all days' $c_{\text{ADCF}}$ values.

### 8.1.2 Determination of ADCF coefficients

To find the optimal $g$ and $p$ values, we derived ADCFs for five subsets of the 2015 Darwin data (data with SZA $> 20°$, $30°$, $40°$, $50°$, and $60°$, all with $H_2O$ column $< 1.1 \times 10^{23}$ molec. cm$^{-2}$) for values of $g$ between $-45$ and $+45$ and $p$ between 1 and 6. We then find the combination of $g$ and $p$ that gives the smallest standard deviation of the ADCF across all five subsets and choose that as the optimal combination. This approach assumes that the values of $g$ and $p$ (and thus the form of $f_c$) which best capture the airmass dependence of a particular window will have the smallest change in ADCF as smaller subsets of data are fit.

This procedure is illustrated for the two TCCON $CO_2$ windows in Fig. 11. In the top panels, the gray lines show the variation in ADCF with the minimum SZA in the subset of data fit to; each line represents one combination of $g$ and $p$. It is clear that the variation in ADCF is much greater for some combinations of $g$ and $p$ than others. The contour plots in Fig. 11 show the standard deviation of ADCF for each $g$ and $p$ combination. In both windows, there is a clear minimum valley. The white stars in the contour plots and thicker black lines in the upper panels show the $g$ and $p$ combination with the smallest standard deviation.

The final step in selecting ADCFs for GGG2020 was to account for potentially spurious temperature dependence in the $X_{\text{gas}}$ values. As we saw with $O_2$ in §6.3, incorrect temperature dependence in the line widths introduces a temperature dependence in retrieved $X_{\text{gas}}$, which could alias into the airmass dependence. While we acknowledge that such temperature dependence of the ADCFs could be due to a real change in the atmosphere, we believe this to be unlikely for two reasons. First, the ADCF is constructed to account only for variations in Xgas that are symmetric around solar noon, and generally changes in atmospheric composition are not perfectly symmetric around solar noon. Second, as we show in Fig. 13, different windows for the same gas have different relationships between the ADCF and temperature. A real change in atmospheric composition would be more likely to show up in all windows for a given gas.

To check this, we derived ADCFs from data from 18 TCCON sites, using two month long subsets of data to sample different temperatures. Figure 13 shows how the $CH_4$ ADCFs vary with potential temperature averaged between 500 and 700 hPa ($\theta_{\text{mid}}$) as an example. (Figure 12 shows how $\theta_{\text{mid}}$ and T700 relate to assist comparisons with Fig. 7.) Here, we see that the 6002 cm$^{-1}$ and 6076 cm$^{-1}$ windows' ADCFs have no or little temperature dependence (Fig. 13b,c), but the 5938 cm$^{-1}$ window has a clear temperature dependence. For each window, we use the value of the fit to this data at $\theta_{\text{mid}} = 310$ K as the final ADCF value. 310 K was chosen as it is approximately the midpoint temperature for the TCCON network, as can be seen in Fig. 13.

The magnitude of this temperature dependence varies from gas to gas: the primary TCCON $CO_2$ windows have almost no slope, while the $N_2O$ windows have slopes of ADCF vs. $\theta_{\text{mid}}$ similar to or larger than the $CH_4$ 5938 window. We plan to

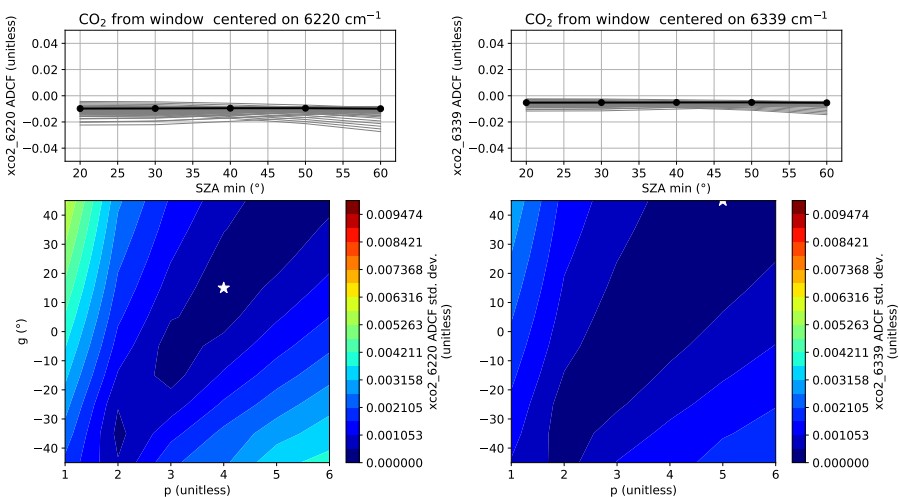

**Figure 11.** Example of how $g$ and $p$ in Eq. (5) were chosen for the two TCCON $CO_2$ windows. The left two panels are for the $CO_2$ window centered at 6220 cm$^{-1}$ and the right two for the window at 6339 cm$^{-1}$. The line plots at the top show how the value of the ADCF changes as we increase the lower limit in SZA for the data fit to. Each gray line represents one combination of $g$ and $p$, with the black line representing the combination with the smallest standard deviation in the ADCF. The contour plots show the standard deviation of the ADCF across different minimum SZAs for each combination of $g$ and $p$. The white star represents the combination with the smallest standard deviation; it corresponds to the test show with the black line in the line plots.

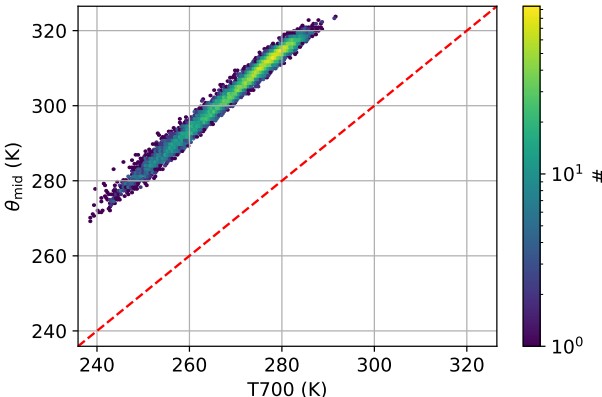

**Figure 12.** A heatmap of the relationship between $\theta_{\mathrm{mid}}$ and T700, taken from the Park Falls TCCON data. The red dashed line denotes the 1:1 line.

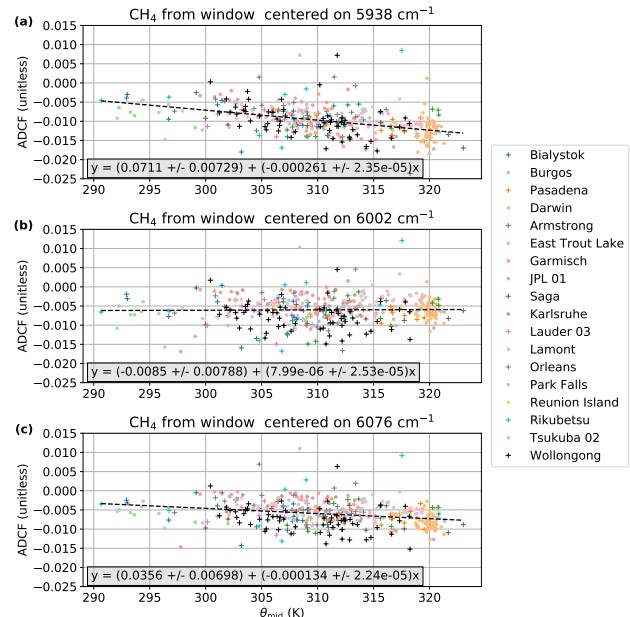

**Figure 13.** ADCFs derived from two month periods from 18 sites throughout the TCCON network versus mean potential temperature between 500 hPa and 700 hPa over the same two month period. Each panel is one of the TCCON $CH_4$ windows. The text inset in each panel gives the intercept and slope of the robust fit through the data shown by the black dashed line.

investigate these temperature dependence behaviors more thoroughly in the next major GGG version and identify spectroscopic improvements that will reduce or eliminate this behavior using a similar approach to that described for $O_2$ in §6.3.

### 8.1.3 Fitting windows excluded in GGG2020

Based on the ADCF analysis, several spectral windows were excluded from the TCCON GGG2020 product. Figure 14 shows the ADCF versus $\theta_{mid}$ plots for two CO windows and two weak $CO_2$ windows. The CO window centered on 4233 cm$^{-1}$ (Fig. 14a) has slightly stronger temperature dependence and clearly larger scatter than the 4290 cm$^{-1}$ CO window (Fig. 14b). We suspect this is due to water interference; the 4233 cm$^{-1}$ CO window has more water lines in it than the 4290 cm$^{-1}$ window. We examined the spectral residuals in both CO windows to try to identify and correct the water interference, but were not able to reduce it to satisfactory levels. Thus, in GGG2020, the $X_{CO}$ product relies on only the 4290 cm$^{-1}$ window.

Similarly, the new $X_{wCO_2}$ product was planned to use two windows, one centered on 6073 cm$^{-1}$ and another on 6500 cm$^{-1}$. However, as shown in Fig. 14c and 14d, the 6500 cm$^{-1}$ window's ADCF have more scatter and stronger temperature dependence than the 6073 cm$^{-1}$ window. As the 6500 cm$^{-1}$ also has more water interference than the 6073 cm$^{-1}$ window, we elected to use only the 6073 cm$^{-1}$ window.

Lastly, we also removed a number of HCl windows. TCCON instruments use HCl lines to assess instrument alignment with an HCl cell that can be illuminated by the solar beam or an internal lamp. TCCON used 16 windows to measure HCl

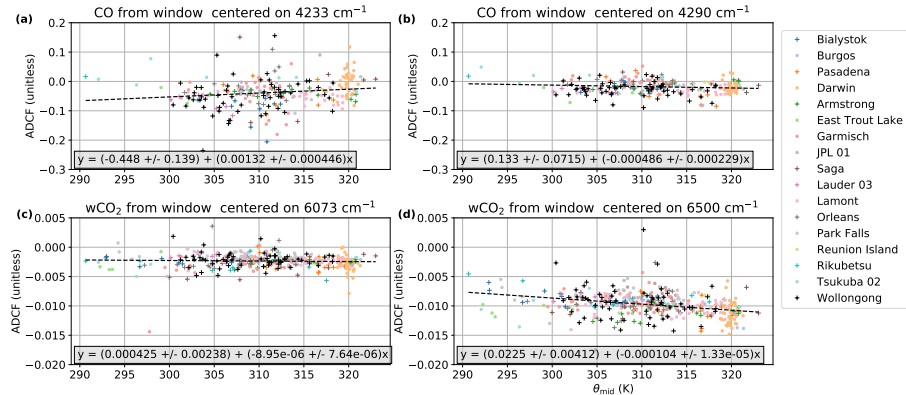

**Figure 14.** Similar to Fig. 13, except for two CO windows **(a, b)** and two weak $CO_2$ windows **(c, d)**.

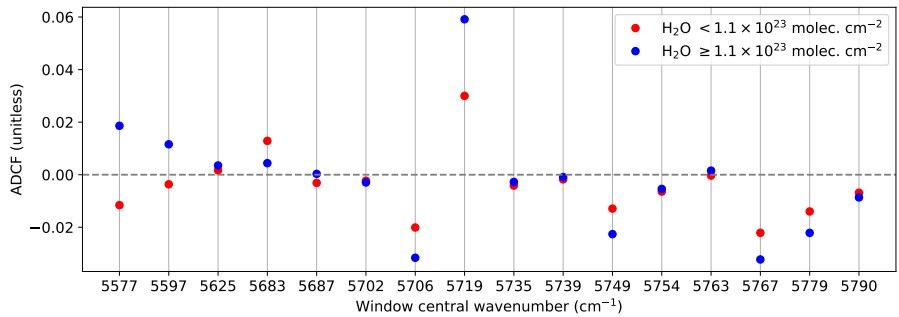

**Figure 15.** ADCF calculated for each HCl window from 2015 Darwin data for two data subsets with different amounts of water in the atmosphere.

in GGG2014, but like the CO and $wCO_2$ windows, many of these have water absorption lines in them. We can diagnose unaccounted for water interference by computing the ADCFs for each HCl window from Darwin 2015 data, split by the
575 amount of water in the column. The result is shown in Fig. 15. Most of the GGG2014 windows have a clear difference in ADCF with small or large water column amounts. Based on this, we chose to only retain the 5625, 5687, 5702, 5735, and 5739 $cm^{-1}$ windows. Most of the windows removed clearly have a water interference. The 5754 and 5763 $cm^{-1}$ windows are special cases. The 5754 $cm^{-1}$ window was rejected because its airmass dependence is slightly more negative than the retained windows. The 5763 $cm^{-1}$ window was rejected because it exhibits a clear temperature dependence in the window-to-window
scale factors (§8.2).

## 8.2 Updated window-to-window averaging

Many gases retrieved by GGG are retrieved in more than one spectral window. GGG retrieves the column amount in each window separately, then averages together the columns with similar averaging kernels to produce a mean value. Specifically,

$$\overline{y}_i = \frac{\sum_j s_j y_{ij}/\epsilon_{ij}^2}{\sum_j s_j^2/\epsilon_{ij}^2} \tag{8}$$

where subscript $j$ represents the spectral window. That is, the average value for the $i$th measurement ($\overline{y}_i$) is an error weighted average of the individual windows' column amounts ($y_{ij}$, with errors $\epsilon_{ij}$) with a mean bias in each window removed by the per-window scale factor, $s_j$. The errors $\epsilon_{ij}$ are the posterior errors in the $X_{\mathrm{gas}}$ amounts as calculated from the spectral residuals.

In GGG2014, the $s_j$ values were determined online, using an iterative process that minimizes the differences between $y_{ij}$ and the corresponding $s_j \overline{y}_i$ values. While this calculates $s_j$ values that best fit the data being averaged, it means that how the windows are combined depends on how much data is averaged at once—processing a month could give different results than processing a year of data, for example. Thus, while GGG2020 retains the capability to compute the $s_j$ values on-the-fly, the $s_j$ values are prescribed for standard TCCON processing, and all sites use the same $s_j$ values.

To determine the standard TCCON $s_j$ values, we used a very similar approach to how we derived the ADCFs in §8.1. Specifically, we calculated the $s_j$ values for two month subsets of data from the same 18 TCCON sites as in §8.1 and fit these values versus $\theta_{\mathrm{mid}}$. As with the ADCFs, we used the values of the fit at $\theta_{\mathrm{mid}} = 310$ K as the final choices of $s_j$.

## 8.3 Updated in situ bias correction

As in GGG2014, the GGG2020 $X_{\mathrm{CO_2}}$, $X_{\mathrm{CH_4}}$, $X_{\mathrm{N_2O}}$, and $X_{\mathrm{H_2O}}$ products are tied to standard scales by in situ aircraft, balloon, and/or radiosonde measurements to remove any mean multiplicative bias introduced by error in absorption line intensity. As the absorption of a gas is the product of its column density and spectroscopic cross section, a bias in the mean line intensity (and therefore the cross section) will by definition lead to a multiplicative bias in the simulated absorption and thus the retrieved column density. Unlike GGG2014, $X_{\mathrm{CO}}$ in GGG2020 is not tied to in situ measurements, due to previous work that found the difference between TCCON $X_{\mathrm{CO}}$ and both NDACC (Kiel et al., 2016a) and MOPITT (Hedelius et al., 2019) $X_{\mathrm{CO}}$ was approximately the magnitude of the in situ correction. Those analyses suggest that the GGG2014 7% CO scaling was likely spurious. However, we do evaluate $X_{\mathrm{CO}}$ against a subset of in situ data from AirCore only below.

Comparison of TCCON data against in situ data follows the following steps:

1. identify in situ vertical profiles in available data and convert to a standardized file format,

2. extend the profiles' tops to 70 km altitude using the standard GGG2020 priors (shown in Laughner et al. (2023) to have good agreement with in situ profiles in the stratosphere) and to the surface by extrapolation or use of surface data,

3. match profiles to available TCCON spectra,

4. run custom retrievals using the matched profiles as the a priori trace gas profile, and

5. compare integrated in situ $X_{gas}$ values against matched TCCON data, accounting for TCCON vertical sensitivity.

Points 1–4 are described in detail in Appendix C. Briefly, we use profiles from:

- the GlobalviewPLUS 5.0 $CO_2$ (Cooperative Global Atmospheric Data Integration Project, 2019) and GlobalviewPlus 2.0 $CH_4$ ObsPack (Cooperative Global Atmospheric Data Integration Project, 2020) products,

- AirCore balloon measurements (Tans, 2009; Karion et al., 2010) flown by NOAA (v20201223, Baier et al., 2021) at multiple TCCON sites and by FMI/LSCE/RUG at the Sodankylä, Finland (Kivi and Heikkinen, 2016) and Nicosia, Cyprus (Rousogenous, in prep) TCCON sites,

- the Infrastructure for Measurement of the European Carbon Cycle (IMECC) campaign,

- Profiles over the Manaus, Brazil TCCON site (Dubey et al., 2016),

- ARM radiosondes over the Darwin, Australia (Deutscher et al., 2010) and Lamont, OK, USA TCCON sites

$CH_4$ profiles have an additional correction to the stratospheric levels obtained from the GGG2020 priors, see §C3 for details. We have addressed the recent change of $CO_2$ data from the X2007 to X2019 WMO scales, which will be covered in §8.3.2. Due to the relative sparsity of $N_2O$ profiles, GGG2020 TCCON $N_2O$ products were evaluated against surface $N_2O$ data and a different approach, which will be covered in §8.3.3. The number of usable profiles for each gas is given in Table 4.

The use of ObsPack data represents a slight methodological change compared to GGG2014. Most of the in situ aircraft profiles used for the GGG2014 in situ correction are included in the ObsPack, and switching to the ObsPack instead of individual campaigns' data files will allow us to use the same tools to ingest future new profiles added to the ObsPack. This also allows us to benefit from the data curation and quality control efforts of the ObsPack team. With the larger number of profiles now available (especially for $CO_2$), we are able to test for correlations with potential sources or metrics of bias. However, the primary purpose of the in situ comparison remains to tie TCCON (and through TCCON, satellite) GHG data to the same metrological scales as in situ GHG data.

### 8.3.1 $CO_2$, $CH_4$, CO, and $H_2O$ in situ comparisons

The first step in comparing TCCON $X_{CO_2}$, $X_{CH_4}$, $X_{CO}$, or $X_{H_2O}$ to their respective in situ profiles is to match each in situ profile to temporally proximate, good quality TCCON retrievals. For this step, we define custom quality filters. A TCCON retrieval is considered to be good quality in this context if:

- Fractional variation in solar intensity (FVSI) is $\leq 0.05$. This is the standard deviation of solar intensity divided by the average solar intensity during the $\sim 80$ s long scan , and filters out observations impacted by intermittent clouds.

- Solar zenith angle (SZA) is $\leq 80°$. This avoids observations at large airmasses, where spectroscopic errors can be more pronounced.

- The unscaled $X_{\text{gas}}$ value is $> 0$ mol mol$^{-1}$. A negative retrieved value is unphysical, and the distribution of retrieved values should not be large enough to make negative values a reasonable part of it.

- The $X_{\text{gas}}$ error is $< 2\epsilon_{\text{median}}$, where $\epsilon_{\text{median}}$ is the median error for that $X_{\text{gas}}$ across all the spectra used for the given gas. This limits for observations where the observed spectra was fit reasonably well.

- The median $X_{\text{luft}}$ (see Eq. 4) for a comparison is between 0.996 and 1.002. $X_{\text{luft}}$ and this rational are explained near the end of this subsection.

For each in situ profile, we require at least 30 TCCON observations (each $\sim 80$ s) passing these quality checks within a certain window of time around the corresponding profile's lowest altitude measurement. Our initial window is $\pm 1$ hour. If 30 points meeting these criteria are not present within $\pm 1$ hour, we increase both the time window and the allowed $X_{\text{gas}}$ error, trying the combinations ($\pm 1$ hr, $< 2\epsilon_{\text{median}}$), ($\pm 2$ hr, $< 3\epsilon_{\text{median}}$), and ($\pm 3$ hr, $< 4\epsilon_{\text{median}}$). We use the smallest of these time/error window that yields 30 passing TCCON observations, but if a profile does not have 30 passing TCCON observations in the ($\pm 3$ hr, $< 4\epsilon_{\text{median}}$) range, it is removed from the comparison.

The remaining in situ profiles are integrated following Wunch et al. (2010), where the integrated in situ $X_{\text{gas}}$ value is calculated as:

$$X_{\text{gas,insitu}} = I(\gamma \mathbf{x}_a, \mathbf{p}, \mathbf{x}_{\text{H}_2\text{O}}) + I(\delta \mathbf{x}, \mathbf{p}, \mathbf{x}_{\text{H}_2\text{O}}) \tag{9}$$

where

- $\mathbf{p}$ is the vector of pressure at each profile level

- $\mathbf{x}_{\text{H}_2\text{O}}$ is the vector of water dry mole fractions at each profile level

- $\gamma \mathbf{x}_a$ is the TCCON posterior profile (i.e. the prior times the retrieved VMR scale factor $\gamma$)

- $\delta \mathbf{x}$ is the difference between the in situ ($x_{\text{insitu},i}$) and TCCON posterior ($x_{a,i}$) profiles, modified by the TCCON averaging kernel ($a_i$): $\delta x_i = a_i(x_{\text{insitu},i} - \gamma x_{a,i})$

$I$ represents the pressure-weighted integration function:

$$I(\mathbf{x}, \mathbf{p}, \mathbf{x}_{\text{H}_2\text{O}}) = \frac{\sum_i x_i \cdot dp_i \cdot D_i}{\sum_i dp_i D_i} \tag{10}$$

$$D_i = g_i \cdot M_{\text{air}} \cdot \left(1 + x_{\text{H}_2\text{O},i} \cdot \frac{M_{\text{H}_2\text{O}}}{M_{\text{air}}}\right) \tag{11}$$

where

- $dp_i$ represents the pressure thickness of layer $i$

– $g_i$ represents the acceleration from gravity at layer $i$,

– $M_{air}$ and $M_{H_2O}$ represent the mean molecular masses of dry air and water, respectively.

The integrated in situ $X_{gas}$ values are compared against the median of the TCCON $X_{gas}$ values from the matched observations. The TCCON $X_{gas}$ values used here have the airmass correction (§8.1) and window-to-window averaging (§8.2) applied.
Because we expect the bias in the TCCON data to arise from incorrect absorption line strengths or broadening coefficients, it should be a multiplicative bias. Therefore, we calculate an uncertainty-weighted mean of the TCCON/in situ $X_{gas}$ values to derive the bias correction. We consider five sources of uncertainty.

1. Measurement error in the in situ data.

2. Uncertainty from the unmeasured portion of the free troposphere. (Will be zero if the in situ vertical profile extends through the tropopause.)

3. Uncertainty from the unmeasured portion of the stratosphere.

4. Random error in the TCCON observations.

5. Bias in the TCCON observations from instrument misalignment or similar hardware concerns.

The calculation of each term and how they are combined for the error bars in Fig. 16 is detailed in Appendix C6.

The results of the TCCON-in situ comparison are shown in Fig. 16. In this plot, the $y$-axes are the ratio of TCCON to in situ $X_{gas}$ amounts and the $x$-axes show $X_{luft}$ (see Eq. 4 in §6.3 for the definition). We will return to the significant of $X_{luft}$ shortly. The use of TCCON to in situ ratios to derive the in situ correction is equivalent to the best fit lines forced through the origin used in Wunch et al. (2010), as the best fit line through the origin is essentially the mean TCCON to in situ ratio. As in Wunch et al. (2010), a ratio (or slope in Wunch et al. (2010)) $> 1$ indicates TCCON $X_{gas}$ values are biased high relative to in situ, and vice versa for ratios $< 1$. The use of ratios directly in Fig. 16 allows us to more clearly identify outliers and evaluate the correlation of the TCCON vs. in situ bias with other variables, such as $X_{luft}$ here.

The ratios from Fig. 16 indicate that the mean biases are within approximately 1% of unity in all cases, with water being the furthest from unity at 0.9883 ($-1.17\%$). The differences among the $CO_2$ products are interesting; the standard $CO_2$ product is biased about 1% high before correction (which is in line with expected uncertainties for the $CO_2$ lines), while the other two $CO_2$ products are much closer to unity (0.08% for $wCO_2$ and 0.14% for $lCO_2$). This suggests that the absorption coefficients in these latter two windows are more accurate than in the standard TCCON windows (which are centered on 6220 and 6339 cm$^{-1}$). However, as the wCO2 and lCO2 are more sensitive to the upper and near-surface atmosphere, respectively, it may be that this reflects other factors, such as the accuracy of the a priori temperatures at those levels.

Additionally, we note the TCCON $X_{CO_2}$ product changed from being 1% low compared to in situ (pre-in situ correction) in Wunch et al. (2015) to 1% high here. We would expect this to be due to changes in spectroscopy, such as an average decrease in $CO_2$ line strengths or increase in $O_2$ line strengths. However, we are in the process of conducting a full attribution study for all the component changes between GGG2014 and GGG2020, and reserve a final conclusion until that is complete.

The CO comparison (Fig. 16e) suggests that, without scaling, the GGG2020 $X_{CO}$ has no significant bias with respect to AirCore CO measurements. Figure 16e shows significant variation in the TCCON/in situ CO agreement, with individual points also having large uncertainty. This resulting $2\sigma$ uncertainty in the mean ratio is significantly larger than for the other gases, at 0.0526. Thus, the mean TCCON/in situ CO ratio is well within its $2\sigma$ uncertainty of 1. We do acknowledge that limiting the CO comparisons to AirCore profiles alone may contribute to a larger uncertainty than if aircraft campaigns were included, due to the use of a CO-spiked fill gas in AirCores (see §2.1 of Martínez-Alonso et al., 2022). However, comparing TCCON $X_{CO}$ to AirCore profiles was significantly more straightforward than including aircraft profiles, since the already-matched AirCore profiles for $CO_2$ and $CH_4$ intrinsically include CO as well. Given the other reasons discussed above for not applying an in situ-derived scaling to GGG2020 $X_{CO}$ and the process needed to match aircraft data with TCCON (see Appendix C1.1), we chose to accept the additional uncertainty from using AirCore profiles only. Future versions of the TCCON data product will reevaluate the inclusion of aircraft profiles alongside AirCore ones.

Figure 16 also provides insight into how instrumental errors affect different TCCON products. Under ideal circumstances, $X_{luft}$ (the quantity on the $x$-axis) should be 1; in practice, the nominal value for the TCCON network is 0.999, due to small residual biases in the $O_2$ spectroscopy. Deviations of $X_{luft}$ from the nominal value indicate either (a) variable errors in spectroscopy, such as temperature or pressure broadening, or (b) instrument issues, such as a misalignment in the beam path. From Fig. 16a, we can see that the TCCON/in situ ratio tends to be less when $X_{luft} < 0.999$ and greater when $X_{luft} > 0.999$. The slope for Fig. 16a is 0.363. This translates to a bias in $CO_2$ of about 0.15%, or approximately 0.5 ppm, when $X_{luft}$ is 0.004 units away from the nominal value of 0.999 ($0.15\% = 0.0015 = 0.363 \times 0.004$). To keep this bias well below the expected 0.25% accuracy, we limit the comparison used here to those where $X_{luft}$ is between 0.996 and 1.002 and have instituted additional quality checks of TCCON data that filter out observations when $X_{luft}$ is outside the range of 0.995 to 1.003 for an extended period of time. Additionally, $X_{luft}$ is now reported in the public data set alongside other $X_{gas}$ retrievals.

We note that the standard $CO_2$ and the near surface-sensitive $lCO_2$ products show the clearest dependence on $X_{luft}$. The reason for this is not clear at this time, though it implies a stronger dependence of these products on ILS compared to the other four products discussed in this section. Future versions of GGG are planned to account for errors in the ILS, which we hope will mitigate this bias and improve the accuracy of $CO_2$ data when $X_{luft}$ is outside the 0.995 to 1.003 range.

The correlation of $X_{CO_2}$ and $X_{lCO_2}$ with $X_{luft}$ implies that we could develop an $X_{luft}$-based bias correction for those $CO_2$ products. Such a correction is planned for a minor update to the GGG data product. Our aim is to quantify the underlying physical drivers of the $X_{CO_2}$ bias and use the correlation of those factors with $X_{luft}$ to derive the bias correction. This would allow us to use the comparison to in situ data shown here as an independent verification of the bias corrections efficacy.

### 8.3.2   Addressing the $CO_2$ scale change from X2007 to X2019 and changing $O_2$ dry mole fraction

The update from the previous WMO $CO_2$ X2007 calibration scale to the new X2019 calibration scale (Hall et al., 2021) occurred late enough in the process of releasing GGG2020 that we were not able to incorporate it into the initial release. Given the clear need expressed by the community to have TCCON data tied to the same scale as in situ data, we have since derived

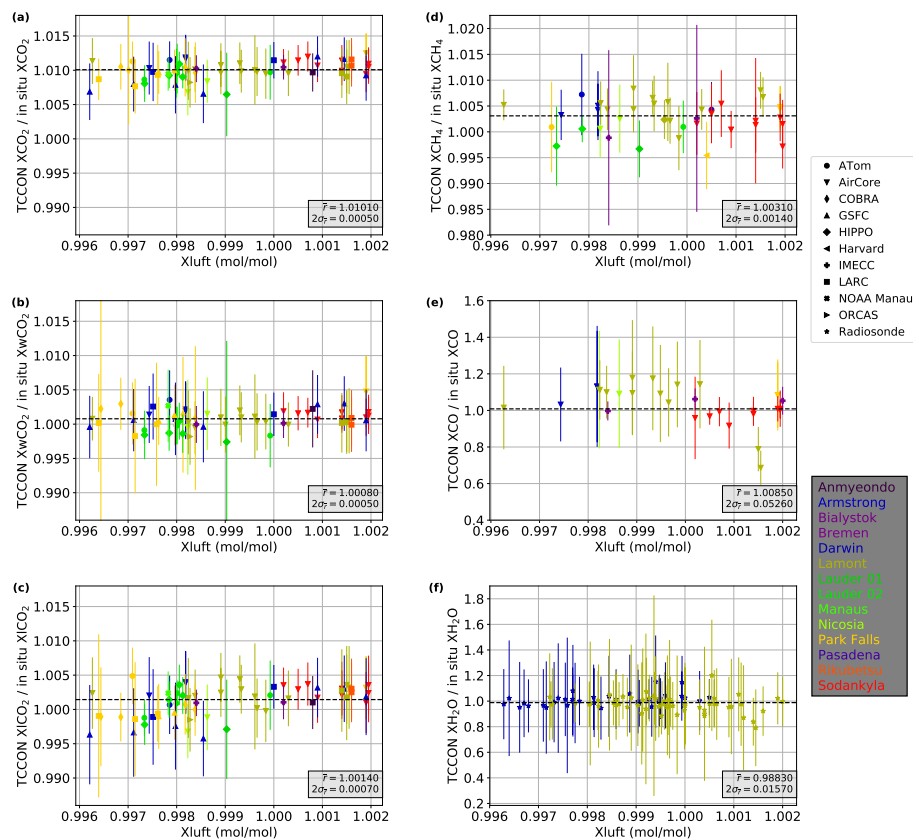

**Figure 16.** Plots of the TCCON/in situ $X_{gas}$ ratios for **(a)** $CO_2$, **(b)** $wCO_2$, **(c)** $lCO_2$, **(d)** $CH_4$, **(e)** CO, and **(f)** $H_2O$. In all plots, the $y$-axis is the ratio of TCCON/in situ $X_{gas}$ and the $x$-axis is the median $X_{luft}$ value for the TCCON observations in a comparison (see text for explanation of $X_{luft}$). The marker style of each comparison indicates the source of the in situ data and the color indicates which TCCON site the comparison occurred at. The text inset in the lower-right corner of each plot gives the uncertainty-weighted mean TCCON/in situ ratio and its $2\sigma$ uncertainty. The dashed black lines mark the mean ratio. Panels a, b, and c are set to use the same $y$-limits; some of the error bars in (b) go outside the $y$-limits.

new in situ correction factors to tie all three TCCON $X_{CO_2}$ products to the X2019 scale. Doing so required obtaining in situ data that had been adjusted to the new scale, which we did in one of three ways:

1. The preferred approach was for the data originator to fully recalibrate their data to the new scale using the updated standards provided by the NOAA Global Monitoring Laboratory. NOAA AirCore and some NOAA ObsPack data (GLOBALVIEWplus v7, Schuldt et al., 2021) followed this approach.[4]

2. The second approach was for the data originator or an intermediate provider to adjust the $CO_2$ data using the linear correction described in §9.1 of Hall et al. (2021). The remaining NOAA ObsPack data not covered by approach #1 followed this approach.

3. The third approach was for us to perform the same linear correction as #2 ourselves. All other data used this approach.

Also recall that the profiles must be extended to 70 km altitude using the TCCON standard priors to ensure that the same vertical extent is captured in the in situ and TCCON column averages. As discussed in Laughner et al. (2023), the standard priors are derived from NOAA data at the Mauna Loa and American Samoa observatories, and so are also intrinsically tied to WMO calibration scales. To ensure consistency throughout the in situ profiles, we used the latest available monthly average $CO_2$ flask data on the X2019 scale as input to the priors when generating the profile extensions. Once this was complete, we redid the analysis described in §8.3 with the in situ profiles adjusted to the X2019 scale to generate updated correction factors.

The overall effect of the scale change for each of the three TCCON $CO_2$ products is shown in Fig. 17 compared to the "raw," un-bias corrected $X_{CO_2}$ value on the $x$-axis. The magnitude is about $+0.15$ ppm for typical current $X_{CO_2}$ values of 400 ppm. In the TCCON data products, there are three $CO_2$ variables with the suffix `_x2019` which are adjusted to the new X2019 scale.

Another source of bias that is of similar magnitude to the effect of the scale change is the assumed $O_2$ dry mole fraction. As shown in Eq. (1), the column-average dry mole fractions reported by TCCON are computed by dividing the column density of the target gas by the $O_2$ column density, and scaling by the mean $O_2$ dry mole fraction in the atmosphere. We have assumed that this dry mole fraction is fixed for the initial GGG2020 data products; however, it is in fact changing over time due to various processes, predominantly fossil fuel combustion and the land biosphere (Keeling et al., 1998; Keeling and Manning, 2014).

Because the effect of ignoring the change in the global average $O_2$ dry mole fraction is of similar magnitude to the X2007 to X2019 scale change, we decided to account for the change in $O_2$ dry mole fraction over time in the $CO_2$ products updated to the X2019 scale. We did *not* retroactively apply this correction to the X2007 $X_{CO_2}$ or the other $X_{gas}$ products, as doing so would change the $X_{gas}$ values and require a new data version. This correction will be applied to all $X_{gas}$ values in the next GGG data version.

Our approach to account for changing $O_2$ dry mole fraction takes advantage of the anticorrelation between atmospheric $O_2$ and $CO_2$ to derive the $O_2$ dry mole fraction from $CO_2$ measured by TCCON. (For our application, this assumption is

---

[4]The ObsPack release notes at https://gml.noaa.gov/ccgg/obspack/release_notes.html#obspack_co2_1_GLOBALVIEWplus_v7.0_2021-08-18 provide information on how to determine which data was fully recalibrated.

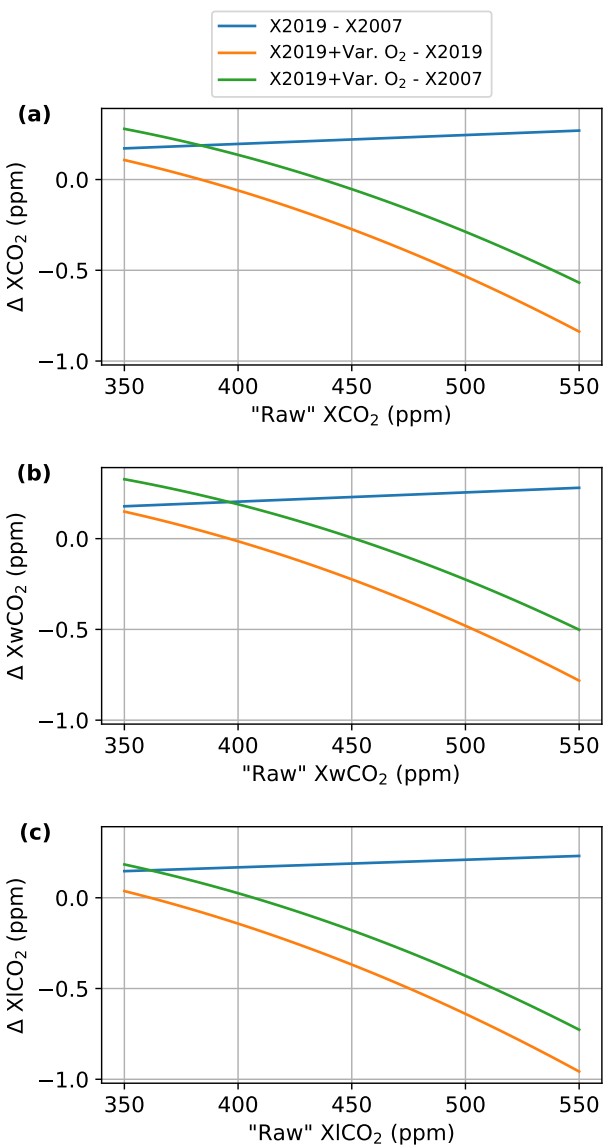

**Figure 17.** The change in TCCON **(a)** $X_{CO_2}$, **(b)** $X_{wCO_2}$, and **(c)** $X_{lCO_2}$ due to the WMO scale change, change in assumed $O_2$ dry mole fraction, and the combination of both. The $x$-axis is the "raw" $X_{CO_2}$ value that has no in situ bias correction and assumes a fixed $O_2$ dry mole fraction. The "X2019 – X2007" line shows the difference due to only the $CO_2$ WMO scale change, the "X2019+Var $O_2$ – X2019" shows the difference due to only the change from fixed to variable $O_2$ dry mole fraction, and the "X2019+Var. $O_2$ – X2007" line shows the total change from both effects combined.

sufficiently accurate; however, we note that this is not generally true for other applications of $O_2/N_2$ ratio data.) Specifically, the value for $f_{O_2}$ in Eq. (1) is calculated as (see Appendix E1 for the full derivation):

$$f_{O_2} = (\alpha - \alpha \cdot f_{O_2,\text{ref}} - f_{O_2,\text{ref}}) \cdot \frac{X_{CO_2} - X_{CO_2,\text{ref}}}{1 - X_{CO_2} - \alpha \cdot X_{CO_2}} + f_{O_2,\text{ref}} \tag{12}$$

where:

- $\alpha = \partial N_{O_2}/\partial N_{CO_2} = -1/0.4575$, i.e. the change in the number of moles of $O_2$ in the atmosphere for a given change in the number of moles of $CO_2$ in the atmosphere. The choice of $-1/0.4575$ comes from the agreement with the measured change in $f_{O_2}$ as shown in Fig. 18. This value is chosen to remove the effect of long term trends in the $O_2$ dry mole fraction, and ignores synoptic-scale variations due to e.g. photosynthesis or fossil fuel emissions.

- $f_{O_2,\text{ref}}$ is the reference value for the dry mole fraction of $O_2$. We use 0.209341 based on the value measured by Aoki et al. (2019) at Hateruma Island, Japan in 2015 and adjusting by $\sim 2$ ppm to approximate the global mean $f_{O_2}$ by using the difference between the annual mean $CO_2$ reported for Hateruma Island by Aoki et al. (2019) and that for the NOAA global marine boundary layer reference. (A revised calculation accounting for possible influence of fossil fuel emissions on Hateruma Island puts the global mean $O_2$ dry mole fraction closer to 0.209347, however the 0.209341 value is what is used in GGG2020.)

- $X_{CO_2,\text{ref}}$ is a reference value for the column-average dry mole fraction of $CO_2$. We use $4 \times 10^{-4}$ (400 ppm) to approximate the value seen in TCCON data during 2015 (the same year as the $f_{O_2,\text{ref}}$ value), though as discussed below, it is not crucial that the $O_2$ and $CO_2$ reference values be for exactly the same time.

- $X_{CO_2}$ is the "raw" measured TCCON $X_{CO_2}$ with airmass correction and assuming $f_{O_2} = f_{O_2,\text{ref}} = 0.209341$.

To validate this approach, we also compute the change in $f_{O_2}$ (including the effect of $CO_2$ dilution) using $\delta(O_2/N_2)$ data measured by the Scripps Institution of Oceanography at Alert, NWT, Canada (station code ALT); La Jolla Pier, California, USA (LJO); and Cape Grim, Australia (CGO) and NOAA $CO_2$ annual trend data (Lan et al., 2023). To approximate a global mean $\delta(O_2/N_2)$ value, we follow §5.15.4.2 of Keeling and Manning (2014) and combine the data from these stations as (ALT + LJO)/4 + CGO/2.

The results of this comparison are shown in Fig. 18. The black line shows the change in $f_{O_2}$ computed using the Scripps $\delta(O_2/N_2)$ data (see Appendix E2 for the methodology), while the other three lines represent $f_{O_2}$ calculated with Eq. (12) and various values of $\alpha$. We can see that Eq. (12) with $\alpha = -1/0.4575$ gives quite good agreement with the change in $f_{O_2}$ computed using the Scripps $\delta(O_2/N_2)$ and NOAA global $CO_2$ data.

The final step in adopting the variable $O_2$ dry mole fraction was to recompute the in situ correction factor once more, using the variable $O_2$ dry mole fraction in the TCCON $X_{\text{gas}}$ values for the comparison. Doing so ensures that any constant multiplicative bias introduced by incorrect or inconsistent values for the $f_{O_2,\text{ref}}$ or $X_{CO_2,\text{ref}}$ values is scaled out. This is why, in

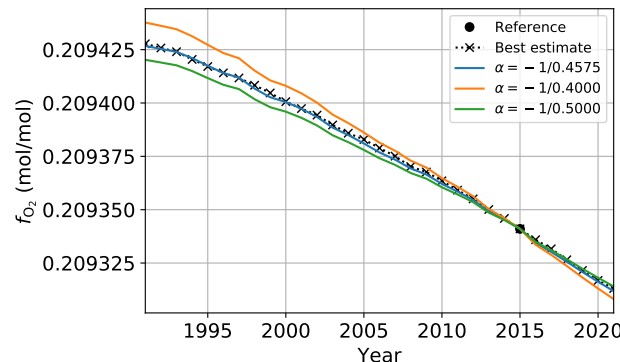

**Figure 18.** Comparison of $f_{O_2}$ values calculated using Eq. (12) for three different values of $\alpha$ versus a best estimate of $f_{O_2}$ using $\delta(O_2/N_2)$ from the Scripps Intitute of Oceanography (Scripps $O_2$ Program, 2022) and NOAA global mean $CO_2$ (Lan et al., 2023) data. The three colored lines also use NOAA global mean $CO_2$ data for the $X_{CO_2}$ and $X_{CO_2,\text{ref}}$ values in Eq. (12). The black circle marks our reference value of $f_{O_2} = 0.209341$.

the discussion above about the choice of those reference values, we note that it is not critical to have the $O_2$ and $CO_2$ reference value exactly consistent.

The orange lines in Fig. 17 show the effects of the change from a fixed $O_2$ dry mole fraction to the variable one. For $X_{CO_2}$ values around 400 ppm, the change is of similar magnitude to the WMO scale change for $CO_2$ products. If $CO_2$ mixing ratios continue to increase in the future, the difference between using the incorrectly fixed and correctly varying $O_2$ dry mole fraction would increase to 0.75 to 1 ppm in magnitude.

    The green lines in Fig. 17 show the combined effect of the $CO_2$ calibration scale change and the switch to a variable $O_2$ dry mole fraction. For low "raw" $X_{CO_2}$ values (i.e. values without the in situ bias correction and using a fixed $O_2$ dry mole fraction) the two effects reinforce each other, but as the raw $X_{CO_2}$ increases, the $O_2$ dry mole fraction change starts to counteract part of the $CO_2$ scale change.

    $X_{CO_2}$, $X_{wCO_2}$, and $X_{lCO_2}$ on the X2019 scale and accounting for the variable $O_2$ dry mole fraction are now available in the public data set as variables `xco2_x2019`, `xwco2_experimental_x2019`, and `xlco2_experimental_x2019`.

Users comparing to other data or model simulations/assimilations on the X2019 scale should use these variables. Anyone needing to compare against data still on the X2007 scale can use `xco2`, `xwco2_experimental`, and `xlco2_experimental` instead.

### 8.3.3 $N_2O$ in situ comparisons

To derive an in situ correction for $N_2O$, we adopted a different approach than the other gases due to the small number of

$N_2O$ profiles over TCCON sites which our matching algorithm found in the NOAA CCGG Aircraft Program v1.0 ObsPack (Sweeney et al., 2018). Figure 19a shows the 10 profiles identified from the ObsPack, and Fig. 19b shows the TCCON/in situ

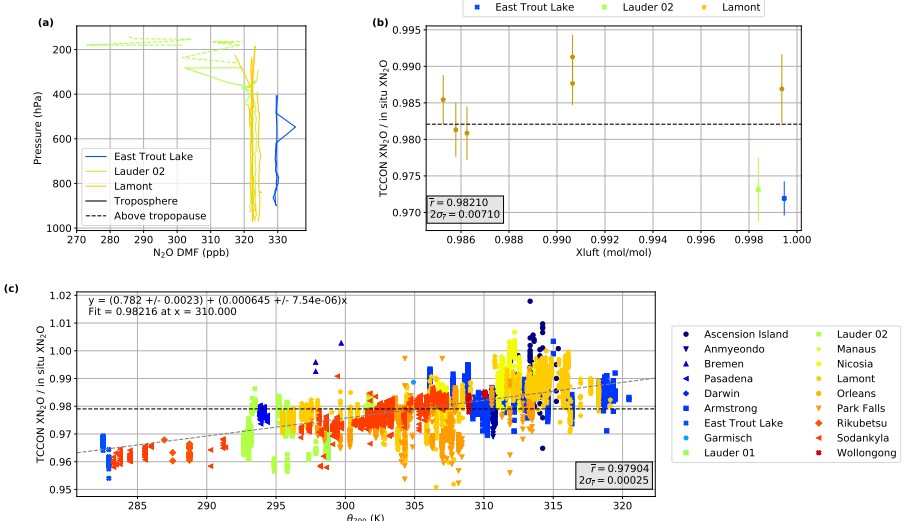

**Figure 19. (a)** The available $N_2O$ profiles over TCCON sites from the NOAA CCGG Aircraft Program v1.0 ObsPack (Sweeney et al., 2018). **(b)** TCCON/in situ ratio vs. $X_{luft}$ similar to Fig. 16, but for $N_2O$. **(c)** The TCCON/in situ $X_{N_2O}$ ratio derived using surface NOAA $N_2O$ data versus mid-tropospheric potential temperature. The dashed gray line is a robust fit to the data. The text in the lower right hand corner gives the mean TCCON/in situ ratio (denoted also by the horizontal dashed black line) and its $2\sigma$ standard deviation. The points are colored by TCCON site.

ratio vs. $X_{luft}$ relationship for these profiles. We note that this scarcity of profiles was partly due to the criteria used to filter for good quality profiles (Appendix C1.1). However, given how well-mixed $N_2O$ is in the troposphere, the criteria intended to ensure a profile had enough vertical resolution to capture plumes of $CO_2$ or $CH_4$ could be relaxed for $N_2O$ in future TCCON/in
situ comparisons to increase the number of available $N_2O$ profiles for comparison.

The available profiles were further restricted by our criteria for coincidence with good quality TCCON observations. 2 of these 10 profiles do not meet the coincidence criteria for inclusion in Fig. 19b, and 5 of the remaining 8 fall outside the allowed $X_{luft}$ range of 0.996 to 1.002. With the available data, it is difficult to distinguish whether there is significant correlation between $X_{luft}$ and TCCON $X_{N_2O}$ bias, and therefore whether those 5 comparisons below $X_{luft} = 0.996$ should be excluded.
As their exclusion would significantly alter the in situ correction for $X_{N_2O}$, we tested a second approach to derive the $N_2O$ correction.

This alternate approach uses NOAA surface $N_2O$ data from the NOAA Halocarbons and other Atmospheric Trace Species (HATS) program (Dutton et al., 2023) combined with the GGG2020 priors to generate pseudo-in situ profiles. This takes advantage of the limited vertical variation in $N_2O$ up to the tropopause seen in Fig. 19a and the good accuracy of the GGG2020
priors in the stratosphere (Laughner et al., 2023). These pseudo-in situ profiles use the HATS $N_2O$ data for the tropospheric VMRs, the GGG2020 priors for VMRs above 380 K potential temperature, and linearly interpolates in between. These pseudo-in situ profiles are then integrated following Eq. (10) to produce a pseudo-in situ $X_{N_2O}$ and compared to TCCON in the same

manner as the other gases. As we are not limited by when an aircraft provided an $N_2O$ profile over a TCCON site, we can compare to TCCON observations from any time. We use spectra from the same sites and days as the other gases, filtered for
the following criteria:

  - FVSI $\leq 0.05$, as for the other gases

  - $X_{\mathrm{luft}}$ between 0.996 and 1.002, as for the other gases

  - The difference between prior HF column density and retrieved HF column density is $< 2 \times 10^{14}$ molec. $\mathrm{cm}^{-2}$.

The filtering on HF column helps to remove cases where the stratosphere prior $N_2O$ used in the pseudo-in situ profiles is
incorrect. HF is a gas found almost exclusively in the stratosphere, and in GGG2020, the HF and $N_2O$ stratospheric priors are coupled. Thus, when the retrieved HF column is substantially different from the prior, that indicates that the HF prior was incorrect, which implies the same for the $N_2O$ profile. HF columns tend to be between 1 and $2 \times 10^{15}$ molec. $\mathrm{cm}^{-2}$, so $2 \times 10^{14}$ molec. $\mathrm{cm}^{-2}$ represents a 10% to 20% error in the HF prior. Given that the stratosphere component of $N_2O$ is $< 20\%$ of the column, and assuming that the percent error in the $N_2O$ prior is similar, this keeps the random error in the pseudo-in situ $X_{N_2O}$
to less that 2% to 4%. All together, these filtering criteria retain approximately 8600 TCCON observations from the initial set of $\sim 20,000$ observations used in the in situ correction analysis.

This larger sample set for $N_2O$ allowed us to identify a correlation in $X_{N_2O}$ bias with atmospheric temperature. Figure 19c shows how the TCCON/in situ $X_{N_2O}$ ratio varies with potential temperature at 700 hPa. As in the ADCF analysis (§8.1), these potential temperature values come from the GEOS FP-IT meteorology used as input to the GGG retrievals. The presence of
this bias suggests that there is an error in the temperature dependence of the $N_2O$ cross sections (similar to that we identified and removed for $O_2$, §6.3). In the near term, we plan to develop a post-processing correction for this temperature bias in $N_2O$ for inclusion in a minor update to the TCCON GGG2020 data within 2–3 years. Long term, the underlying error in the spectroscopic model will be corrected so that the next major TCCON data release will have improved $X_{N_2O}$ data.

For GGG2020, we elected to choose the $X_{N_2O}$ in situ correction as the value of the fit in Fig. 19c at 310 K potential
temperature. This is consistent with the choice of ADCF values at the same temperature (§8.1). The value of 0.9822 is very close to the mean TCCON/in situ ratio using the 8 true in situ profiles in Fig. 19b. That both methods agree gives us confidence that this is a reasonable value to use for the in situ correction. We are also investigating applying the slope from Fig. 19c to TCCON $X_{N_2O}$ as a temperature-based bias correction. Figure 20 demonstrates the difference this correction would make, both in comparison to the in situ data (Fig. 20a,b) and to the column-average dry mole fractions themsleves (Fig. 20c).

**8.3.4   In situ bias correction summary**

A summary of the in situ correction factors, their errors, and what in situ calibration scales each product is tied to are given in Table 4. Because these correction factors are the mean TCCON/in situ ratio, dividing the airmass-corrected and window-averaged values by these correction factors removes the mean TCCON-in situ bias.

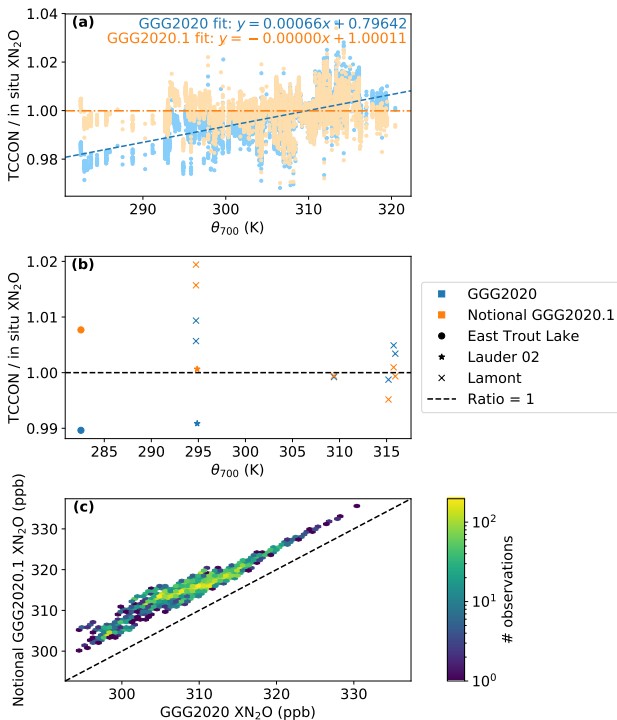

**Figure 20.** Future correction for $X_{N_2O}$. **(a)** Similar to Fig. 19c, except showing the ratio between TCCON and the surface-derived $X_{N_2O}$ from GGG2020 with the in situ correction factor of 0.9821 applied in blue, and the expected temperature-corrected $X_{N_2O}$ in orange, with their respective fits. **(b)** Similar to Fig. 19b, but like panel (a) of this figure, comparing the ratios of GGG2020 and temperature-corrected $X_{N_2O}$ to in situ. **(c)** A 2D histogram comparing the current and notional corrected $X_{N_2O}$

In the TCCON data, users will find two sets of $X_{CO_2}$ variables. Those with the `_x2019` suffix (`xwco2_experimental_x2019`, `xlco2_experimental_x2019`, and `xco2_x2019`) are those tied to the WMO X2019 $CO_2$ scale and which use the variable $O_2$ dry mole fraction. Those $CO_2$ variables without the `_x2019` suffix remain tied to the WMO X2007 $CO_2$ scale and still use the fixed $O_2$ dry mole fraction. All other gases (`xch4`, `xco`, etc.) also still use the fixed $O_2$ dry mole fraction.

Releasing the rescaled $X_{CO_2}$ as new variables, rather than creating a new TCCON data version with the existing variables rescaled, was chosen for several reasons. First, it is logistically simpler, allowing us to provide this update more quickly. Second, during this transitional period when existing $CO_2$ data is available on both the X2007 and X2019 scales, having both X2007 and X2019 $X_{CO_2}$ allows users to switch back and forth easily if they need to match up with other datasets on a mix of both scales. Third, this approach provides for release of more recent TCCON data without disrupting existing users workflows— users do not have to worry about existing variables changing, but can switch their analyses to use the updated $X_{CO_2}$ variables if and when they wish. Incorporating the variable $O_2$ dry mole fraction for all gases is planned for an upcoming minor revision of the TCCON data (tentatively called "GGG2020.1"). Likewise, a temperature-corrected $X_{N_2O}$ product will be included in GGG2020.1 or the follow on GGG2020.2, depending on development time.

| $X_{gas}$ product | Correction factor | CF error | Calibration scale | $N$ | $f_{O_2}$ |
|---|---|---|---|---|---|
| $X_{CO_2}$ | 1.0101 | 0.0005 | WMO X2007 | 67 | Fixed |
| $X_{CO_2}$_x2019 | 1.0090 | 0.0005 | WMO X2019 | 70 | Var. |
| $X_{wCO_2}$ | 1.0008 | 0.0005 | WMO X2007 | 67 | Fixed |
| $X_{wCO_2}$_x2019 | 0.9996 | 0.0005 | WMO X2019 | 69 | Var. |
| $X_{lCO_2}$ | 1.0014 | 0.0007 | WMO X2007 | 67 | Fixed |
| $X_{lCO_2}$_x2019 | 1.0006 | 0.0007 | WMO X2019 | 69 | Var. |
| $X_{CH_4}$ | 1.0031 | 0.0014 | WMO X2004 | 40 | Fixed |
| $X_{N_2O}$ | 0.9821 | 0.0098 | NOAA 2006A | N/A | Fixed |
| $X_{CO}$ | 1.000 | 0.0526 | N/A | 31 | Fixed |
| $X_{H_2O}$ | 0.9883 | 0.0157 | ARM Radiosondes | 94 | Fixed |

**Table 4.** In situ correction factors and their errors for each $X_{gas}$ product evaluated against in situ data. The "Calibration scale" column indicates which scale or source these data are tied to by the AICFs. The $N$ column indicates how many profiles are used to calculate the AICF for that gas. The $f_{O_2}$ column indicates what $O_2$ dry mole fraction was used in the column density to column-average dry mole fraction conversion: "Fixed" means $f_{O_2} = 0.2095$ in Eq. (1) and "Var." means that the variable dry mole fraction described in §8.3.2 was used.

## 9  Uncertainty budget

To calculate an uncertainty in the GGG2020 dataset, we selected three days from the East Trout Lake dataset spanning a range of atmospheric water vapor, surface temperature and solar zenith angle (Figure 21). Each known source of uncertainty is

modeled or perturbed by a realistic amount in the GFIT forward model (the quantitative amounts are described in the following paragraphs), and we compute the percent fractional difference in $X_{gas}$ between the perturbed and unperturbed value. The total uncertainty is computed as the sum in quadrature of the individual uncertainties. For each gas, we have plotted the contributions of each source as a function of solar zenith angle for the June 11, 2019 date in Figures 23–25. The same figures for cold, dry February 18 are in the Appendix in Figures B1–B3, and for warm, wet July 23 are in Figures B4–B6. The sum in quadrature

of all the sources of error for each gas are plotted for the three days together in Figures 26–28. Each source of uncertainty included in our error budget is described below. Table 5 towards the end of this section summarizes the error budget for the primary TCCON products.

**Field of view**

The field of view (FOV) is the maximum solid angle viewed by the detector element, and its value is set by the field stop

diameter inside the instrument. It is an important parameter in the GFIT forward model because it defines the extent of off-axis rays that pass through the interferometer, ultimately limiting the spectral resolution of a spectrum. The field stop diameter is set by a physical pinhole ranging from 0.5-1.3 mm drilled into a thin plate within the instrument, and its size can be in error by a few percent. Here, we increase FOV by 7% to reflect any uncertainty in the field stop diameter.

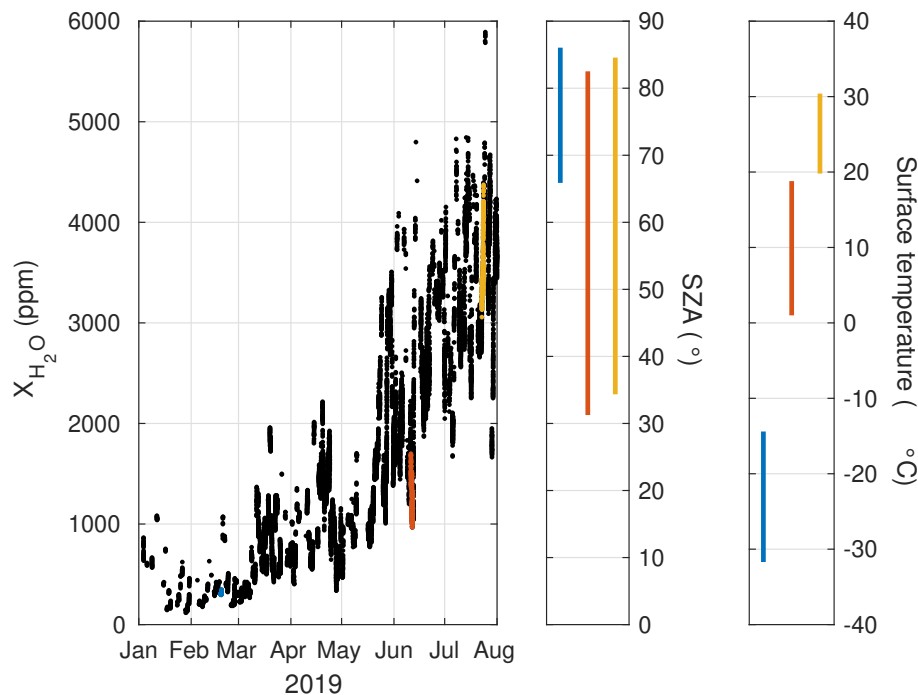

**Figure 21.** The three dates chosen for the error budget calculations are from East Trout Lake on February 18 (blue), June 11 (red), and July 23 (yellow), 2019. These dates were chosen to span a range of water vapor, solar zenith angle, and surface temperature. In the left panel, the black data points show the full East Trout Lake record between Jan and Aug 2019 for reference.

**Continuum basis functions**

In GGG2020, the number of continuum basis functions has been optimized to improve the spectral fits without over fitting the data (see §7). Here, we increase the number of continuum basis functions fitted by 1 in all windows that have widths $>5$ cm$^{-1}$ to assess the sensitivity of our choice of the number of basis functions to the retrieved $X_{\mathrm{gas}}$ value. The gases excluded from this test because of their fitting window widths are HF, HCl, and some $H_2O$ and HDO windows.

**Solar pointing**

The observer-sun Doppler stretch (OSDS) is a calculation made by GFIT based on the Earth-Sun radial velocity and the Earth's rotational velocity component, under the assumption that the solar tracker is imaging the centre of the Sun. It defines the Doppler stretch of the solar absorption lines relative to the telluric (atmospheric) absorption lines. If the solar tracker is not imaging the exact centre of the Sun, the solar lines may be Doppler-shifted relative to the telluric lines, creating systematic residuals in the spectral fits. Here we increase the OSDS by 2 ppm to assess the sensitivity of the retrievals to a small pointing

error from the Doppler stretch component alone. This error affects carbon monoxide more than the other gases because for every telluric CO line in the spectrum, there is also a solar CO absorption line beneath, making it difficult to distinguish solar from telluric CO absorption. In GGG2014 and previous versions, this was a particular problem, because the pointing was assumed to be in the centre of the solar disk. In GGG2020, however, the solar-gas stretches are now fitted, reducing the impact

of an OSDS error on the CO retrievals (see Wunch et al., 2015, Fig. 13). (A solar-gas stretch is when the solar absorption lines'
frequencies have to be stretched due to unaccounted Doppler effects, e.g., pointing away from the center of the solar disk.)

Solar tracker pointing offsets also affect the ray tracing in GFIT, causing errors in the airmasses calculated for a given spectrum. This error impacts all gas retrievals, but should mostly cancel in the ratio between the gas of interest and oxygen. It shows up most prominently in $X_{\text{luft}}$, as that is not a ratio between two retrieve gas columns (see §6.3 and Eq. 4). Here, we add a 0.05 degree pointing offset (poff), which represents a pointing error of about 20% of the solar radius.

**Prior**

We modify the priors in several ways to estimate the uncertainties caused by various errors in the a priori profiles.

– **A priori pressure profile** (prior pressure). We multiplied the pressure at each atmospheric level in the prior by 1.002 to scale up the pressure by 0.2% at all altitudes. For the HCl cell pressure error, we added 0.14 hPa (0.138 atm) to the cell pressure, following the "pessimistic" uncertainty budget in Hase et al. (2013, P3565). (The purpose of the HCl cells will be described in §9.10.)

– **A priori temperature profile** (prior temperature). We added 1 K to each atmospheric level in the prior.

– **A priori profile shape** (prior shift). We shifted the a priori profiles down by one atmospheric level. In GGG2014, we shifted the priors down by 1 km, so this is a slightly different approach, but the level spacing is about 1 km in altitude near the tropopause, where this shift is most important for well-mixed tropospheric gases like $N_2O$ and $CH_4$, and HF, a stratospheric gas. $H_2O$ and HDO are not shifted as part of this process, but are modified in an independent test.

– **A priori boundary layer CO** (prior CO enhanced). The GEOS FP-IT CO profiles are created using an old emission inventory, and tend to significantly overestimate emissions in urban regions that have reduced their emissions over time (e.g., Los Angeles). However, because of the coarse spatial resolution of GEOS FP-IT, sites that are located near to an urban centre can be affected by the urban enhancements in the model. We therefore add an additional test that affects only the CO error budget, in which we add 25 ppb to the altitudes below 2 km to estimate the uncertainty caused by the incorrect lower atmosphere shape in the GGG2020 CO prior profiles.

– **A priori $H_2O$ and HDO** (prior h2o/hdo). We modified the water and HDO profiles by reducing the values in the first 1 km by 50%.

**Surface pressure**

The surface pressure measurements we collect as part of our on-site meteorological data are important for determining the bottom altitude when integrating the total columns. The largest surface pressure uncertainty permitted by the TCCON data protocol is 0.3 hPa, but we have seen these instruments drift by up to 1 hPa. Here, we add 1 hPa to the surface pressure (pout) to calculate the sensitivity of the retrievals to this error.

**Nonlinearity**

Detector nonlinearities, described in §4.1, cause a discrepancy between the low-resolution spectral envelope and the high

resolution spectral lines, resulting in an offset at zero in the spectrum. These zero level offsets are most readily observed in regions of the spectrum where strong absorption lines absorb all the incident light (Abrams et al., 1994). Here, we add 0.001 (0.1%) to the zero level offset (ZLO) parameter in GFIT, a large ZLO observed in the network.

**Instrument line shape**

The instrument line shape (ILS) of a Fourier transform spectrometer quantifies the optical alignment of the instrument, and is independent of the alignment of the solar image. The ILS is characterized by two parameters: the modulation efficiency and phase error. The modulation efficiency is the broadening or narrowing of the ideal spectral line width in the instrument, and the phase error is the asymmetrical component of the spectral line that is caused by the misalignment. It is not currently possible to model phase error within GFIT, but we can model imperfect modulation efficiency. The TCCON data protocol 945 requires that the instrument modulation efficiencies must be within 5% of a perfect alignment. The modulation efficiency of a perfectly aligned interferometer is defined as a value of 1.0 at all optical path differences, taking self-apodization into account, and therefore the maximum and minimum modulation efficiency acceptable in the network is 1.05 and 0.95, respectively. Here we model two cases: a "shear" misalignment, where the modulation efficiency of the spectrometer increases linearly to 1.05 as a function of optical path difference, and an "angular" misalignment, where the modulation efficiency drops linearly to 0.95 as 950 a function of optical path difference. (See section 8 of Wunch et al. (2015) for more details on the mathematical forms for these misalignments.) We confirmed the misalignment by passing synthetic spectra generated by GFIT with these misalignments through LINEFIT (v14.8 Hase et al., 1999), a program designed to assess instrument line shapes (see Figure 22).

Because GGG2020 cannot model phase errors, these sensitivity studies are likely to underestimate the full effect of ILS errors, and therefore we include both the "shear" and "angular" misalignment in the sum.

**Other sources of error**

This error budget does not include radiometric noise or spectroscopic errors. We omit radiometric noise because Wunch et al. (2011) showed that random noise does not introduce a bias in $X_{CO_2}$ because TCCON spectra have a high signal-to-noise ratio due to the direct-sun viewing geometry and the strength of our target gases' absorption lines. We omit spectroscopic errors in this section because mean and SZA-dependent spectroscopic errors are removed by the post processing corrections (§8.1, 960 §8.3).

### 9.1 General comments on the results

This error budget was calculated by perturbing the retrieval as described above for data from a single site (East Trout Lake). Its purpose is to evaluate sources of bias that can affect an individual instrument, rather than provide an assessment of the actual magnitude of site-to-site biases. That magnitude has been assessed in sections §8.1 to §8.3. We generally expect the sensitivities 965 identified here to hold for all TCCON sites; however, we acknowledge that these results were derived from a single site.

The method of simulating modulation efficiency errors in GGG2014 (Wunch et al., 2015) was incorrect, resulting in an inferred uncertainty from ILS errors that is too large, likely by about a factor of 2 (see Appendix B1 for details). The change from the errant ILS modeling to our current model, on its own, will produce an apparent overall uncertainty reduction for GGG2020 when compared with GGG2014, but there have been no improvements in GGG2020 with respect to fitting imperfect

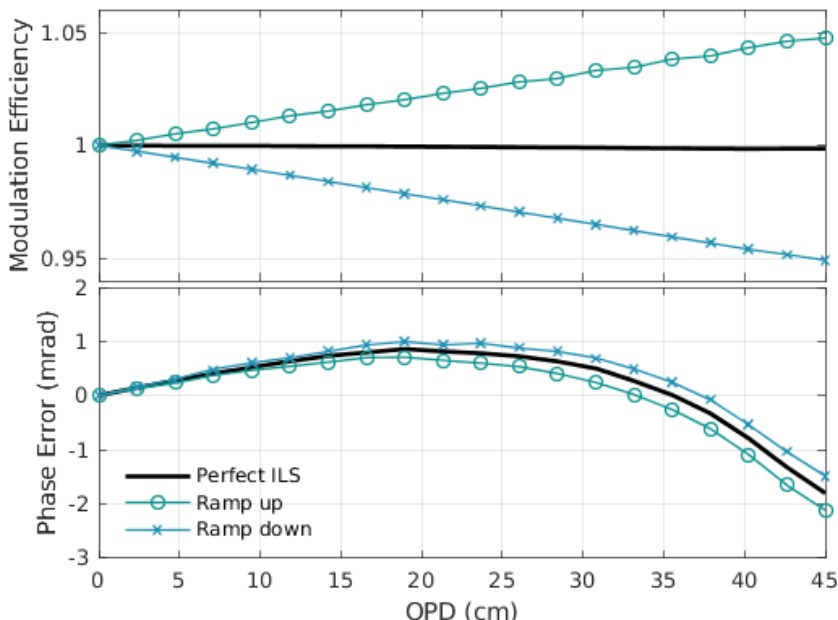

**Figure 22.** Synthetic spectra were generated using GFIT to simulate shear and angular misalignment with 5% change from the ideal line shape at a maximum optical path difference of 45 cm. These spectra were then passed through LINEFIT 14.8 to confirm that the modulation efficiency and phase errors were as expected.

ILS. However, there are several other improvements in GGG2020 that have resulted in systematic reductions in the uncertainty, including higher order continuum fitting (§7), solar-gas stretch fitting (§9), and gas-specific spectroscopy (§6.1) and line shape fitting improvements (§6.2).

In GGG2014, our retrievals were performed on a 1 km grid, and we shifted the profiles down by 1 level (or 1 km at all altitudes). In GGG2020, our retrievals are on a grid that increases in spacing with altitude, and a shift down by 1 level is

975 roughly 1 km at the tropopause, but smaller below and larger above. This change is most likely to affect the retrievals of gases for which there is a rapid change in abundance near the tropopause and above: $N_2O$, $CH_4$, and HF. Therefore, our shift for the GGG2020 error budget represents a larger perturbation to the a priori shape for these gases, which will cause larger errors in retrievals. However, because HF is a species found primarily in the stratosphere, and $N_2O$ and $CH_4$ are species found primarily in the troposphere, retrievals of HF can be used to diagnose and reduce the impact of the profile shift errors on $X_{N_2O}$ and $X_{CH_4}$

(e.g., Washenfelder et al., 2003; Saad et al., 2014, 2016; Wang et al., 2014).

In each section below, we will discuss the results for each gas, keeping in mind the reductions in error from the ILS model, and the inflation of error from the prior shifts.

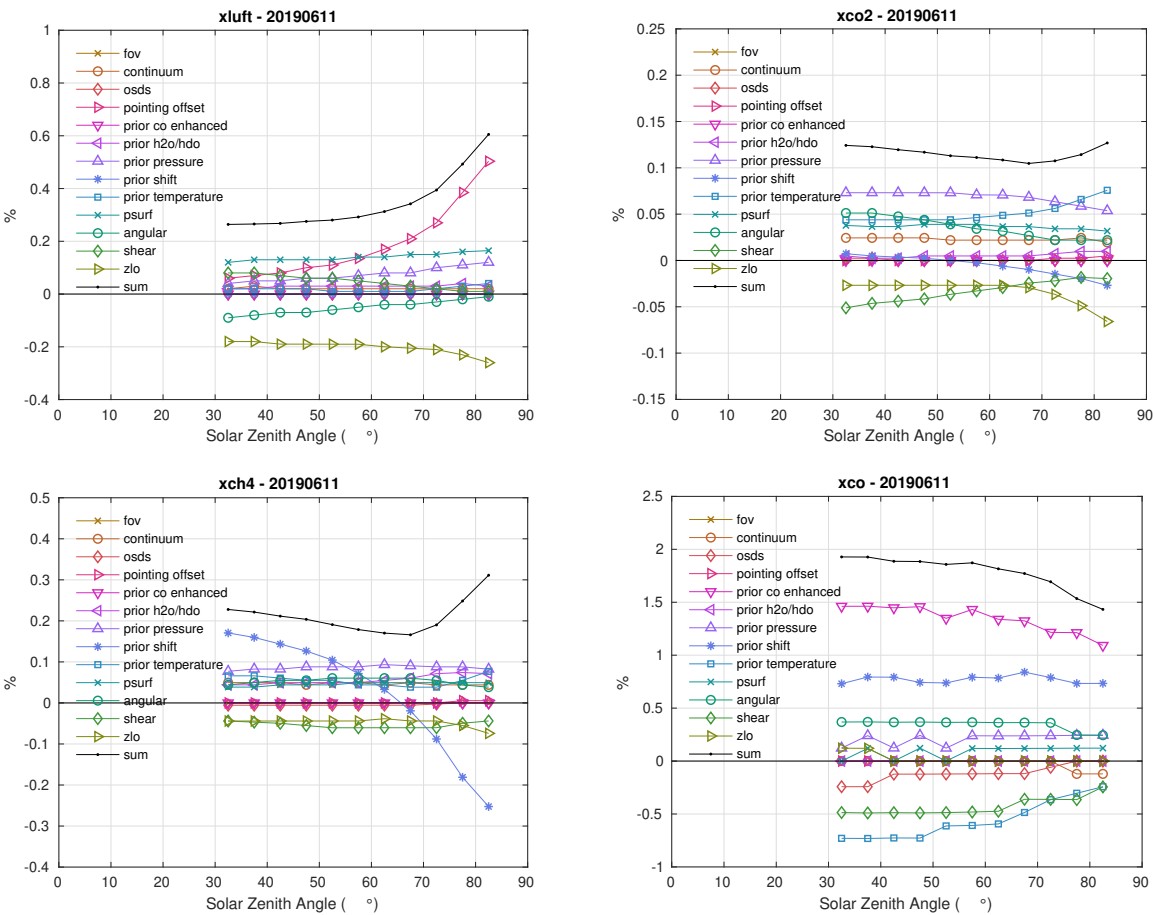

**Figure 23.** June 11, 2019 error budget from East Trout Lake. The figures show the percent difference between the perturbed test and the standard retrieval plotted as a function of solar zenith angle. "Sum" in the legend means the quadrature sum of the other terms. The retrievals plotted here are $X_{\mathrm{luft}}$, $X_{\mathrm{CO_2}}$, $X_{\mathrm{CH_4}}$, and $X_{\mathrm{CO}}$.

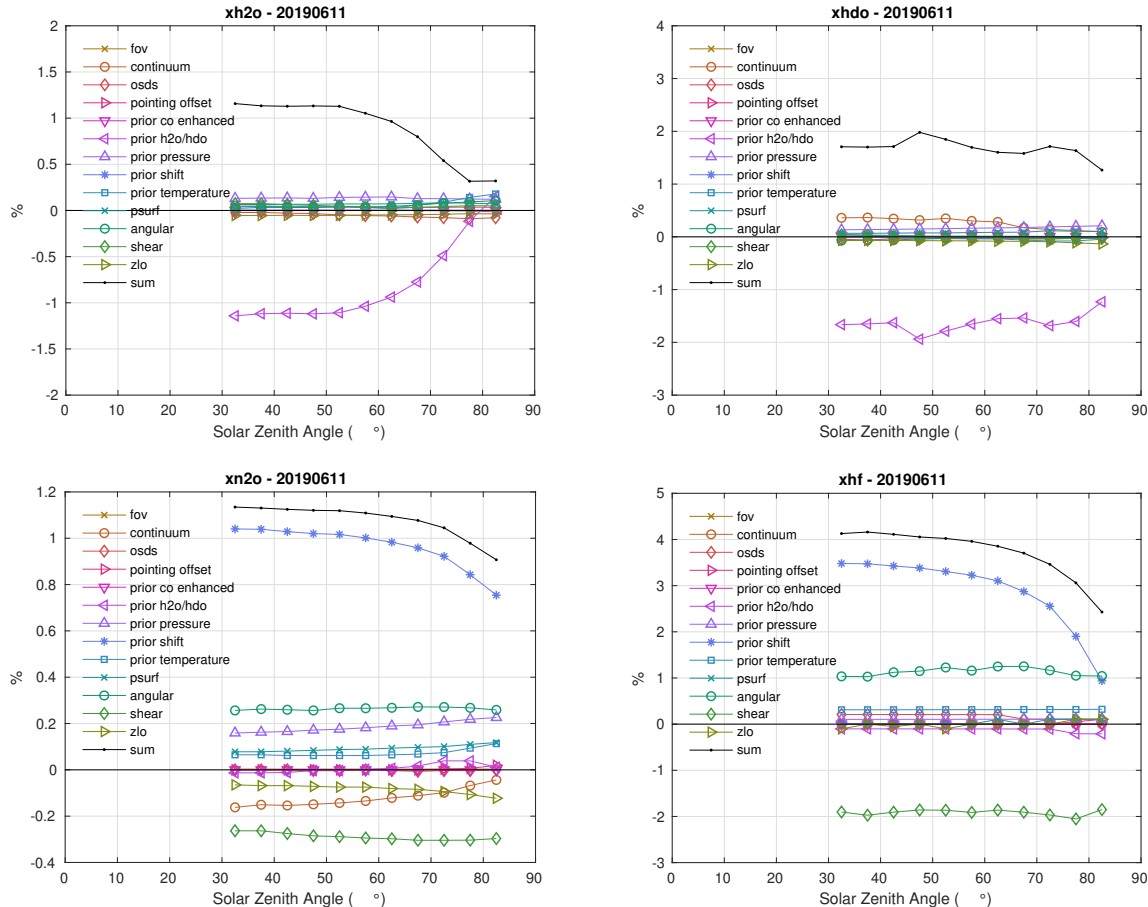

**Figure 24.** As in Figure 23, but for $X_{H_2O}$, $X_{HDO}$, $X_{N_2O}$, and $X_{HF}$.

## 9.2 $X_{luft}$

$X_{luft}$ is the column-averaged amount of dry air, and is equivalent to the parameter $X_{air}$ in GGG2014 (see Eq. 4 in §6.3 for
a definition). The error budget for $X_{luft}$ (Figures 23 and 26) is very similar to that of $X_{air}$ in GGG2014, with uncertainties smaller than 0.7% for all solar zenith angles less than 82°. The error is dominated by pointing offsets at large solar zenith angles, and zero level offsets contribute significantly to the error at all solar zenith angles.

## 9.3 $X_{CO_2}$

The $X_{CO_2}$ error budget is smaller than for GGG2014 (Wunch et al., 2015), mostly from the reduced continuum fitting errors.
The GGG2020 errors are below 0.16% (∼0.6 ppm) for solar zenith angles less than 82°, though if extrapolated linearly to

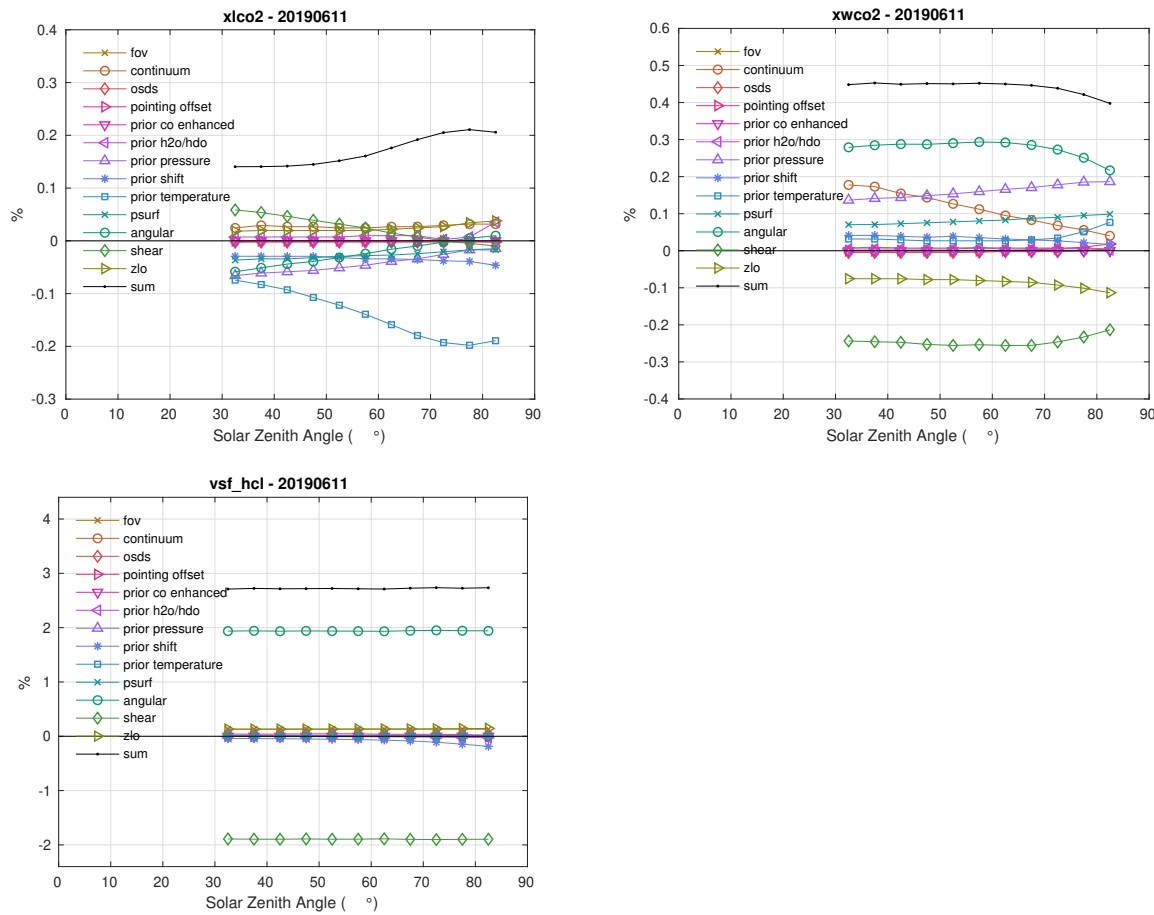

**Figure 25.** As in Figure 23, but for $X_{1CO_2}$, $X_{wCO_2}$, and HCl scale factors (vsf_hcl).

smaller solar zenith angles, the error could become larger than 0.15% at 0 degrees (Figures 23 and 26). The largest sources of error at lower solar zenith angles are from prior pressure offsets and misalignment. At larger solar zenith angles, the error becomes dominated by prior temperature errors and zero level offsets.

## 9.4 $X_{CH_4}$

The $X_{CH_4}$ error budget is smaller than for GGG2014 (Wunch et al., 2015). There is a significant reduction in the errors associated with observer-sun Doppler stretch (OSDS) offsets and continuum fitting errors. The GGG2020 errors are below 0.4% ($\sim$7 ppb) for solar zenith angles less than 82° (Figures 23 and 26). The largest sources of error at lower solar zenith angles are from prior profile shifts and prior pressure errors. At larger solar zenith angles, the error is dominated by prior

profile shifts. Errors caused by profile shifts can be mitigated by extracting the tropospheric partial column of $X_{CH_4}$ using the Saad et al. (2014) or Wang et al. (2014) methods.

### 9.5 $X_{CO}$

The $X_{CO}$ spectral fitting has been substantially improved in GGG2020, largely because of our reduced sensitivity to errors in the observer-sun Doppler stretch (OSDS), and also because we removed one of the fitted windows from our standard analysis in GGG2020 that had relatively poorer spectral fits. The GGG2020 errors are below 2% ($\sim 2$ ppb assuming a 100 ppb column) for all SZA $< 82°$. The largest sources of error are the prior CO enhancement, the prior shift, prior temperature, and shear misalignment (Figures 23 and 26).

### 9.6 $X_{H_2O}$ and $X_{HDO}$

The error budget for water and HDO is roughly the same as for GGG2014 and earlier, with total errors under 2% in $X_{H_2O}$ and 3% in $X_{HDO}$ over all solar zenith angles less than $82°$. The largest component of the error budget for water vapor and HDO is the shape of the a priori profile, which dominates the error budget for all solar zenith angles below $75°$ for water, and over all solar zenith angles below $82°$ for HDO (Figures 24 and 27).

### 9.7 $X_{N_2O}$

The $X_{N_2O}$ error budget is roughly the same as in GGG2014, with total errors less than 1.25% ($\sim 4$ ppb) over all solar zenith angles. The largest source of error is the prior shift, which is not surprising, given the rapid chemical destruction of $N_2O$ above the tropopause, though the magnitude of the error is about twice as large as it was for GGG2014. As discussed above, this is likely caused by differences in the way we shift the profile, and could be mitigated by extracting the tropospheric partial column by adapting the Saad et al. (2014) approach. Other contributors to the total error include the prior pressure, and shear and angular misalignments (Figures 24 and 27).

### 9.8 $X_{HF}$

HF has only a single absorption line (4038.96 cm$^{-1}$) that is located on the wing of a strong water absorption feature, so the retrievals tend to be noisy, especially at high solar zenith angles and under wet conditions. The $X_{HF}$ error budget has reduced in GGG2020 compared with GGG2014, with total errors now less than 5% over all solar zenith angles. In GGG2014, the errors were typically below 8%, but that error was dominated by the much larger shear misalignment. The largest source of error in GGG2020 is the prior shift, followed closely by shear misalignment (Figures 24 and 27).

### 9.9 $X_{lCO_2}$ and $X_{wCO_2}$

In GGG2014 and previous versions, we did not retrieve strong ("lCO$_2$") and weak ("wCO$_2$") CO$_2$ bands. The strong CO$_2$ retrieval errors are dominated by prior temperature errors, and the weak CO$_2$ errors are dominated by both shear and angular

misalignments, errors in the prior pressure, adjustments to the continuum curvature, and zero level offsets (Figures 25 and 28). The strong $lCO_2$ retrieval errors are less than 0.3% over all solar zenith angles, and the weak $wCO_2$ retrievals have around 0.5% errors at all solar zenith angles, declining slightly at higher angles.

## 9.10 VSF HCl

In this error budget, we have included the scale factors retrieved for HCl (vsf_hcl in Figs. 25 and 28). In the East Trout Lake instrument and most others in the network, a sealed HCl cell filled with a known quantity of gas (Hase et al., 2013) is placed permanently in the solar beam inside the evacuated spectrometer to monitor long-term changes in ILS or a leak of outside air into the cell. Because the quantity of gas in the cell is significantly larger than the atmospheric abundance, the atmospheric component is negligible and largely independent of surface pressure or other atmospheric adjustments. Therefore, deviations of the HCl scale factors from 1 indicates a drift in ILS. To assess the HCl retrieval sensitivity to changes in ILS and other parameters, we include the HCl scale factors in our error budget.

The retrieval errors in the scaling factors retrieved for HCl in a sealed cell are dominated by errors in the instrument line shape with no significant solar zenith angle dependence. This is a comforting result, showing that our HCl retrievals are a good diagnostic for instrument line shape drift. The HCl retrievals are not included in the standard public data files as they are used primarily for diagnostic purposes.

## 9.11 Uncertainty estimate comparison

For six products ($X_{CO_2}$, $X_{wCO_2}$, $X_{lCO_2}$, $X_{CH_4}$, $X_{CO}$, and $X_{H_2O}$) we can compare the uncertainty estimates derived from the error budgets with those computed from in situ comparisons similar to those in §8.3 but with one difference: the comparisons in §8.3 use the in situ vertical profiles as the prior trace gas profiles in the TCCON retrievals; the in situ comparisons in this section use standard TCCON GGG2020 prior profiles. For the in situ uncertainty, we use the mean absolute deviation (MAD) of the TCCON $X_{gas}$ values from the in situ $X_{gas}$ values after removing the mean bias for each $X_{gas}$ (i.e. the correction factor in Table 4). We use MAD over standard deviation because it is less sensitive to outliers. To convert the percent error from the error budget into a column-average dry mole fraction, we use the mean total percent error across all three days used in the error budget (18 Feb, 11 June, and 23 July 2019) binned by SZA in 5° increments. We interpolate this to the mean SZA of all spectra used in the in situ comparison for that gas and multiply this interpolated mean percentage by the mean TCCON $X_{gas}$ value across all the in situ comparisons. We then add the error from the in situ data to the error budget value in quadrature for comparison to the MAD. The results are presented in Table 5.

It is important to acknowledge that the MAD values calculated from the in situ comparison are (for most gases) may be less than the error budget, for several reasons. First, in situ profiles are usually taken when the target TCCON station is near optimal performance, so those comparisons are unlikely to capture the full range of error sources. Second, the in situ profiles are heavily concentrated over certain TCCON sites, also limiting how representative they are. Finally, the TCCON $X_{gas}$ values compared against the in situ values are averaged over a minimum of 2 hours. This will reduce sources of random error. However, we believe this is still a worthwhile evaluation of measurement accuracy because (a) there is real physical variation

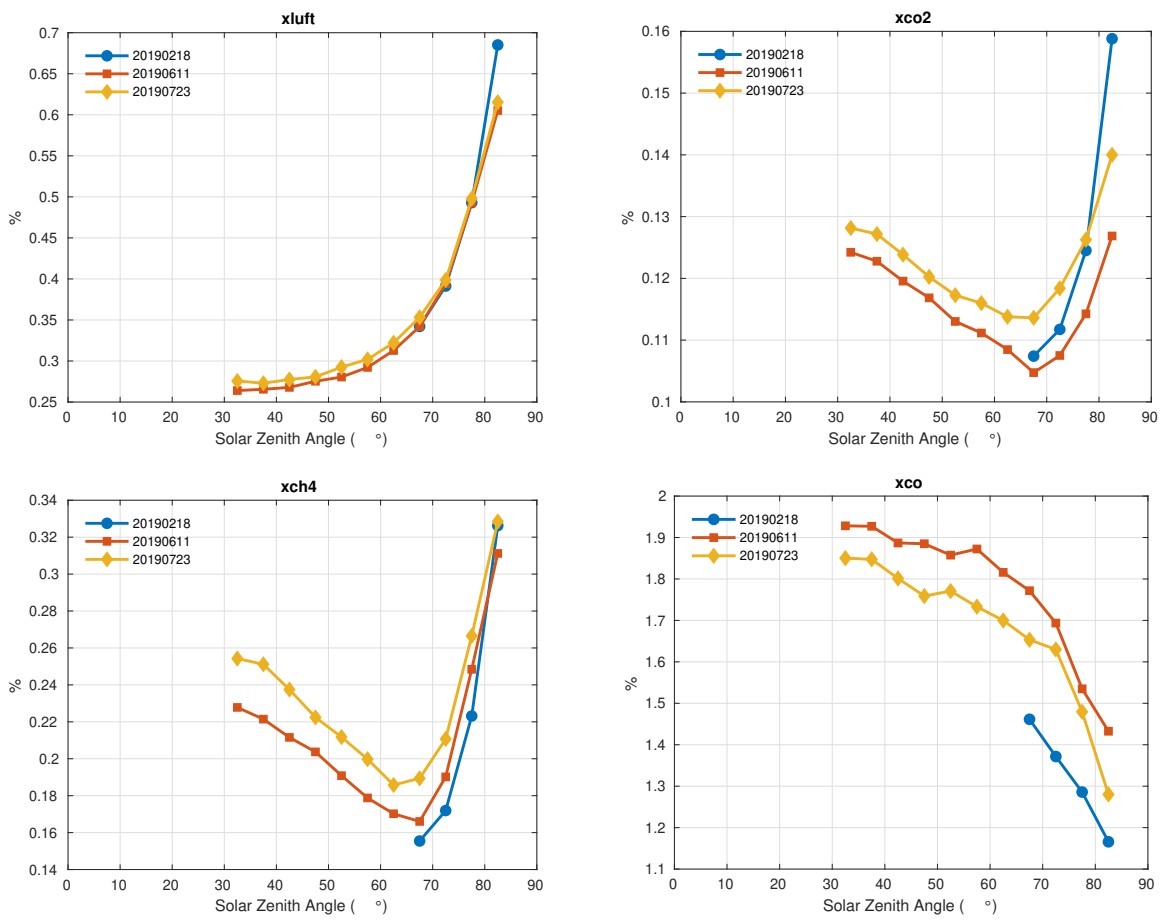

**Figure 26.** These figures show the sum in quadrature of all the errors plotted in Figure 23 for all three dates. The errors plotted here are for $X_{\mathrm{luft}}$, $X_{\mathrm{CO_2}}$, $X_{\mathrm{CH_4}}$, and $X_{\mathrm{CO}}$.

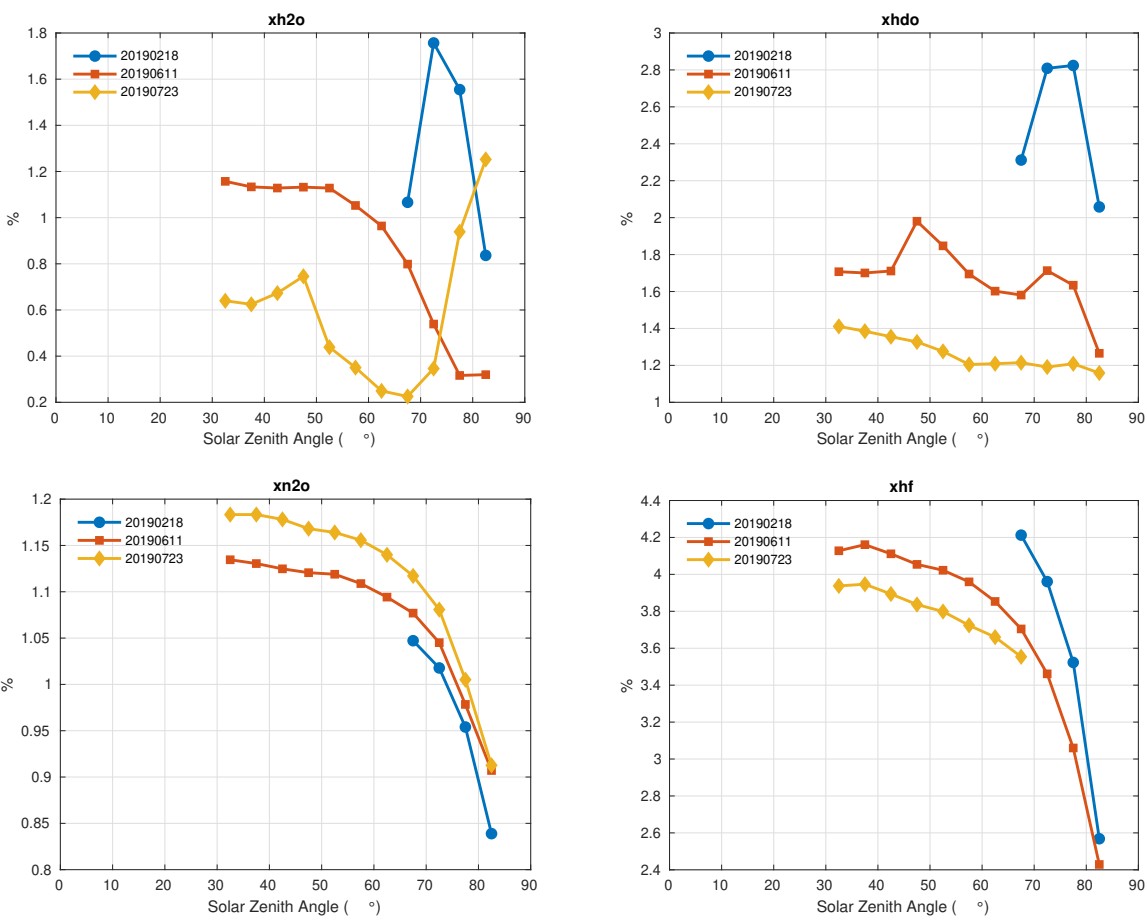

**Figure 27.** As in Figure 26, but for $X_{H_2O}$, $X_{HDO}$, $X_{N_2O}$, and $X_{HF}$. $X_{HF}$ values above 68° SZA are not available on 2019-07-23 because the HF lines were blacked out by $H_2O$ absorbance.

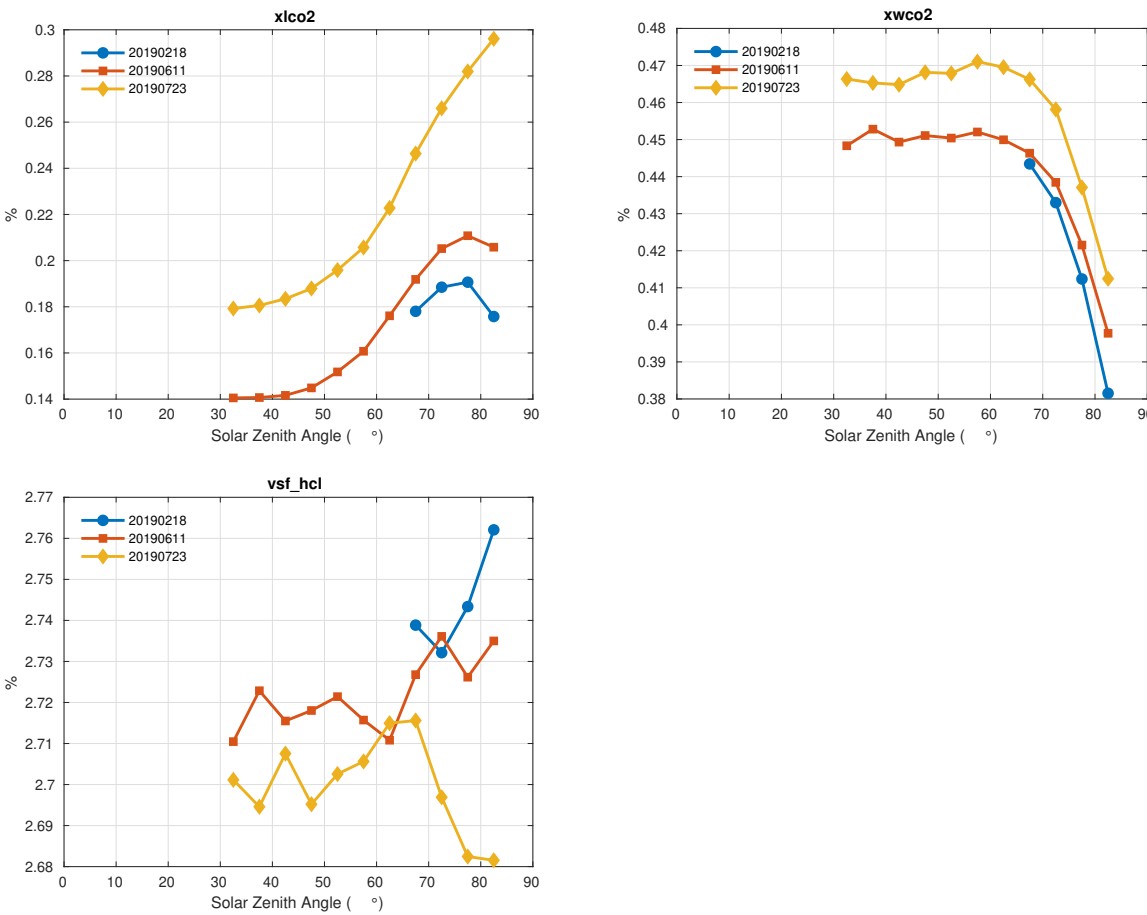

**Figure 28.** As in Figure 26, but for $X_{\mathrm{lCO_2}}$, $X_{\mathrm{wCO_2}}$, and HCl scale factors (vsf_hcl).

in the atmosphere during the in situ profile, and the time averaging is necessary to account for that and (b) many of the factors considered in the error budget will not average out over the coincidence window. For example, angular or shear misalignment of the instrument would be essentially constant over an entire day.

For three $X_{\mathrm{gas}}$ products ($X_{\mathrm{CO_2}}$, $X_{\mathrm{lCO_2}}$, and $X_{\mathrm{CH_4}}$) the MAD and error budget estimates are similar, which gives us con-
1065 fidence in the error budget estimates. For $X_{\mathrm{wCO_2}}$, the error budget estimate is much larger than the MAD value. It may be that the error budget tested larger errors in the stratosphere temperature or VMR prior profile than were observed during the in situ comparisons, as the $X_{\mathrm{wCO_2}}$ product is more sensitive to the upper atmosphere than the other $CO_2$ products in GGG2020. (Pressure errors could be another source of the overestimate, but the pressure perturbation test was designed to avoid introducing an overly large perturbation to the stratosphere.) As we treat the in situ-derived errors as a lower estimate, this situation is
1070 acceptable, but will be investigated in the future.

| Gas | SZA | MAD | MAD - in situ | Budget | $\epsilon_{\text{in situ}}$ |
|---|---|---|---|---|---|
| $X_{CO_2}$ (ppm) | 46° | 0.42 | 0.41 | 0.47 | 0.053 (0.30) |
| $X_{wCO_2}$ (ppm) | 46° | 0.43 | 0.42 | 1.8 | 0.062 (0.36) |
| $X_{lCO_2}$ (ppm) | 46° | 0.75 | 0.75 | 0.66 | 0.038 (0.24) |
| $X_{CH_4}$ (ppb) | 46° | 4.9 | 4.5 | 3.9 | 2.0 (9.6) |
| $X_{CO}$ (ppb) | 43° | 8.1 | 7.6 | 1.7 | 2.8 (14.0) |
| $X_{H_2O}$ (ppm) | 52° | 140 | 100 | 33 | 100 (950) |

**Table 5.** A comparison of typical errors calculated from the differences between TCCON and in situ $X_{\text{gas}}$ values ("MAD" in the table, i.e. mean absolute deviation), errors calculated from the error budget ("Budget" in the table), and the quadrature difference of the error budget and in situ error ("MAD – in situ"). "Gas" indicates which $X_{\text{gas}}$ product the error is for and the units of the values in the last four columns. "SZA" gives the solar zenith angle for which the error budget percent was taken to calculate the "Error budget" column. $\epsilon_{\text{in situ}}$ gives the total $2\sigma$ uncertainty on the in situ data. The first number in this column is the result of formally propagating the in situ error into its MAD value, the second number in parentheses is the simple mean across all the TCCON/in situ comparisons. Note that $\epsilon_{\text{in situ}}$ includes estimated uncertainty due to unmeasured parts of the profile; see Table C7 in the appendix for a breakdown of the individual error components.

Both $X_{CO}$ and $X_{H_2O}$ had larger MAD values than their respective error budgets. For $X_{CO}$, the difference in error estimates is 5.9 ppb. The "MAD - in situ" column uses the propagated value for $\epsilon_{\text{in situ}}$ (2.8 ppb). The uncertainty in individual comparisons (the parenthetical numbers in Table 5) is quite a bit larger; if part of this error is systematic (such as from drift in calibration tanks, e.g., Andrews, 2019), that could explain the remaining difference. For $X_{H_2O}$, this is because of uncertainty in the radiosondes used to compare against. The radiosondes used at ARM have a 4 or 5% uncertainty in relative humidity (https://www.arm.gov/publications/tech_reports/handbooks/sonde_handbook.pdf, last accessed 10 Apr 2023). When we propagate this uncertainty to the mean absolute deviation, it works out to 100 ppm. Subtracting this in quadrature from the MAD estimate reduces the mismatch in error estimates, but $\sim$ 70 ppm remains. We do note that in Fig. 16f, the TCCON/radiosonde values for $X_{\text{luft}} \approx 1$ (i.e. when the TCCON instrument was operating best) seem to be systematically $> 1$ at Darwin and $< 1$ at Lamont, suggesting that subtracting the in situ bias in quadrature might be underestimating its impact on the TCCON/radiosonde $X_{H_2O}$ mismatch.

## 10 Conclusions

The GGG2020 TCCON data product incorporate numerous improvements to the GGG retrieval, based both on first-principle understanding and empirical evaluation. To review:

– The interferogram-to-spectrum conversion has added checks and diagnostics for detector nonlinearity or saturation, as well as a modification to the phase correction that reduces bias between forward and reverse scans of the interferometer.

| Issue | Current mitigation | Future correction |
|---|---|---|
| $X_{CO_2}$ bias correlated with $X_{luft}$ | Data with out-of-family $X_{luft}$ excluded from public files, effect on public data passing this filter should be $< 0.5$ ppm | Empirical bias correction (GGG2020.1); improvements in ILS treatment (future major version) |
| $X_{N_2O}$ bias correlated with temperature | Users should be cautious when interpreting seasonal patterns of $X_{N_2O}$ | Empirical bias correction (GGG2020.1); improve $N_2O$ spectroscopy (future major version) |
| $X_{CH_4}$ airmass dependence correlates with temperature | None, effect expected to be minor | Empirical bias correction if needed (GGG2020.2), improve $CH_4$ spectroscopy (future major version) |

**Table 6.** Known biases in GGG2020 data along with current action taken by the TCCON data providers or recommended for users to take to mitigate the impact of each bias. The final column identifies future plans to correct each bias, with the GGG version in which those corrections will be implemented.

– The solar and telluric spectroscopic linelists used in the GGG forward model have been updated to reflect new laboratory and atmospheric/solar observing studies, to include non-Voigt lineshapes and line mixing, and to reduce an observed temperature and water dependence in the $O_2$ column amounts.

– The a priori inputs of atmospheric state (temperature, pressure, and composition) have increased temporal resolution and the trace gas profiles have been updated to better reflect both atmospheric growth rates of key species and gradients in their mixing ratios across the tropopause.

– Improvements to fitting the continuum and channel fringes in the spectra.

– A more flexible airmass correction applied to $X_{gas}$ value from individual spectral windows, rather than multi-window

averages of said values.

– A change to how retrieved $X_{gas}$ values from multiple spectral windows measuring the same gas are averaged together that eliminates a dependence on how many observations were averaged at once.

– An updated in situ correction factor that increases the number of profiles used to tie TCCON to the calibration scales used by in situ GHG measurements.

– Improvements to user-friendliness in how AKs and prior profiles are reported in public files.

There remains work to be done to further improve the TCCON data product. Implementing the capability in GGG to account for errors in ILS remains a high priority. This was planned for inclusion in GGG2020, but could not be completed in time. It is expected that this capability will be an important tool to eliminate the $X_{CO_2}$ bias seen in comparison with in situ profiles as $X_{luft}$ deviates from its nominal 0.999 value. A second high priority objective is to investigate the temperature dependence

seen in the $N_2O$ and (to a much lesser extent) $CH_4$ data and correct the underlying spectroscopic terms.

We currently plan to develop minor releases, GGG2020.1 and GGG2020.2, within the next several years to address the highest priority issues (Table 6) that will include additional post-processing bias corrections to address the bias of $X_{CO_2}$ versus $X_{luft}$ and $X_{N_2O}$ and $X_{CH_4}$ versus temperature. This may allow us to release data from the early years of several sites, which is currently flagged as poor quality due to out-of-bounds $X_{luft}$ as well as improve the $X_{N_2O}$ data substantially. As this would be a post-processing-only update, the reprocessing could be completed very rapidly.

At time of writing, 26 TCCON sites have reprocessed their existing data with GGG2020. Several sites are still in the process of carrying out this reprocessing, in many cases to improve the data quality based on new diagnostics available in GGG2020. Work is ongoing towards completing these sites' reprocessing. Extensions to the existing data records will be released monthly going forward.

## 11 Code and data availability

All TCCON GGG2020 data is linked through tccondata.org and stored as DOI-tagged datasets on CaltechDATA (data.caltech. edu). Each TCCON site has a separate repository and DOI on CaltechDATA; these are listed in Table 2. If a future correction requires a revision of previously published data, that revision will receive a new DOI. Users are encouraged to check tccondata. org for the latest revisions of data rather than relying on Table 2. A repository containing the full set of TCCON GGG2020 data is also available on CaltechDATA with the DOI 10.14291/TCCON.GGG2020 (Total Carbon Column Observing Network (TCCON) Team, 2022). Users are asked to cite the individual sites' data records rather than the combined record as this helps track usage of site data and thus support the ongoing operation of these sites. We provide a citation generator at https: //tccondata.org/metadata/siteinfo/genbib/. All data is provided in netCDF format, and additional documentation for the data is available at https://tccon-wiki.caltech.edu/. The GGG2020 retrieval software is archived on CaltechDATA (Toon, 2023) as well as publicly available through GitHub at https://github.com/TCCON/GGG.

## Appendix A: Abbreviations and additional tables

Abbreviations used in this paper are listed in Table A1. The retrieval windows used in GGG2020 are given in Tables A2 and A3.

## Appendix B: Error budget

For completeness, we include the error budget figures equivalent to Figures 23–25 for February and July at East Trout Lake in Figs. B1 to B6. February is extremely cold (-30 to -15°C) and dry (<500 ppm $X_{H_2O}$), with short days and large solar zenith angles. July is warm (20 to 30°C) and humid (3000 to 4500 ppm $X_{H_2O}$), causing the HF absorption feature to be blacked out by adjacent $H_2O$ lines at higher solar zenith angles, causing unreliable retrievals of HF.

| Abbreviation | Meaning | Notes |
|---|---|---|
| ADCF | Airmass dependent correction factor | See SZA note |
| AICF | Airmass independent correction factor | Also call the "in situ correction factor" |
| AK | Averaging Kernel | Refers to column averaging kernels unless otherwise indicated |
| CBF | Continuum basis function | |
| FT | Free troposphere | |
| FFT | Fast Fourier transform | |
| FOV | Field of view | |
| FTIR | Fourier transform infrared | |
| FTS | Fourier transform spectrometer | |
| FVSI | Fraction variation in solar intensity | |
| ILS | Instrument line shape | |
| IR | Infrared | |
| GGG | - | The name of the retrieval, not an abbreviation |
| GHG | Greenhouse gas | |
| LM | Line mixing | |
| MIR | Mid infrared | |
| MOPD | Maximum optical path difference | |
| MOPITT | Measurements of Pollution in the Troposphere | An instrument on the Terra satellite |
| NDIR | Nondispersive infrared | |
| NIR | Near infrared | |
| OSDS | Observer-sun Doppler stretch | |
| RH | Relative humidity | |
| RMS | Root mean square/squared | |
| qSDV | Quadratice speed-dependent Voigt | |
| SZA | Solar zenith angle | "SZA-" and "airmass-dependence" are used equivalently |
| TCCON | Total Carbon Column Observing Network | |
| UTC | Coordinated Universal Time | |
| VMR | Volume mixing ratio | |
| VSF | VMR scale factor | |
| $X_{\mathrm{gas}}$ | Column-average dry mole fraction | "$X_{\mathrm{gas}}$" is generic; "$X_{\mathrm{CO_2}}$", "$X_{\mathrm{CH_4}}$", etc. are specific |
| $X_{\mathrm{luft}}$ | Column-average dry mole fraction of dry air | This is a diagnostic quantity defined in Eq. (4) |
| ZLO | Zero level offset | |
| ZPD | Zero path difference | |

**Table A1.** Abbreviations used in this paper.

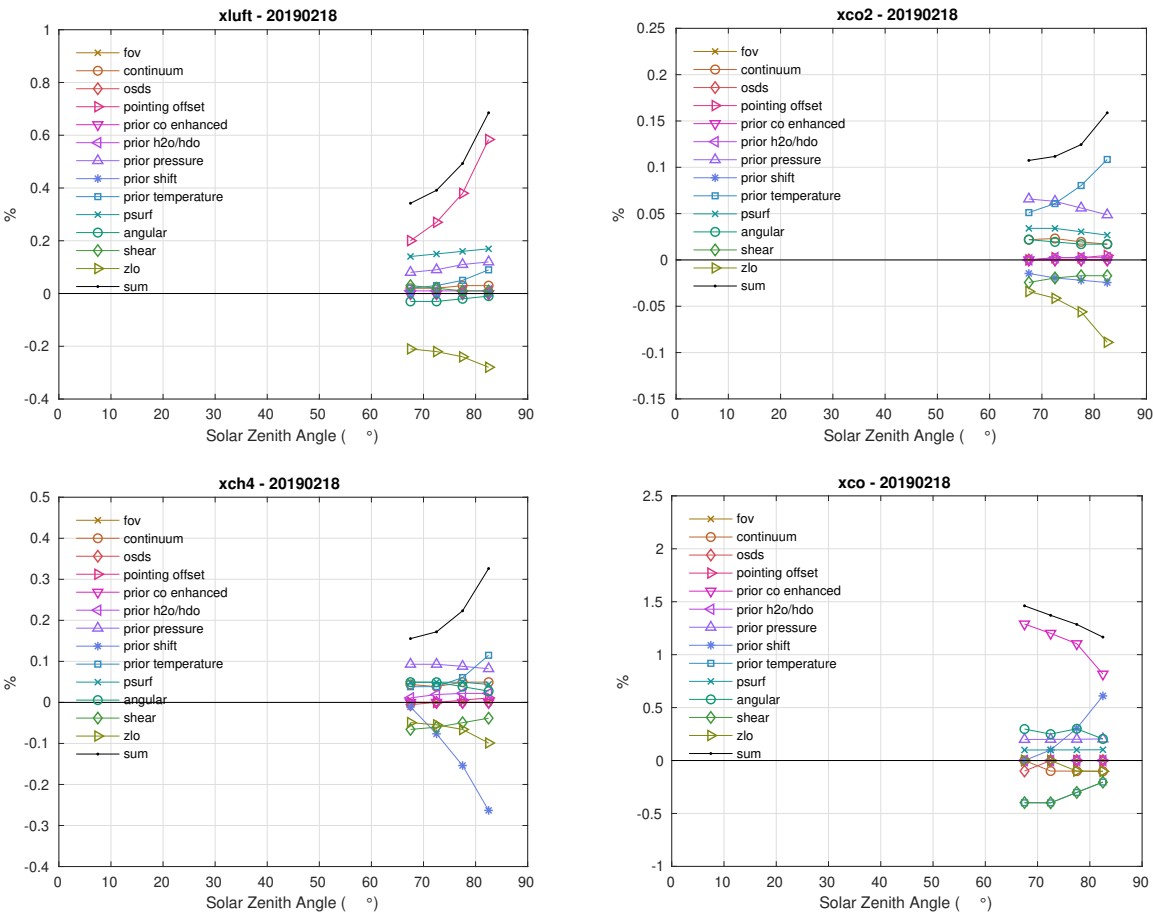

**Figure B1.** February 18, 2019 error budget from East Trout Lake. The figures show the percent difference between the perturbed test and the standard retrieval plotted as a function of solar zenith angle. The retrievals plotted here are $X_{\mathrm{luft}}$, $X_{\mathrm{CO_2}}$, $X_{\mathrm{CH_4}}$, and $X_{\mathrm{CO}}$.

| Freq. range (cm$^{-1}$) | Target gas | Other gases | qSDV | LM | # CBFs |
|---|---|---|---|---|---|
| 4262.2 to 4318.8 | CO | $CH_4$, $H_2O$, HDO | - | - | 4 |
| 4373.5 to 4416.9 | $N_2O$ | $CH_4$, $H_2O$, HDO | - | - | 4 |
| 4418.6 to 4441.7 | $N_2O$ | $CH_4$, $H_2O$, HDO, $CO_2$ | - | - | 2 |
| 4682.9 to 4756.1 | $N_2O$ | $CH_4$, $H_2O$, $CO_2$ | - | - | 3 |
| 5880.0 to 5996.0 | $CH_4$ | $CO_2$, $H_2O$, $N_2O$ | $CH_4$ | $CH_4$ | 4 |
| 5996.4 to 6007.6 | $CH_4$ | $CO_2$, $H_2O$, HDO | $CH_4$ | $CH_4$ | 2 |
| 6007.0 to 6145.0 | $CH_4$ | $CO_2$, $H_2O$, HDO | $CH_4$ | $CH_4$ | 5 |
| 4809.7 to 4896.0 | $lCO_2$ | $^{13}CO_2$, $C^{16}O^{18}O$, $C^{16}O^{17}O$, $H_2O$, HDO | $lCO_2$ | $lCO_2$ | 3 |
| 6041.8 to 6105.2 | $wCO_2$ | $H_2O$, $CH_4$ | $CH_4$ | $CH_4$ | 2 |
| 6180.0 to 6260.0 | $CO_2$ | $H_2O$, HDO, $CH_4$ | $CO_2$ | $CO_2$ | 3 |
| 6297.0 to 6382.0 | $CO_2$ | $H_2O$, HDO | $CO_2$ | $CO_2$ | 3 |
| 7765.0 to 8005.0 | $O_2$ | $O_2$ CIA, $H_2O$, HF, $CO_2$, HDO | $O_2$ | - | 5 |

**Table A2.** Retrieval windows used for CO, $N_2O$, $CH_4$, $O_2$ and the three $CO_2$ products in GGG2020. "Freq. range" gives the edges of the window. "Target gas" gives the gas whose column abundance is obtained from that window. "Other gases" lists gases that are retrieved as interferents in that window (note that the $lCO_2$ window includes several $CO_2$ isotopologs and the $O_2$ window includes the $O_2$ collision induced absorption). "qSDV" and "LM" indicate which species in that window include non-Voigt speed dependent and line mixing lineshape information, respectively, for their main lines (§6.2). A dash in these columns indicates no gas has speed dependence or line mixing information, respectively. Note that $CH_4$ uses full line mixing, while $CO_2$ uses the Rosenkranz approximation. "# CBF" indicates the number of basis functions used to fit the continuum in that window (§7).

## B1 ILS

We created synthetic spectra in GGG2020 with different ILS errors, following the formulation for the "shear" and "angular" misalignments tested for the GGG2014 error budget, and for the new formulation in GGG2020. We then passed these synthetic spectra through an ILS quantification program called LINEFIT (v14.8) (Hase et al., 1999), which calculates the modulation efficiency and phase error of the spectra. Here, we plot the LINEFIT-derived modulation efficiencies for these four cases in Figure B7. The GGG2020 shear and angular misalignments represent a ramp-up and ramp-down from 1.0 at zero path difference to 5% offsets at 45 cm optical path difference, as expected. Unfortunately, the GGG2014 "shear" and "angular" misalignments both model shear misalignments of different magnitudes. The GGG2014 "shear" case is, in fact, more like a 15% ramp up as a function of optical path difference, and the GGG2014 "angular" case is more like a 3% ramp up. This will essentially double the inferred error from the ILS in GGG2014, when compared with GGG2020.

| Freq. range (cm$^{-1}$) | Target gas | Other gases | qSDV | LM | # CBFs |
|---|---|---|---|---|---|
| 4038.8 to 4039.1 | HF | $H_2O$ | - | - | 2 |
| 4563.9 to 4566.4 | $H_2O$ | $CO_2$, $CH_4$ | - | - | 2 |
| 4568.8 to 4571.9 | $H_2O$ | $CO_2$, $CH_4$ | - | - | 2 |
| 4570.5 to 4573.0 | $H_2O$ | $CO_2$, $CH_4$ | - | - | 2 |
| 4575.9 to 4577.8 | $H_2O$ | $CH_4$ | - | - | 2 |
| 4593.3 to 4604.1 | $H_2O$ | $CH_4$, $CO_2$, $N_2O$ | - | - | 2 |
| 4609.9 to 4612.2 | $H_2O$ | $CH_4$, $CO_2$, $N_2O$ | - | - | 2 |
| 4620.9 to 4623.1 | $H_2O$ | $CO_2$, $N_2O$ | - | - | 2 |
| 4630.9 to 4632.2 | $H_2O$ | | - | - | 2 |
| 4697.6 to 4701.6 | $H_2O$ | $CO_2$, $N_2O$ | - | - | 2 |
| 4731.0 to 4738.2 | $H_2O$ | $CO_2$, $N_2O$ | - | - | 2 |
| 4755.8 to 4766.5 | $H_2O$ | $CO_2$ | - | - | 2 |
| 6075.0 to 6078.8 | $H_2O$ | $CH_4$, HDO, $CO_2$ | - | - | 2 |
| 6098.9 to 6099.8 | $H_2O$ | $CO_2$ | - | - | 2 |
| 6125.1 to 6126.6 | $H_2O$ | HDO, $CO_2$, $CH_4$ | - | - | 2 |
| 6176.9 to 6178.1 | $H_2O$ | HDO, $CO_2$, $CH_4$ | - | - | 2 |
| 6254.1 to 6257.8 | $H_2O$ | $CO_2$, HDO | $CO_2$ | $CO_2$ | 2 |
| 6297.4 to 6305.3 | $H_2O$ | $CO_2$, HDO | - | - | 2 |
| 6390.9 to 6394.0 | $H_2O$ | HDO | - | - | 2 |
| 6400.6 to 6401.7 | $H_2O$ | HDO, $CO_2$ | - | - | 2 |
| 6467.9 to 6471.4 | $H_2O$ | $CO_2$, HDO | - | - | 2 |
| 4052.9 to 4056.2 | HDO | $H_2O$, $CH_4$ | - | - | 2 |
| 4063.2 to 4072.0 | HDO | $H_2O$, $CH_4$ | - | - | 2 |
| 4112.1 to 4120.1 | HDO | $H_2O$, $CH_4$ | - | - | 2 |
| 4211.5 to 4213.4 | HDO | $H_2O$, $CH_4$ | - | - | 2 |
| 4227.0 to 4238.0 | HDO | $H_2O$, $CH_4$, CO | - | - | 2 |
| 6307.3 to 6352.8 | HDO | $H_2O$, $CO_2$ | - | - | 4 |
| 6352.3 to 6402.5 | HDO | $H_2O$, $CO_2$ | $CO_2$ | $CO_2$ | 4 |
| 6437.4 to 6478.8 | HDO | $H_2O$, $CO_2$ | - | - | 4 |
| 5624.9 to 5625.2 | HCl | $H_2O$, $CH_4$ | - | - | 2 |
| 5687.1 to 5688.2 | HCl | $H_2O$, $CH_4$ | - | - | 2 |
| 5701.6 to 5702.4 | HCl | $H_2O$, $CH_4$ | - | - | 2 |
| 5734.8 to 5735.3 | HCl | $H_2O$, $CH_4$ | - | - | 2 |
| 5738.5 to 5740.0 | HCl | $H_2O$, $CH_4$ | - | - | 2 |

**Table A3.** Same as Table A2, but for HF, $H_2O$, HDO, and HCl.

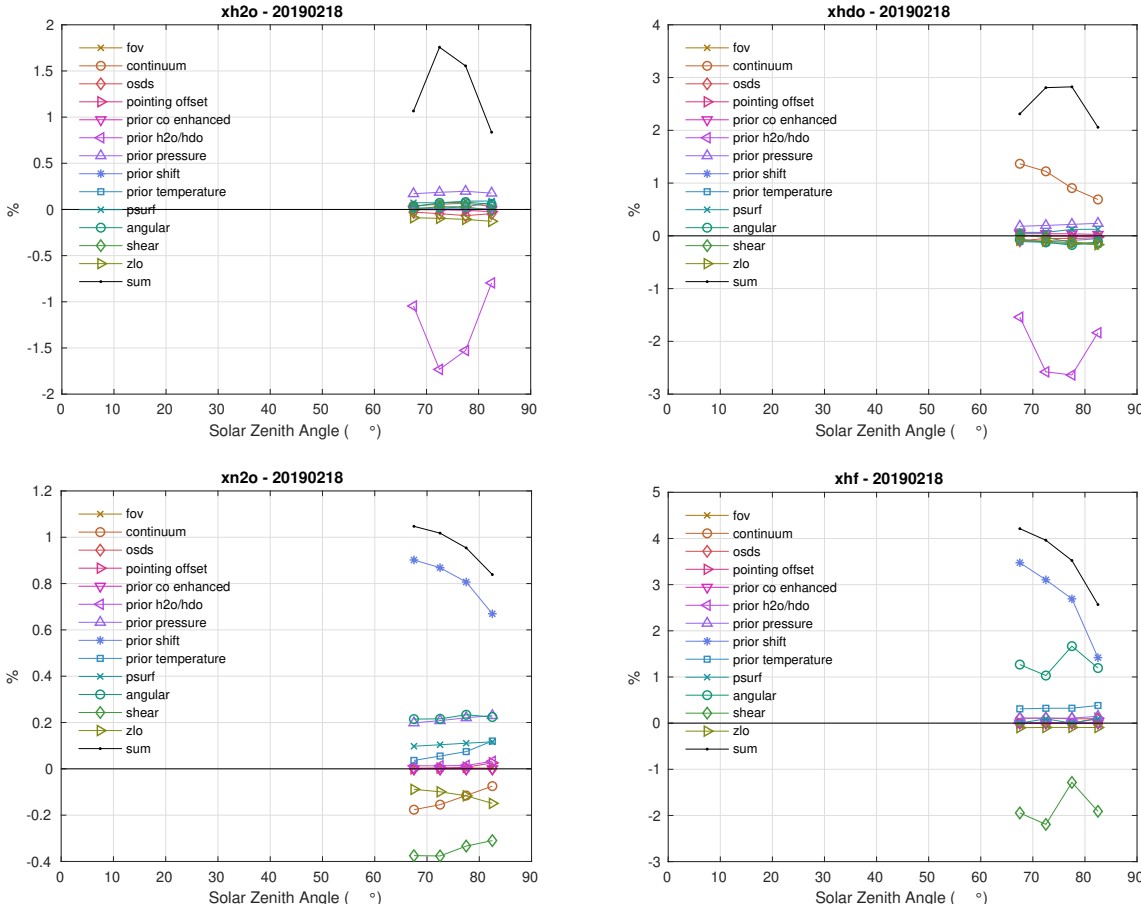

**Figure B2.** As in Figure B1, but for $X_{\mathrm{H_2O}}$, $X_{\mathrm{HDO}}$, $X_{\mathrm{N_2O}}$, and $X_{\mathrm{HF}}$.

## Appendix C: AICF profile selection

### C1  CO$_2$, CH$_4$, CO

In situ profiles for $CO_2$, $CH_4$, and CO were drawn primarily from the NOAA $CO_2$ ObsPack (Cooperative Global Atmospheric Data Integration Project, 2019), NOAA $CH_4$ ObsPack (Cooperative Global Atmospheric Data Integration Project, 2020), NOAA AirCore dataset (Baier et al., 2021), additional AirCore launches at the Sodanklyä and Nicosia TCCON sites, the Infrastructure for Measurement of the European Carbon Cycle (IMECC) campaign, and the GO-AMAZON campaign. The ObsPack contains data from numerous providers across different institutions; Tables C1 and C2 provide a detailed breakdown. For the NOAA ObsPack Aircraft and AirCore profiles, the procedure used to match these data to TCCON sites will be detailed

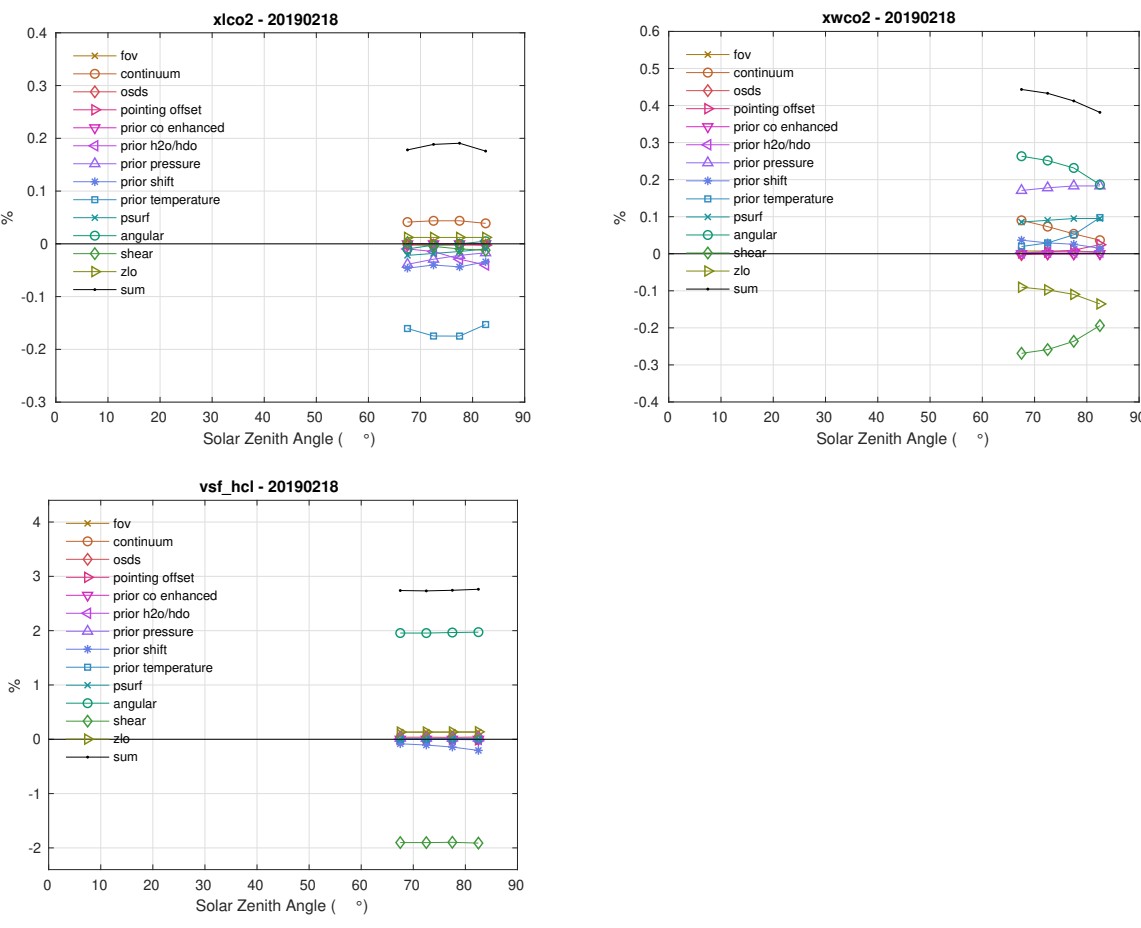

**Figure B3.** As in Figure B1, but for $X_{\text{l}CO_2}$, $X_{\text{w}CO_2}$, and HCl scale factors (vsf_hcl).

in the following subsections. For the remaining sources, the profiles were already associated with specific TCCON sites, so no colocation was required.

All airborne data sources used for these profiles are listed in Tables C1 and C2. Ground data used to extend some of the profiles to the surface are listed in Table C3.

### C1.1 ObsPack

The ObsPack data is provided as a single time series per measurement campaign or similar source. To extract individual profiles from these files, we:

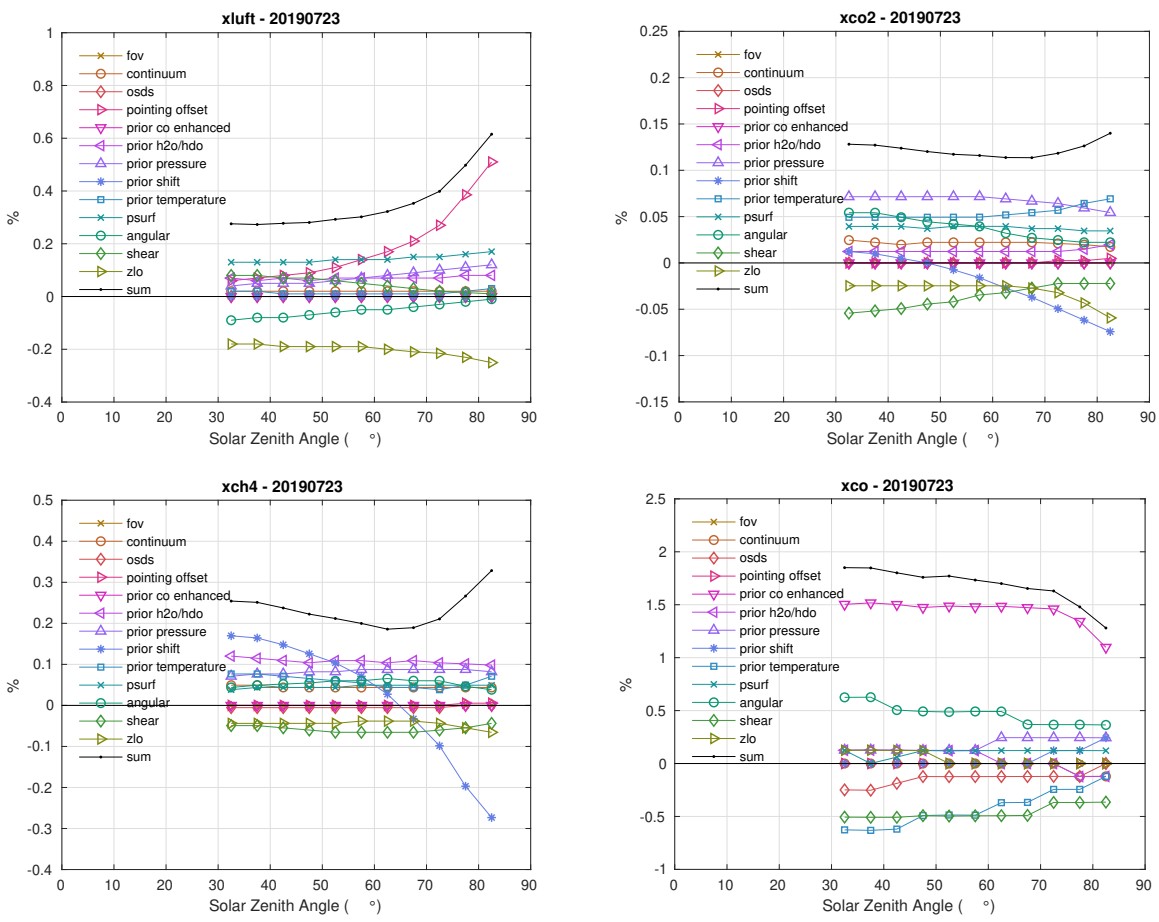

**Figure B4.** July 23, 2019 error budget from East Trout Lake. The figures show the percent difference between the perturbed test and the standard retrieval plotted as a function of solar zenith angle. The retrievals plotted here are $X_{luft}$, $X_{CO_2}$, $X_{CH_4}$, and $X_{CO}$.

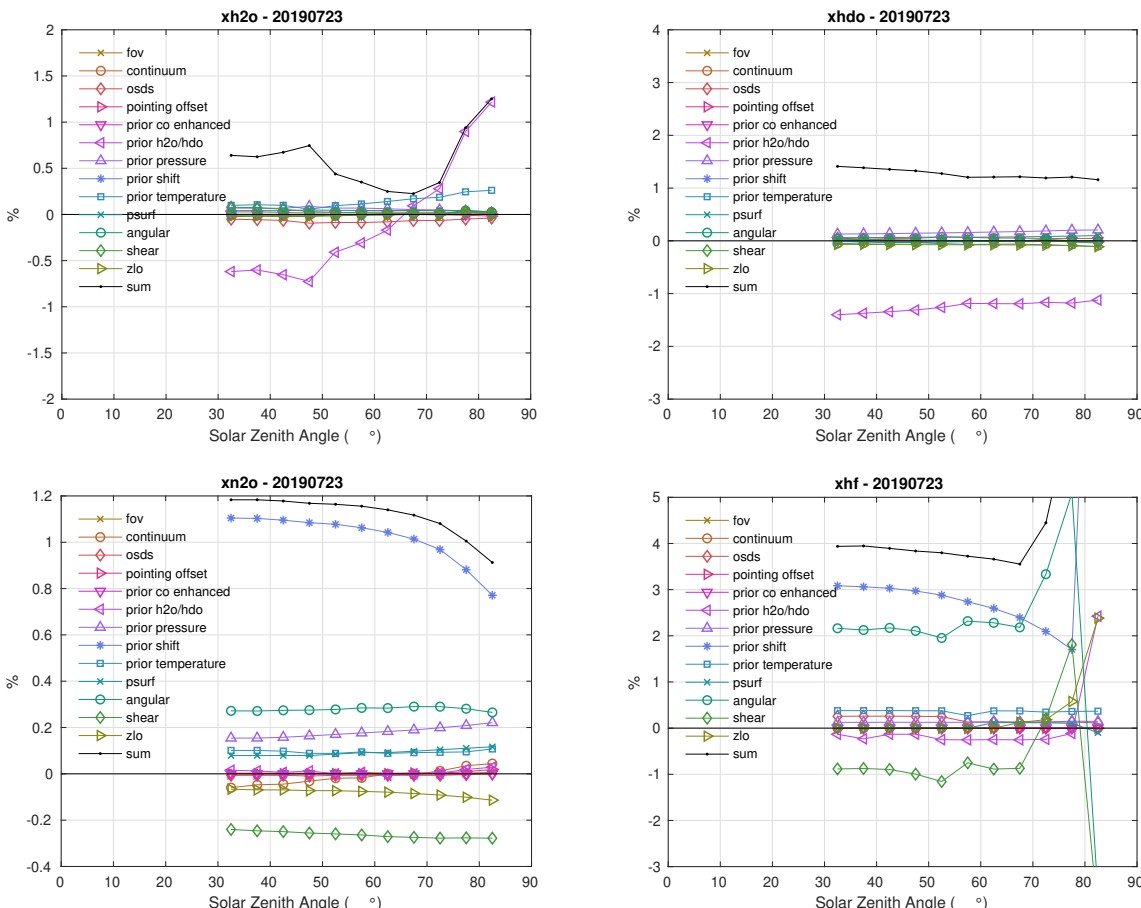

**Figure B5.** As in Figure 23, but for $X_{H_2O}$, $X_{HDO}$, $X_{N_2O}$, and $X_{HF}$.

1. Scan all files for data points within $2°$ (total distance) of an active TCCON site. When one is found, we store the list of data points surrounding it in time that fall a box $10°$ longitude width and $5°$ latitude tall centered on the TCCON site as a "chunk." A chunk extends forward and backward in time from the point closest to the TCCON site and stops at the first data point in each direction that is outside the $10° \times 5°$ box. Any profiles derived from this chunk are assigned to the TCCON station it passes closest to.

2. Further filter the chunks based on the lowest altitude, highest altitude, number of data points, and minimum distance to a TCCON site. This step was done interactively to find the filtering criteria that gave the best balance between number of chunks retained and the usefulness of the profile(s) within the chunk. The final criteria used were:

    – Minimum altitude below 2000 m

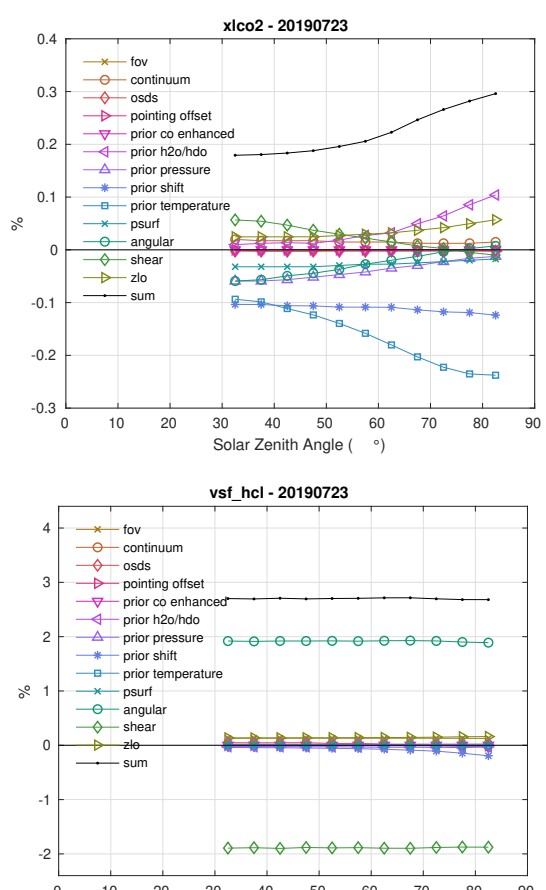

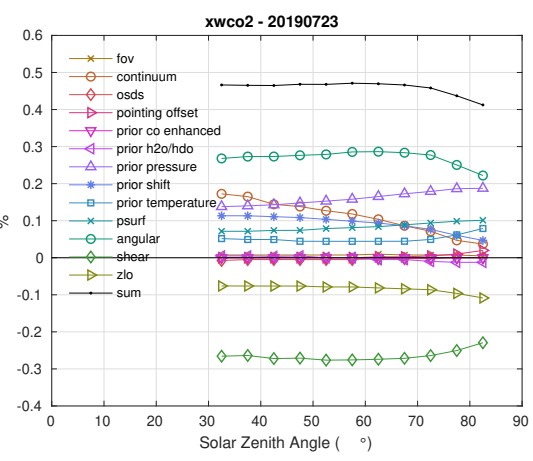

**Figure B6.** As in Figure B4, but for $X_{\mathrm{lCO_2}}$, $X_{\mathrm{wCO_2}}$, and HCl scale factors (vsf_hcl).

– Maximum altitude above 7500 m

– At least 20 data points

– Approached within 0.1° of a TCCON station

3. These filtered chunks were then individually evaluated and specific data points within them chosen by hand to use as profiles. In this process, we considered the latitude/longitude position of the aircraft, the profile of altitude versus time, and the profile of $CO_2$ or $CH_4$ versus altitude. We generally selected as profiles times when the aircraft was consistently ascending or descending, and excluded times of level flight. However, this had to be handled on a case-by-case basis to allow for profiles with a period of level flight in between two legs of an ascent or descent. If a chunk contained multiple ascending/descending legs, we would split them if:

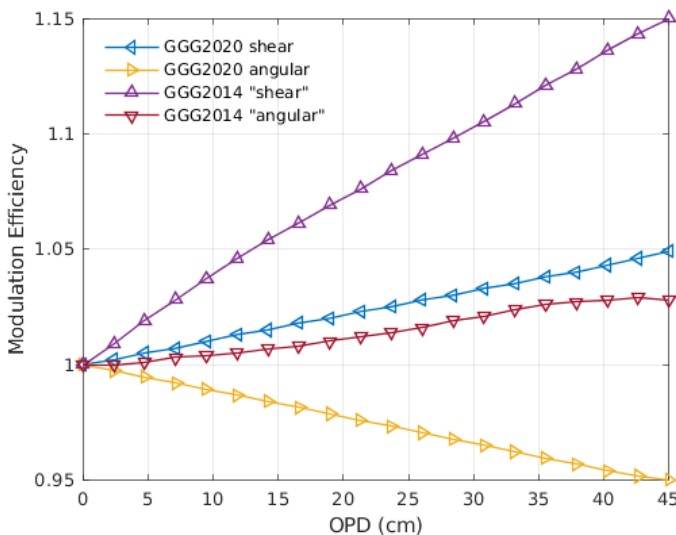

**Figure B7.** The modulation efficiencies tested in GGG2014 and GGG2020.

    – there was a clear separation in time, or

    – the legs measured different airmasses (evidenced by different $CO_2$ or $CH_4$ dry mole fractions)

4. For each profile, we check for ground data in the ObsPack that can be used to extend the profile to the surface. We identified which ground files in the ObsPack are near which TCCON sites by hand. We interpolate any data within 4 hours of the lowest altitude measurement in a profile to the time of the lowest altitude profile measurement. In cases where ground data is only available before or after the lowest profile measurement, we use the closest ground data in time.

### C1.2 AirCore

As AirCore data intrinsically provides discrete profile, matching these data to TCCON sites was much simpler. For NOAA AirCores, we search all files for those where the mean latitude and longitude of the profile were within 1° (total distance) of a TCCON site. We use a looser distance compared to the aircraft as it is unlikely that an AirCore would be within 1° of a TCCON site by happenstance if it was not intended to match with that TCCON. However, since it is possible that the balloon trajectory drifted significantly depending on the winds, we use the looser distance criterion to allow for that.

### C2 $H_2O$

Profiles for the $H_2O$ AICF come from radiosonde data provided by the Department of Energy Atmospheric Radiation Measurement (ARM) facility (Keeler and Burk). The data were downloaded from https://adc.arm.gov/discovery/#/results/instrument_

**Table C1.** Airborne profile data used in the AICF calculation. "$CO_2$ Obspack" is the $CO_2$ GLOBALVIEWplus v5.0 ObsPack (Cooperative Global Atmospheric Data Integration Project, 2019) and "$CH_4$ ObsPack" the $CH_4$ GLOBALVIEWplus v2.0 ObsPack (Cooperative Global Atmospheric Data Integration Project, 2020). The "TCCON sites" column indicates at which sites profiles were used; the IDs are mapped to locations in Table 2 and numbers of profiles per site are given in Tables C4 and C5. In the "Providers" column, affiliations are given in parentheses. If only one affiliation is listed, it applies to all individuals named. Abbrevations: NASA = National Aeronautics and Space Administration; LaRC = Langley Research Center; Harvard U. = Harvard University; CSUSB = California State University San Bernadino; GSFC = Goddard Space Flight Center; NCAR = National Center for Atmospheric Research; NOAA = National Oceanic and Atmospheric Administration; GML = Global Monitoring Laboratory; FMI = Finnish Meteorological Institure; CARE-C = Climate and Atmosphere Research Center; LSCE/IPSL = Laboratoire des Sciences du Climat et de l'Environnement.

| Source | Campaign or ID | Providers | TCCON sites |
|---|---|---|---|
| $CO_2$ ObsPack | $CO_2$ Budget and Regional Airborne Study - Maine (COB2004) | Steve Wofsy (Harvard U.) | pa |
| $CO_2$ ObsPack | Deep Convective Clouds & Chemistry (DC3), DC8 aircraft | Andreas Beyersdorf (CSUSB) & Yonghoon Choi (SSAI) | oc |
| $CO_2$ ObsPack | Goddard Space Flight Center (GSFC) | Stephan Randolph Kawa, James Brice Abshire, & Haris Riris (NASA GSFC) | df, pa |
| $CO_2$ ObsPack | HIAPER Pole-to-Pole Observations (HIPPO) | Steve Wofsy (Harvard U.), & Britton Stephens (NCAR) | ll, wg |
| $CO_2$ ObsPack | Intercontinental Chemical Transport Experiment - North America (INTEX-NA) | Stephanie A. Vay (NASA LaRC) & Yonghoon Choi (SSAI) | pa |
| $CO_2$ ObsPack | Korea-United States Air Quality Study (KORUS-AQ) | Joshua P. DiGangi, & Yonghoon Choi (SSAI) | an, df, rj |
| $CO_2$ ObsPack | $O_2/N_2$ Ratio and $CO_2$ Airborne Southern Ocean Study (ORCAS) | Britton Stephens (NCAR), Colm Sweeney (NOAA GML), Kathryn McKain (NOAA GML), Eric Kort (U. Michigan) | oc |
| $CO_2$ ObsPack | Studies of Emissions and Atmospheric Composition, Clouds and Climate Coupling by Regional Surveys (SEAC4RS), ER-2 aircraft | Steve Wofsy (Harvard U.) | df |
| $CO_2$ ObsPack | Studies of Emissions and Atmospheric Composition, Clouds and Climate Coupling by Regional Surveys (SEAC4RS), DC8 aircraft | Andreas Beyersdorf (CSUSB) & Yonghoon Choi (SSAI) | oc |

class_code::sonde%2Fprimary_meas_type_code::atmtemp in March 2021. Two ARM sites are close enough to TCCON locations to be useful: the Southern Great Plains (SGP) site's Central Facility (facility code C1) is near the Lamont, OK, USA

**Table C2.** Table C1, continued. ARM = Atmospheric Radiation Monitoring facility.

| Source | Campaign or ID | Providers | TCCON sites |
|---|---|---|---|
| $CO_2$ ObsPack | Stratosphere-Troposphere Analyses of Regional Transport (START-08) | Steve Wofsy (Harvard U.) | pa |
| $CO_2$ ObsPack | Atmospheric Tomography Mission (ATom) | Kathryn McKain (NOAA GML), Colm Sweeney (NOAA GML), Steve Wofsy (Harvard U.), Bruce Daube (Harvard U.), Roisin Commane (Harvard U.) | ae, df, eu, ll, oc, pa |
| Other $CO_2$ | NOAA Manaus | John Miller (NOAA GML) | ma |
| $CH_4$ ObsPack | HIAPER Pole-to-Pole Observations (HIPPO) | Steve Wofsy, Greg Santoni, & Jasna Pittman (Harvard U.) | ll,oc,pa,wg |
| $CH_4$ ObsPack | Stratosphere-Troposphere Analyses of Regional Transport (START08) | Steve Wofsy (Harvard U.) | pa |
| $CH_4$ ObsPack | Atmospheric Tomography Mission (ATom) | Kathryn McKain & Colm Sweeney (NOAA GML) | ae,ci,df,eu,ll,oc,pa |
| IMECC Repository ($CO_2$, $CH_4$, CO) | Infrastructure for Measurement of the European Carbon Cycle (IMECC) | Various | bi,br,gm,je,ka,or |
| NOAA AirCores ($CO_2$, $CH_4$, CO) | N/A | Bianca Baier & Colm Sweeney (NOAA GML) | df,oc,pa,so |
| Sodankylä AirCores ($CO_2$, $CH_4$, CO) | N/A | Huilin Chen (RUG) & Rigel Kivi (FMI) | so |
| Nicosia AirCores ($CO_2$, $CH_4$, CO) | N/A | Pierre-Yves Quéhé (CARE-C, Cyl) & Thomas Laemmel (LSCE/IPSL) | ni |
| Radiosondes ($H_2O$) | Southern Great Plains (SGP) Lamont Central Facility and Tropical Western Pacific (TWP) Darwin Facility | ARM | db, oc |

TCCON site, and the Tropical Western Pacific (TWP) site's Darwin facility (code C3) is near the Darwin, Australia TCCON site.

These facilities produce more radiosonde observations than we can feasibly use in the AICF calculation, so we must choose a subset. We use the following steps for each site:

1. Identify radiosonde profiles that are coincident with another trace gas profile ($CO_2$, CO, $CH_4$, or $N_2O$).

2. Identify radiosonde profiles not in the set identified in Step 1 that have at least 30 TCCON spectra within $\pm 3$ hours of the time of the profile's lowest altitude measurement and

3. Combine the profiles from step 1 with randomly selected profiles from step 2 to collect 50 total profiles. (We use a seed of 42—chosen in reference to "The Hitchhiker's Guide to the Galaxy"—to ensure repeatability across runs.)

4. Finally, remove any profiles from this set of 50 that have a maximum altitude $< 15$ km.

Once we have assembled a pool of radiosonde profiles, we convert the relative humidity (RH) values stored in the files to water dry mole fractions. Based on the convention described in Miloshevich et al. (2006), we assume that the definition of RH is the ratio of water vapor pressure to the saturation water vapor pressure over liquid water and calculate the $H_2O$ dry mole fraction as

$$f_{H_2O,\text{wet}} = \frac{\text{RH} \cdot \text{SVP}}{p} \tag{C1}$$

$$f_{H_2O,\text{dry}} = \frac{f_{H_2O,\text{wet}}}{1 - f_{H_2O,\text{wet}}} \tag{C2}$$

where RH is the relative humidity as a fraction (i.e. 0 to 1), SVP is the saturation vapor pressure of water over liquid water calculated using Eq. 6 of Miloshevich et al. (2004) (see also Eq. 15 of Wexler, 1976), and $p$ is the atmospheric pressure (in the same units as SVP).

### C3    Constructing full profiles

In order to ensure a proper comparison between the in situ and TCCON column amounts, the in situ profiles must extend to the top of the TCCON retrieval altitude grid, 70 km. No aircraft or balloon-borne profile reaches this altitude, therefore, similarly to Wunch et al. (2010), we extend the in situ profiles using the GGG2020 prior profiles (Laughner et al., 2023).

The differences between Wunch et al. (2010) and our approach stem from (1) the GGG2020 priors do a better job of representing trace gas profiles in the stratosphere and (2) we have enough additional profiles over TCCON sites to be selective

about which ones we use. This is why we filtered the ObsPack data to "chunks" that have data up to at least 7500 m altitude (§C1.1), to limit the altitude that needs to be filled in above the top of the profile.

There are three ways that profiles are extended up to 70 km altitude, depending on their top altitude:

1. If the profile's top is above 380 K potential temperature (i.e. reaches the stratospheric overworld), then we append the GGG2020 priors for levels above the profile top.

2. If the profile's top is below 380 K potential temperature but at or above 7.5 km, then the in situ profile's values are binned to the same altitude grid (see below) and then we do a constant value extrapolation of the top binned value up to the tropopause altitude. We use the GGG2020 prior above 380 K potential temperature again, and connect the two parts of the profile by linearly interpolating the trace gas dry mole fractions with respect to potential temperature between the tropopause and 380 K. This case covers profiles where the top of the measured profile is expected to be a better representation of the unmeasured free troposphere than the GGG2020 priors.

3. If the profile's top is below 7.5 km, then we use the GGG2020 priors for all levels above the profile top. The case assumes that profiles that do not reach above 7.5 km do not constrain the free troposphere well enough to supplant the GGG2020 priors. While we filtered the ObsPack data for "chunks" that have data above 7.5 km, we still have a few profiles with ceilings below 7.5 km from chunks that needed to be split into multiple profiles.

For #2, we calculate the binned in situ profile values for the highest altitude of the GGG retrieval grid below the in situ profile's ceiling ($z_{\mathrm{GGG,k}}$) as:

$$\overline{f}_{\mathrm{obs}} = \frac{\sum_{i=1}^{n_{\mathrm{obs}}} w_i f_{\mathrm{obs},i}}{\sum_{i=1}^{n_{\mathrm{obs}}} w_i} \tag{C3}$$

$$w_i = \begin{cases} (z_{\mathrm{obs},i} - z_{\mathrm{GGG},k-1})/(z_{\mathrm{GGG},k} - z_{\mathrm{GGG},k-1}) & \text{if } z_{\mathrm{GGG},k-1} \le z_{\mathrm{obs},i} < z_{\mathrm{GGG},k} \\ (z_{\mathrm{GGG},k+1} - z_{\mathrm{obs},i})/(z_{\mathrm{GGG},k+1} - z_{\mathrm{GGG},k}) & \text{if } z_{\mathrm{GGG},k} \le z_{\mathrm{obs},i} < z_{\mathrm{GGG},k+1} \\ 0 & \text{otherwise} \end{cases} \tag{C4}$$

Figure C1 shows an example of the weights for one short profile at the Armstrong TCCON site.

There is a special case for $CH_4$ applied when integrating the in situ profile to calculate the in situ-derived $X_{CH_4}$. Previous work (e.g. Washenfelder et al., 2003; Saad et al., 2014, 2016) established that there is a strong correlation between $CH_4$ and HF in the stratosphere. Since this correlation is encoded into the GGG2020 priors (Laughner et al., 2023), we can use the difference between the prior and posterior HF column (which is almost entirely found in the stratosphere) from the TCCON retrievals to adjust the levels in the in situ $CH_4$ profiles that use the GGG2020 profiles.

Specifically, when calculating the in situ $X_{CH_4}$, we get the slope of $CH_4$ vs. HF mixing ratios used by the GGG2020 priors for the year and region (tropics, midlatitudes, or polar vortex) of the profile (see §3.5 and Fig. 11 of Laughner et al., 2023). We then multiply this slope by the difference between the prior and median posterior HF profile of all the TCCON observations matched with the in situ profile in question in order to get the expected change in the $CH_4$ priors to better match the true stratospheric profile. Finally, we multiply this profile difference by the TCCON AK and integrate only the levels in the total in situ profile obtained from the GGG2020 priors. The integration uses Eq. (10) and add the integrated change to the in situ $X_{CH_4}$ as a posterior adjustment.

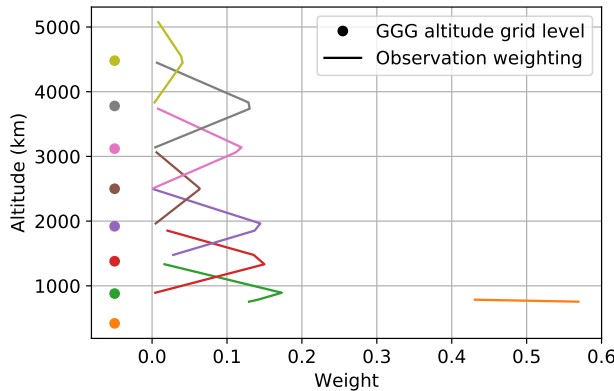

**Figure C1.** An example of the weighting functions from Eq. (C4). Lines indicate the weights applied to the observed mole fractions and circles indicate the GGG altitude grid levels that correspond to those weights—like colors match.

Again, note that this correction is only applied when integrating the in situ profiles to obtain the true $X_{CH_4}$ value to compare the TCCON retrievals against. When using the in situ profiles as priors in the TCCON retrievals, the levels taken from the GGG2020 priors are not adjusted in this fashion.

## C4 Grouping temporally proximate profiles

There are several cases where multiple profiles are available within a short time of each other (such as different legs of a missed approach or duplicate AirCore launches). Because we use the observed profiles as the prior in the TCCON retrievals from which the AICF is derived, this presents a technical challenge. Ideally, we want to use the same prior for all retrievals matched up with a given profile for comparison. Our temporal coincidence criterion can be up to $\pm 3$ hours, therefore, in cases with two or more profiles within a few hours, if for each TCCON retrieval we used the observed priors closest in time to it, this would result in a change of prior partway through our coincidence window.

Our solution was to merge profiles close enough in time for this to occur, but only for use as priors. Each individual observed profile still contributes one point on Fig. 16. This does mean that the prior will not exactly match any of the observed profiles those retrievals are compared against, but we consider that an acceptable error, given that we do apply an AK correction to the integrated in situ profile.

To find profiles that need to be merged, we first identify which TCCON observations would match with that profile. We ignore the quality filtering criteria from §8.3.1 during this step, and only try to find the time window ($\pm$ 1, 2, or 3 hours) necessary to match at least 30 TCCON observations to each profile. If any two profiles from the same TCCON site are matched to any of the same TCCON observations, they are grouped together in the list of profiles, to be averaged together when creating the custom priors in §C5. This initial list is written out to a text file so that it can be modified by hand later, as needed.

## C5 Running custom TCCON retrievals

As mentioned in §8.3, when we run the TCCON retrievals for the AICF calculation, we use as custom priors the in situ profiles that a given TCCON observation will be compared against. This reduces error in the TCCON $X_{gas}$ value that arises from an incorrect prior profile and thus improves the accuracy of the AICF. There are several technical considerations in how we handle this matching. In order to make those considerations clear, let us first describe how the GGG retrieval accepts inputs describing both the prior profiles and the TCCON observations to retrieve on.

GGG takes a list of TCCON spectra to retrieve as input in the "runlog" file. This lists each spectrum on which to run the retrieval in order. For the AICF retrievals, we combined all the spectra from all the relevant TCCON sites into a single runlog.

The priors (including temperature and pressure as well as trace gas mixing ratios) are written to a ".mav" file. This file is organized into blocks. Each block indicates the first spectrum from the runlog which the priors contained in the .mav block apply to. During the retrieval, GGG iterates through the spectra contained in the runlog. When it reaches the spectrum defined as the first spectrum of the next block in the .mav file, it loads the priors from that block before continuing.

In inserting the in situ profiles into the .mav file as priors, we had three objectives:

1. Retain the standard priors for gases and times that we did not have in situ profiles available.

2. Ensure that the in situ profiles were used as priors for any spectra that they *might* be compared against.

3. Ensure that any in situ profiles were only applied to the TCCON site where and day when they were measured.

To meet these objectives, our approach to inserting the in situ profiles as priors was:

– Divide the runlog into chunks by site and day, so that each chunk only has spectra from one site on one day.

– For each unique site/day chunk, collect all the in situ profiles from that day.

– Average together any in situ profiles grouped together in the list created in §C4. For this, we used an approach that considers whether each in situ profile contributed observations to a given level in the regridded profile. For a level on the retrieval grid where none of the in situ profiles provided any data points (i.e. the observed profiles were extrapolated or had the GGG2020 prior appended to it), both profiles are weighted equally. For a level where at least one of the in situ profiles had observed data, each profile is weighted by the fraction of data for that level that came from observations.

– For gases that only have one profile (after averaging) for that site/day, assign that profile to all the .mav blocks for that site/day.

– For gases that have multiple profiles that are not merged together (§C4), use the first profile in the day for all .mav blocks up until the first spectrum that could be compared with the second profile in the day (for our coincidence criteria, this will be the spectra 3 hours before the floor time of the second profile). Introduce a new .mav block on that profile that switches to the second profile. Repeat for third, fourth, etc. profiles if present. Assign the last profile to cover all .mav blocks through the end of the day.

Once the profiles are assigned to their .mav blocks, they must be averaged from their native vertical resolution to the GGG retrieval altitude grid and, if multiple profiles for the same gas were present for the same block, they must be averaged together.

For the vertical regridding, we use the same approach as described in §C3 where we do a weighted average of the observed mixing ratios, where the weights are maximized when the observed altitude equals the altitude of the GGG retrieval level they are being averaged to, and which decrease linearly to the adjacent GGG retrieval levels (Fig. C1, Eq. C4).

We found that it is crucial that we use geopotential height as the altitude for the regridding, as that did a better job ensuring that the observed profiles followed hydrostatic balance. To compute geopotential height for the in situ profiles, we take pressure and geopotential height from the two GEOS FP-IT files (Lucchesi, 2015) that bound the profile's lowest altitude in time and average the GEOS FP-IT data, and weight each by the time difference between the GEOS FP-IT profile and the time of the lowest altitude measurement in the in situ profile, giving greater weight to profiles nearer in time to the in situ profile. We then interpolate the GEOS FP-IT geopotential altitude on the logarithm of pressure to the pressures in the in situ profile.

The final consideration in preparing the custom priors is that we always retain the pressure and temperature profiles from the standard GEOS FP-IT priors used in GGG2020. This is because our testing found it very difficult to maintain hydrostatic balance if we used the observed pressure and temperature. This, in turn, caused greater error in the retrieved $X_{\mathrm{gas}}$ values, as the air column would be incorrect.

Once the custom priors were generated, the TCCON retrievals could be run as normal. The standard post processing corrections for airmass dependence (§8.1) and window-to-window averaging (§8.2) were applied as well. AKs were calculated for each spectrum retrieved as used to smooth the in situ profiles and account for the TCCON vertical sensitivity (§8.3).

## C6 Uncertainty in TCCON/in situ comparisons

For the TCCON/in situ ratios in §8.3, we considered five sources of uncertainty for the comparisons. We chose twice the standard deviation as our metric for deriving uncertainty (rather than $1\sigma$ to be conservative), and use that consistently for all random error terms.

**1. In situ measurement error** ($\epsilon_{\mathrm{meas}}$): This accounts for the error in individual in situ measurements that make up the profiles. To be conservative, we assume the worst-case scenario with 100% correlated error at all levels. The uncertainty in $X_{\mathrm{gas}}$ is then calculated as:

$$\epsilon_{\mathrm{meas}} = \int c(p) + 2\sigma(p)\, dp - \int c(p)\, dp \tag{C5}$$

where $c(p)$ is the measured mixing ratio and $\sigma(p)$ the uncertainty at each level. The integrals represent the pressure-weighted integration, Eq. (10). The uncertainty values are those reported in the original data files where available or a typical value chosen in consultation with the data providers.

**2. Unmeasured free troposphere** ($\epsilon_{\mathrm{FT}}$): This accounts for uncertainty due to the portion of the free troposphere not measured by a given profile. For each profile, we first calculate $\sigma_{\mathrm{obs,FT}}$, the standard deviation of measurements above 750 hPa

and below the tropopause (as determined by GEOS FP-IT meteorology). We then create a perturbed profile,

$$c'(p) = \begin{cases} c(p) + 2\sigma_{\text{obs,FT}} & \text{if interp/extrap at p} \\ c(p) & \text{otherwise} \end{cases} \tag{C6}$$

which adds this standard deviation to interpolated or extrapolated levels above the top of the measured profile. The uncertainty in $X_{\text{gas}}$ is calculated as:

$$\epsilon_{\text{FT}} = \int c'(p)\, dp - \int c(p)\, dp \tag{C7}$$

This error will be zero for profiles that do not require extrapolation or interpolation to reach the stratospheric overworld (i.e. altitudes with potential temperature $\geq 380$ K).

**3. Bias in stratospheric prior** ($\epsilon_{\text{strat}}$): This represents expected bias in the column from the use of GGG2020 priors for levels in the stratosphere. This uses the retrieved vs. prior HF column as a proxy for error in the stratospheric prior. As discussed in §8.3.3, HF is predominately found in the stratosphere, so the difference between the retrieved and prior HF columns gives information about whether the stratospheric profile was biased high or low. We calculate the bias as:

$$\epsilon_{\text{strat}} = 2 \cdot (X_{\text{HF,post}} - X_{\text{HF,prior}}) \cdot \frac{\partial X_{\text{gas}}}{\partial X_{\text{HF}}} \tag{C8}$$

The derivative $\partial X_{\text{gas}}/\partial X_{\text{HF}}$ has to be calculated for each gas. For $CO_2$ we use $8.09 \times 10^3$, which was derived from East Trout Lake TCCON data by comparing prior and posterior $wCO_2$ and HF columns. (East Trout Lake is positioned to see significant stratospheric variability due to the polar vortex, and $wCO_2$ is the GGG2020 $CO_2$ product with enhanced sensitivity to the stratosphere.) For $CH_4$, this is drawn from the $CH_4$:HF slopes used in the GGG2020 priors (Laughner et al., 2023).

AirCore profiles are treated specially, as they always reach into the stratosphere. For these profiles, we create a perturbed profile, $c'(p)$, where the levels in the stratosphere filled by the GGG2020 priors have the difference between the top of the AirCore profile and the corresponding level in the prior added to them. The difference between the integral of these profiles become the stratospheric error. Mathematically, that is

$$c'(p) = \begin{cases} c(p) + 2[c_{\text{prior}}(p_{\text{obs. top}}) - c_{\text{AirCore}}(p_{\text{obs. top}})] & \text{if using prior at p} \\ c(p) & \text{otherwise} \end{cases} \tag{C9}$$

$$\epsilon_{\text{strat,AirCore}} = \int c'(p)\, dp - \int c(p)\, dp \tag{C10}$$

**4. Random error in TCCON $X_{\text{gas}}$ value** ($\epsilon_{\text{std. xgas}}$): This represents random error in the TCCON observations. Because we require at least 30 TCCON observations coincident with a profile for a valid comparison, we use twice the standard deviation among those coincident observations as the metric of random error. The coincidence windows vary between 2 and 6 hours wide, so the standard deviation likely includes some true change in the data, and can therefore be considered conservative.

**5. Bias in TCCON derived from $X_{\text{luft}}$** ($\epsilon_{X_{\text{luft}}}$): This represents bias in retrieved $X_{\text{gas}}$ values resulting from instrument hardware issues diagnosed from deviations in $X_{\text{luft}}$ from the nominal network value (0.999, see §8.3). The bias is calculated

as:

$$\epsilon_{X_{\mathrm{luft}}} = \frac{\partial r}{\partial X_{\mathrm{luft}}} \cdot (X_{\mathrm{luft,median}} - 0.999) \cdot X_{\mathrm{gas,median}} \tag{C11}$$

Here, $X_{\mathrm{luft,median}}$ and $X_{\mathrm{gas,median}}$ are the median values of TCCON $X_{\mathrm{luft}}$ and the target $X_{\mathrm{gas}}$ across the 30+ coincident observations for the comparison. 0.999 is the nominal value of $X_{\mathrm{luft}}$ that represents a well-operating instrument. The $\partial r/\partial X_{\mathrm{luft}}$ value is how the TCCON/in situ ratio changes with $X_{\mathrm{luft}}$, and was derived for $X_{\mathrm{CO_2}}$, $X_{\mathrm{wCO_2}}$, $X_{\mathrm{lCO_2}}$, and $X_{\mathrm{CH_4}}$ by an unweighted robust fit through similar plots of TCCON/in situ ratio vs. $X_{\mathrm{luft}}$ as Fig. 16, but with TCCON retrievals that used the standard trace gas priors instead of custom ones built from the in situ profiles. The values used are given in Table C6.

**Full error calculation:** As the error terms include a mix of random ($\epsilon_{\mathrm{meas}}$, $\epsilon_{\mathrm{FT}}$, $\epsilon_{\mathrm{std.\ xgas}}$) and systematic ($\epsilon_{\mathrm{strat}}$, $\epsilon_{X_{\mathrm{luft}}}$) errors, the in situ and TCCON total errors are calculated as:

$$\epsilon_{\mathrm{in\ situ}} = \sqrt{\epsilon_{\mathrm{meas}}^2 + \epsilon_{\mathrm{FT}}^2} + |\epsilon_{\mathrm{strat}}| \tag{C12}$$

$$\epsilon_{\mathrm{TCCON}} = \sqrt{\epsilon_{\mathrm{std.\ xgas}}^2} + |\epsilon_{X_{\mathrm{luft}}}| \tag{C13}$$

The first term in the second equation is written as a root of a square to indicate that if additional random TCCON error terms were to be added in the future, they should add in quadrature. The uncertainty in the TCCON/in situ ratio ($X_{\mathrm{gas,TCCON}}/X_{\mathrm{gas,in\ situ}}$) follows standard error propagation ($\epsilon_{\mathrm{total}} = \sum_i (\sigma_x \cdot \partial f(x)/\partial x)^2$):

$$\epsilon_{\mathrm{total}} = \frac{\epsilon_{\mathrm{TCCON}}^2}{\epsilon_{\mathrm{in\ situ}}^2} + \frac{\epsilon_{\mathrm{in\ situ}}^2 X_{\mathrm{gas,TCCON}}^2}{\epsilon_{\mathrm{in\ situ}}^4} \tag{C14}$$

Note that Eq. (C12) is applied to each individual TCCON/in situ comparison, while the statistics in Table 5 are averaged over all the comparisons for a given gas. Therefore, the values of $\epsilon_{\mathrm{in\ situ}}$, $\epsilon_{\mathrm{meas}}$, $\epsilon_{\mathrm{FT}}$, and $\epsilon_{\mathrm{strat}}$ in Table 5 do not directly relate to each other through Eq. (C12). As noted in the caption for Table 5, the non-parenthetical values in the last four columns formally propagate the error from the individual comparisons, such that the values shown in the table (which we will denote generally as $\epsilon_{\mathrm{formal}}$) are calculated from the individual comparisons' values with

$$\epsilon_{\mathrm{formal}}^2 = \sum_{i=1}^{n} \left( \frac{1}{n} \epsilon_{\mathrm{indiv},i} \right)^2 \tag{C15}$$

where $\epsilon_{\mathrm{indiv},i}$ denotes individual comparisons' error values and $n$ is the number of individual observations. Conversely, the parenthetical numbers in Table 5 give the simple mean, i.e.:

$$\epsilon_{\mathrm{mean}} = \frac{1}{n} \sum_{i=1}^{n} \epsilon_{\mathrm{indiv},i} \tag{C16}$$

## Appendix D: Comparison between TCCON and NOAA surface N$_2$O

For Fig. 19, we constructed N$_2$O profiles to compare TCCON $X_{N_2O}$ against using NOAA surface data. This approach takes advantage of how well-mixed N$_2$O is in the troposphere to build a large set of comparison. The approach, in detail, is as follows.

The TCCON vs. in situ comparison shown in Fig. 19 calculates an in situ $X_{N_2O}$ from N$_2$O profiles using Eq. (9) as with the other $X_{gas}$ quantities in §8.3. These N$_2$O profiles are constructed using the NOAA surface N$_2$O VMR from the surface to the tropopause, the GGG2020 N$_2$O prior for levels with potential temperature greater than 380 K, and linearly interpolating the N$_2$O VMR with respect to potential temperature between the tropopause and 380 K level.

For the tropospheric N$_2$O VMRs, we obtained monthly average NOAA global N$_2$O data from https://gml.noaa.gov/hats/combined/N2O.html (last access 10 May 2021). For sites at latitudes north of 23° N or south of 23° S, we use the northern and southern hemispheric averages, respectively (`GML_NH_N2O` and `GML_SH_N2O` in the combined NOAA N$_2$O file). For equatorial latitudes between 23° S and 23° N, we used the average of the Mauna Loa and American Samoa N$_2$O data (`GML_mlo_N2O` and `GML_smo_N2O` in the combined file). For each comparison point in Fig. 19, we used the N$_2$O VMR from that month as the tropospheric VMR of the profile.

The comparisons selected for Fig. 19 meet the following criteria:

– The difference between the prior and posterior HF column must be $< 2 \times 10^{14}$ molec. cm$^{-2}$ in magnitude. Since HF is almost entirely in the stratosphere, this limits the comparisons to cases where the GGG2020 prior stratospheric profiles are reasonably accurate, thus limiting error in the in situ $X_{N_2O}$ from an incorrect assumed stratosphere

– $X_{luft}$ must be in the range $[0.996, 1.002)$. This ensures we are considering data when the TCCON instrument was well aligned, as discussed in §8.3.1

– FVSI must be $\leq 0.05$. This limits the comparison to mostly cloud-free observations.

## Appendix E: Variable O$_2$ dry mole fraction derivations

### E1 Trends in O$_2$ dry mole fraction from trends in $X_{CO_2}$

The derivation of Eq. (12) begins from the definition of $f_{O_2}$:

$$f_{O_2} = \frac{N_{O_2}}{N + N_{O_2} + N_{CO_2}} \tag{E1}$$

where:

– $N_{O_2}$ and $N_{CO_2}$ are the number of moles of O$_2$ and CO$_2$, respectively,

– $N$ is the number of moles of gases other than O$_2$ or CO$_2$ in H$_2$O-free air, and

- $N_\text{tot}$ (used below) is $N + N_{O_2} + N_{CO_2}$

Defining $\alpha = \partial N_{O_2}/\partial N_{CO_2}$, taking the derivative of $f_{O_2}$ with respect to $N_{CO_2}$, and simplifying gives:

$$\frac{\partial f_{O_2}}{\partial N_{CO_2}} = \left( \frac{\alpha(N + N_{CO_2})}{N_\text{tot}} - \frac{N_{O_2}}{N_\text{tot}} \right) \cdot \frac{1}{N_\text{tot}} \tag{E2}$$

Recognizing that $N_{O_2}/N_\text{tot} = f_{O_2}$ and $(N + N_{CO_2})/N_\text{tot} = 1 - f_{O_2}$ as well as converting the derivative to a ratio of small but finite differences (represented by $\delta$ in place of $\partial$) gives:

$$\frac{\delta f_{O_2}}{\delta N_{CO_2}} = (\alpha - \alpha \cdot f_{O_2} - f_{O_2}) \cdot \frac{1}{N_\text{tot}} \tag{E3}$$

$$\Rightarrow \delta f_{O_2} = (\alpha - \alpha \cdot f_{O_2} - f_{O_2}) \cdot \frac{\delta N_{CO_2}}{N_\text{tot}} \tag{E4}$$

Finally, to convert $\delta N_{CO_2}/N_\text{tot}$ into terms of $X_{CO_2}$ and $X_{CO_2,\text{ref}}$, we start by defining:

$$X_{CO_2,\text{ref}} = \frac{N_{CO_2}}{N_\text{tot}} \tag{E5}$$

and

$$X_{CO_2} = \frac{N_{CO_2} + \delta N_{CO_2}}{N_\text{tot} + \delta N_{CO_2} + \delta N_{O_2}} \tag{E6}$$

as well as $\delta N_{O_2} = \alpha \cdot \delta N_{CO_2}$. Substituting this and $N_{CO_2} = X_{CO_2,\text{ref}} \cdot N_\text{tot}$ from Eq. (E5) in Eq. (E6) and rearranging gives:

$$\frac{\delta N_{CO_2}}{N_\text{tot}} = \frac{X_{CO_2} - X_{CO_2,\text{ref}}}{1 - X_{CO_2} - \alpha \cdot X_{CO_2}} \tag{E7}$$

Substituting Eq. (E7) in Eq. (E4) yields the final version of Eq. (12).

### E2  $O_2$ dry mole fraction from $O_2/N_2$ data

Measurements of atmospheric $O_2$ concentration are commonly reported as $10^{-6}$ relative deviations in the $O_2/N_2$ ratio (denoted $\delta(O_2/N_2)$ and given in units of per meg) to avoid the complexities of dilution effects from changes in $CO_2$ and other trace species on the $O_2$ dry mole fraction. To convert from available measurements of trends in $\delta(O_2/N_2)$, we must convert to units of ppm and account for the diluting effect of trends in $CO_2$. The equation for the black line in Fig. 18, based on Scripps $\delta(O_2/N_2)$ and NOAA global mean $CO_2$ data, is slightly different from Eq. (12). As above, the derivation starts with Eq. (E1), but now since we have measured values for the change in $N_{O_2}$ and $N_{CO_2}$, our change in $f_{O_2}$ will instead be:

$$\delta f_{O_2} = \frac{\partial f_{O_2}}{\partial N_{O_2}} \cdot \delta N_{O_2} + \frac{\partial f_{O_2}}{\partial N_{CO_2}} \cdot \delta N_{CO_2} \tag{E8}$$

In this case, both $\partial N_{O_2}/\partial N_{CO_2}$ and $\partial N_{CO_2}/\partial N_{O_2}$ are 0, since we have measurements of both $O_2$ and $CO_2$ and therefore can treat their changes as orthogonal. That leads to the following expressions for the derivatives in Eq. (E8):

$$\frac{\partial f_{O_2}}{\partial N_{O_2}} = \frac{1 - f_{O_2,\text{ref}}}{N_{\text{tot}}} \tag{E9}$$

$$\frac{\partial f_{O_2}}{\partial N_{CO_2}} = -\frac{f_{O_2,\text{ref}}}{N_{\text{tot}}} \tag{E10}$$

Inserting these into Eq. (E8) gives:

$$\delta f_{O_2} = (1 - f_{O_2,\text{ref}}) \cdot \frac{\delta N_{O_2}}{N_{\text{tot}}} - f_{O_2,\text{ref}} \cdot \frac{\delta N_{CO_2}}{N_{\text{tot}}} \tag{E11}$$

$\delta N_{O_2}/N_{\text{tot}}$ can be expressed in terms of $\delta(O_2/N_2)$ values by using the definition of $\delta(O_2/N_2)$ (Keeling et al., 1998):

$$\delta(O_2/N_2) = \frac{(O_2/N_2)_{\text{sample}}}{(O_2/N_2)_{\text{reference}}} - 1 \tag{E12}$$

and assuming that the amount of $N_2$ in the atmosphere does not change. Multiplying this definition by $f_{O_2,\text{ref}}$ gives:

$$\delta(O_2/N_2) \cdot f_{O_2,\text{ref}} = \left[ \frac{(N_{O_2} + \delta N_{O_2})/N_{N_2}}{N_{O_2}/N_{N_2}} - 1 \right] \cdot \frac{N_{O_2}}{N_{\text{tot}}} \tag{E13}$$

$$= \frac{\delta N_{O_2}}{N_{\text{tot}}} \tag{E14}$$

$\delta N_{CO_2}/N_{\text{tot}}$ can be expressed as in Eq. (E7) except with $\alpha = 0$ (again, this is because we have measurements of dry mole fractions of $CO_2$ and $O_2$). The final equation used for the "best estimate" line in Fig. 18 is therefore:

$$f_{O_2,\text{ref}} + \delta f_{O_2} = f_{O_2,\text{ref}} + (1 - f_{O_2,\text{ref}}) \cdot \delta(O_2/N_2) \cdot f_{O_2,\text{ref}} - \frac{X_{CO_2} - X_{CO_2,\text{ref}}}{1 - X_{CO_2}} \cdot f_{O_2,\text{ref}} \tag{E15}$$

where $f_{O_2,\text{ref}}$ is the 0.209341 value obtained in §8.3.2 by adjusting Aoki et al. (2019). As noted in §8.3.2, the $\delta(O_2/N_2)$ data used is a weighted average of the ALT, LJO, and CGO sites with weights of $\frac{1}{4}$, $\frac{1}{4}$, and $\frac{1}{2}$, respectively. Note that the NOAA global mean $CO_2$ (rather than TCCON $X_{CO_2}$) is used for $X_{CO_2}$ and $X_{CO_2,\text{ref}}$ in this equation.

*Author contributions.* J.L. Laughner led the development of the new airmass correction (§8.1), window-to-window averaging (§8.2), in situ scaling (§8.3), and miscellaneous changes in §3. G.C. Toon is the main developer of GGG. J. Mendonca developed the non-Voigt treatment of the spectral line shape (§6.2). C. Petri contributed to the development of the phase correction update (§4.2). S. Roche developed the new retrieval grid (§5.1), meteorological resampler (§5.2), and netCDF writer. D. Wunch carried out the sensitivity tests (§9). D. Wunch, C.M. Roehl, G.C. Toon, P.O. Wennberg, and J.L. Laughner conducted the $O_2$ study in §6.3. J.-F. Blavier is a key developer of I2S. D.W.T. Griffith contributed to various aspects of GGG2020 development. P. Heikkinen, R. Kivi, and M.K. Sha first diagnosed the nonlinearity issue from

§4.1 and developed a correction methodology. R.F. Keeling and B.B. Stephens consulted on the approaches to parameterize the change in $O_2$ dry mole fraction (§8.3.2). M. Kiel performed tests of the phase correction threshold (§4.2) and choices of NCBFs (§7) C.M. Roehl, N. Deutscher, P. Jeseck, D. Pollard, M. Rettinger, S. Roche, M.K. Sha, Y. Té and D. Wunch all participated in a beta test of GGG2020. N.M. Deutscher, J. Gross, B. Herkommer, P. Jeseck, I. Morino, H. Ohyama, C. Petri, J. Notholt, D. Pollard, M. Rettinger, S. Roche, E. McGee, K. Strong, C.M. Roehl, M.K. Sha, K. Shiomi, R. Sussmann, Y. Té, V. Velazco, D. Wunch, and M. Zhou provided data used to derive the corrections in §8.1 and §8.2. B.C. Baier, B.B. Stephens, H. Chen, R. Kivi, Y. Choi, X. Lan, T. Laemmel, K. McKain, J. Miller, H. Riris, C. Rousogenous, and S.C. Wofsy provided in situ data used in §8.3. P.O. Wennberg provided input to all elements of this work. All authors reviewed the manuscript.

*Competing interests.* The authors declare no competing interests.

*Acknowledgements.* The authors gratefully acknowledge the use of GNU Parallel (Tange, 2011) in the GGG processing. The authors also thank James Abshire for providing $CO_2$ data used in deriving the in situ correction (§8.3). A portion of this research was carried out at the Jet Propulsion Laboratory (JPL), California Institute of Technology, under a contract with NASA (80NM0018D0004). Government sponsorship is acknowledged. Support for Caltech TCCON sites and partial support for JLL, MK, CMR, and POW provided by NASA grants NNX17AE15G and 80NSSC22K1066. Material from BBS and RFK is based upon work supported by the National Center for Atmospheric Research, which is a major facility sponsored by the National Science Foundation under Cooperative Agreement No. 1852977.

MR and RS acknowledge funding by the German Helmholtz Research Program "Changing Earth – Sustaining our Future" within the Research Field "Earth and Environment". The Paris TCCON site has received funding from Sorbonne Université, the French research center CNRS and the French space agency CNES. The Cyprus TCCON site and AirCore flights have received funding from the European Union's Horizon 2020 research and innovation programme under grant agreement No. 856612 and the Cyprus Government. The TCCON site at Réunion Island has been operated by the Royal Belgian Institute for Space Aeronomy with financial support since 2014 by the EU project ICOS-INIWRE (Grant agreement ID: 313169), the ministerial decree for ICOS (FR/35/IC1 to FR/35/C6), ESFRI-FED ICOS-BE project (EF/211/ICOS-BE) and local activities supported by LACy/UMR8105 and by OSU-R/UMS3365 – Université de La Réunion. The Eureka TCCON measurements were made at the Polar Environment Atmospheric Research Laboratory (PEARL) by the Canadian Network for the Detection of Atmospheric Change (CANDAC), primarily supported by the Natural Sciences and Engineering Research Council of Canada, Environment and Climate Change Canada, and the Canadian Space Agency. TCCON sites at Tsukuba, Rikubetsu and Burgos are supported in part by the GOSAT series project. Burgos is supported in part by the Energy Development Corporation Philippines.

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

**Table C3.** Ground in situ data used in validating the priors. "CO$_2$ Obspack" is the CO$_2$ GLOBALVIEWplus v5.0 ObsPack (Cooperative Global Atmospheric Data Integration Project, 2019) and "CH$_4$ ObsPack" the CH$_4$ GLOBALVIEWplus v2.0 ObsPack (Cooperative Global Atmospheric Data Integration Project, 2020). The "TCCON sites" column indicates which sites profile were used at, the IDs are mapped to locations in Table 2. In the "Providers" column, affiliations are given in parentheses. If only one affiliation is listed, it applies to all individuals named. Abbreviations: NDIR = Nondispersive infrared; NOAA GML = National Oceanic and Atmospheric Administration Global Monitoring Laboratory; PSU = Pennsylvania State University; U. of WI = University of Wisconsin; USGS = United States Geological Survey; LBNL = Lawrence Berkeley National Laboratory; ARM = Atmospheric Radiation Measurement; CRDS = cavity ring-down spectroscopy; NIWA = National Institute of Water & Atmospheric Research Ltd.

| Source | Measurement type | Providers/partners | Location | TCCON site |
|---|---|---|---|---|
| CO$_2$ ObsPack | Programmable flask packages | Arlyn Andrews (NOAA GML), Peter Bakwin, Ken Davis (PSU), Ankur Desai (U. of WI), & Dan Baumann (USGS) | Park Falls, WI, USA | pa |
| CO$_2$ ObsPack | Li-cor NDIR on tower | Arlyn Andrews (NOAA GML), Ed Dlugokencky (NOAA GML), Ken Davis (PSU), Ankur Desai (U. of WI), & Dan Baumann (USGS) | Park Falls, WI, USA | pa |
| CO$_2$ ObsPack | CRDS on tower | Sebastien Biraud & Margaret Torn (LBNL) | Southern Great Plains ARM site, OK, USA | oc |
| CH$_4$ ObsPack | Programmable flask packages | Arlyn Andrews (NOAA GML), Ed Dlugokencky (NOAA GML), Ankur Desai (U. of WI), & Dan Baumann (USGS) | Park Falls, WI, USA | pa |
| CH$_4$ ObsPack | CRDS on tower | Arlyn Andrews (NOAA GML), Ankur Desai (U. of WI), & Dan Baumann (USGS) | Park Falls, WI, USA | pa |
| CH$_4$ ObsPack | Flask | Ed Dlugokencky (NOAA GML), Sebastien Biraud (LBNL), & Margaret Torn (LBNL) | Southern Great Plains ARM site, OK, USA | oc |
| CH$_4$ ObsPack | CRDS on tower | Sebastien Biraud & Margaret Torn (LBNL) | Southern Great Plains ARM site, OK, USA | oc |
| NIWA (direct) | Licor 7000 NDIR (CO$_2$), in situ GHG FTS (CH$_4$) | Dan Smale (NIWA) | Lauder, New Zealand | ll |

**Table C4.** The number of profiles in the $CO_2$ in situ correction from each campaign or other data source identified and used for each TCCON site. The "Found" column gives the number of profiles identified for that campaign & site, the "Used" column gives the number of those profiles which could be used in the in situ comparison after matching with TCCON data. The definitions of the site IDs can be found in Table 2; "we" refers to an instrument in Jena, Germany for which GGG2020 data is not available at time of writing.

| Campaign | Site | Found | Used | Campaign | Site | Found | Used |
|---|---|---|---|---|---|---|---|
| ATom | ae | 4 | 0 | INTEX-NA | pa | 3 | 3 |
| | df | 1 | 1 | KORUS-AQ | an | 1 | 1 |
| | eu | 2 | 0 | | df | 1 | 1 |
| | ll | 4 | 4 | | rj | 2 | 2 |
| | oc | 1 | 0 | ORCAS | oc | 1 | 1 |
| | pa | 1 | 1 | SEAC4RS | df | 1 | 1 |
| COB2004 | pa | 5 | 4 | | oc | 2 | 0 |
| DC3 | oc | 3 | 2 | START-08 | pa | 2 | 0 |
| GO-Amazon | ma | 2 | 1 | AirCore | df | 3 | 3 |
| GSFC | df | 8 | 7 | | ni | 3 | 2 |
| | pa | 2 | 2 | | oc | 19 | 13 |
| HIPPO | ll | 7 | 5 | | pa | 2 | 2 |
| | wg | 1 | 0 | | so | 16 | 9 |
| IMECC | bi | 2 | 2 | | | | |
| | br | 2 | 0 | | | | |
| | gm | 1 | 1 | | | | |
| | je | 1 | 0 | | | | |
| | ka | 1 | 0 | | | | |
| | or | 2 | 0 | | | | |

**Table C5.** Same as Table C4 but for the $CH_4$ in situ correction.

| Campaign | Site | Found | Used | Campaign | Site | Found | Used |
|---|---|---|---|---|---|---|---|
| ATom | ae | 4 | 0 | IMECC | bi | 2 | 2 |
| | ci | 2 | 1 | | br | 2 | 0 |
| | df | 1 | 1 | | gm | 1 | 0 |
| | eu | 1 | 0 | | je | 1 | 0 |
| | ll | 1 | 1 | | ka | 1 | 0 |
| | oc | 1 | 0 | | or | 2 | 0 |
| | pa | 1 | 1 | START-08 | pa | 2 | 1 |
| HIPPO | ll | 5 | 3 | AirCore | df | 3 | 3 |
| | oc | 4 | 1 | | ni | 3 | 2 |
| | pa | 1 | 0 | | oc | 19 | 13 |
| | wg | 1 | 0 | | pa | 2 | 2 |
| | | | | | so | 16 | 9 |

**Table C6.** Values of $\partial r / \partial X_{\mathrm{luft}}$ in Eq. (C11). Gases not listed here use 0 for $\partial r / \partial X_{\mathrm{luft}}$.

| Gas | $\partial r / \partial X_{\mathrm{luft}}$ |
|---|---|
| $CO_2$ | 0.363 |
| $wCO_2$ | 0.206 |
| $lCO_2$ | 0.928 |
| $CH_4$ | 0.0609 |

| Gas | $\epsilon_{\mathrm{meas}}$ | $\epsilon_{\mathrm{FT}}$ | $\epsilon_{\mathrm{strat}}$ | $\epsilon_{\mathrm{std.\ xgas}}$ | $\epsilon_{X_{\mathrm{luft}}}$ |
|---|---|---|---|---|---|
| $X_{CO_2}$ (ppm) | 0.033 (0.16) | 0.032 (0.12) | 0.061 (0.072) | 0.13 (0.97) | 0.03 (0.22) |
| $X_{wCO_2}$ (ppm) | 0.037 (0.16) | 0.038 (0.15) | 0.075 (0.10) | 0.23 (1.6) | 0.017 (0.12) |
| $X_{lCO_2}$ (ppm) | 0.025 (0.14) | 0.020 (0.067) | 0.057 (0.060) | 0.16 (1.18) | 0.08 (0.56) |
| $X_{CH_4}$ (ppb) | 0.65 (3.1) | 0.19 (0.49) | 3.4 (6.3) | 0.86 (5.0) | 0.07 (0.15) |
| $X_{CO}$ (ppb) | 1.9 (9.3) | 0.13 (0.39) | 0.24 (4.8) | 0.44 (2.0) | 0 (0) |
| $X_{H_2O}$ (ppm) | 100 (950) | 0 (0) | 0 (0) | 26 (200) | 0 (0) |

**Table C7.** The magnitudes of each uncertainty component for the AICF comparison. As in Table 5, the first number in each column is the overall contribution of that term to the AICF according to formal error propagation, and the number in parentheses is the simple mean across all the TCCON to in situ comparisons. The units for each gas's error values are given in the first column.