# Peer review of "The Total Carbon Column Observing Network's GGG2020 Data Version"

_Earth System Science Data, 2023_

## Author Comment (AC1)

**The Total Carbon Column Observing Network's GGG2020 Data Version**

**Response to Referee #1 (Denis Jouglet)**

Joshua L. Laughner on behalf of all coauthors

February 26, 2024

Thank you to Dr. Jouglet for taking the time to thoroughly read our manuscript and provide detailed feedback. We have considered each comment; in most cases we have added material to address any noted confusion or ambiguity. In some cases, we elected not to do so to maintain the flow of the manuscript. Below, comments from Dr. Jouglet are in red, our responses are in blue, and quotes of the manuscript are in black. Unless otherwise stated, line, figure, table, and section numbers used in comments and responses refer to the *original* manuscript while such cross-references in quotes of the new version of the paper refer to the *revised* version.

This paper is not self-sufficient, since many rationales can only be found in previous papers: Wunch et al. 2010, 2011 and 2015. I think that scientists not familiar with the data cannot understand this paper on its own. I had to go back and forth to these papers. This would at least require systematic references to these papers (probably including the section, figure or table number), and in most cases a quick reminder (for example in a more detailed introduction).

We have tried to clarify such points as much as possible based on specific comments from all three reviewers.

Some assumptions made in these previous works are not fully described in these papers. In some cases, I was even not able to find the answer (see examples below).

We have done our best to address any specific cases noted by the reviewers.

The paper is very long. I think that all parts deserve publication, but it could be split or rearranged if possible (mostly the uncertainty budget).

We agree that it is very long, but our goal is that this be a reference for GGG2020, so we prefer to keep all piece together.

**Section 1**

l 40: did you think of permanently removing channels in the center lines that become saturated at large SZA, so as to homogenize the biases along the SZA?

No, we do not expect this would give better results. This would give up information available from those line cores at smaller SZAs, which covers the most common times for satellites to observe.

l 48 : the assumption of "consistent across sites" seems to be in contradiction with Wunch et al 2011 appendix A.a (0.2ppm). Even if instrument are perfectly consistent, their different environments (boreal vs tropics) could translate in apparent inconsistencies.

The in situ correction is aimed only at removing systematic biases, which we expect to arise primarily from the spectroscopic parameters used in the retrieval and thus will be consistent across sites by construction. Instrumental biases from e.g. imperfect ILS would change over time, and so cannot be accounted for with the static in situ correction. We have edited this line to include:

> "However, since all TCCON sites use the same retrieval (and thus the same forward model), we use a single mean scaling factor to remove the mean bias caused by errors in the spectroscopic parameters. **It is not intended to correct biases from instrument artifacts, such as an imperfect instrument line shape (ILS), as such biases can change over time.** The scaling factors for the various gases..."

l 58: be careful that the refraction effect may differ in the O2 and the other gas windows.

Yes, but more so for instruments that use the A-band, which is farther from the CO2 bands than the singlet-delta O2 band TCCON uses, therefore path length effects will be smaller for TCCON than most satellite missions. Also, this effect will not introduce a significantly different path length for the direct-sun geometry TCCON measures in. We have added:

> "Because $O_2$ and the primary TCCON gases are measured on the same detector, many biases related to the detector and pointing will be cancelled out (Wunch et al., 2011, Appendices A and B). **Note that TCCON uses the $^1\Delta$ $O_2$ band around 7885 cm$^{-1}$, rather than the A-band (around 13080 cm$^{-1}$, commonly used by satellite missions to avoid interference from airglow). The $^1\Delta$ $O_2$ band is closer in frequency to the near-IR $CO_2$ and $CH_4$ bands than the $O_2$ A-band; this minimizes differences frequency-dependent effects (e.g. scattering) between the $O_2$ and $CO_2$ or $CH_4$ bands.**"

**Section 2**

l 85 : I have a question a bit beyond the scope of this paper, but I did not get the answer in previous papers. What is exactly your definition of AK? In a bayesian framework, AK are usually defined as information coming from the measurement with respect to the prior. According to Wunch et al 2010 the retrieval is least square fitting of a scaling factor of an a priori profile, without an explicit value of any prior of this scaling factor and associated uncertainty.

We have added references to another source which describes these differences in more detail in the introduction:

> "GGG retrieves trace gas column amounts by iteratively scaling an a priori vertical trace gas profile until the best fit between a spectrum simulated from those trace gas profiles by the built-in forward model and the observed spectrum is found. **(This differs slightly from the Bayesian framework described in Rodgers (2000). Please refer to section 3.4 of Roche (2021) for a discussion of specific differences.)**"

**Section 3**

Large works have been done for improving spectroscopy, but no estimation of the gains on Xgas accuracy is given. Following section 7.1, they seem not to prevent the ACDF empirical correction. Can you provide an estimation?

We are in the process of carrying out a study to attribute the differences between GGG2014 and GGG2020 to specific improvements between the two versions. This question is better left to that study.

section 3.3: The O2 column is estimated from the 1.27µm band (always this band?). The O2 spectroscopic parameters are optimized so that Xluft, which is the ratio of O2 column from spectroscopy to O2 column from local pressure measurement, is close to 1 with low variance. Only the O2 spectroscopic parameters are tuned, which means that the surface pressure measurement is assumed to be the truth. Therefore, why using the O2 band and not directly the surface pressure measurement for the O2 column estimation?

As mentioned in the introduction (immediately following Eq 1), because the O2 window is measured on the same detector as the GHG species, certain types of instrumental error will cancel out between the columns. We have clarified this in the second bullet:

> "It normalizes for path length. Observations at **higher** surface elevations will have smaller column densities compared to those from lower altitudes, due to the shorter vertical extent. Normalizing to the $O_2$ column removes this effect.
>
> Because $O_2$ and the primary TCCON gases are measured on the same detector, many biases related to the detector and pointing partially cancelled out (**e.g. ILS, mis-pointing, zero-level offsets,** Wunch et al., 2011, Appendices A and B). Note that TCCON uses..."

section 3.3: you choose to change some spectroscopic coefficients from your empirical observations. This is fully understandable. Did you have any discussion about that with spectrocopists? Are your changes inside the uncertainties given by spectroscopists?

Yes, we discussed with spectroscopists and chose the range to vary the coefficients for temperature- and pressure- broadening to fall within the range of uncertainties for those parameters in the spectroscopic literature. We have noted this in the revised text:

> "In each test, we scaled the temperature dependence, pressure dependence, or both of all lines in the $O_2$ band, **covering a reasonable range of estimates from the literature.**"

l.173: You choose to use only one value, T700, but the temperature vertical profile may be heterogeneous, in particular in the boundary layer. Do you not think that this approximation could be source of error?

We chose T700 specifically to avoid variations near the surface, as we expected that those variations would be limited to the bottom    1 km ( 10%) of the total column, and less correlated with synoptic scale changes affecting the free troposphere ( 60% to 70% of the column). The challenge is that, as we only retrieve a single piece of vertical information, we need a temperature metric related to the majority of the column. We are beginning a new study to investigate spectroscopic biases in other gases, and plan to revisit O2 in this study as well. We have included our rationale for using T700 in the revised manuscript:

> "T700 is taken from the a priori meteorology data, **and was chosen on the assumption that this is a reasonable metric for temperature variations in the free troposphere containing the majority ($\sim$ 60%, 800 to 200 hPa) of the $O_2$ column.**"

fig 2: In panel (b) a slope can still be observed.

Correct, we were not able completely removed the temperature dependence, only reduced it in magnitude. The last sentence of sect. 3.3 stated this; we have made it clearer that the slope is not zero:

> "The effect on the $X_{\text{luft}}$ vs. T700 relationship is shown in Fig. 7b, **where although not reduced to zero**, the slope is reduced by a factor of 4 compared to its pre-optimization value."

**Section 4**

Do you not think that cross-sections computed for interpolated met profile for each hour of a day would provide an improvement in the retrieval?

Perhaps. However, as GGG is designed, this would require recalculating absorption coefficients even more frequently, so we chose to balance the computation speed and time resolution of the met data.

**Section 5**

l.242: Does detector saturation often happen? Why do you not adjust the gains to the maximum possible intensity of the place? This is quite deterministic, depending mostly on AOD (not strongly on SZAs in the SWIR when AOD is low and SZA not at extreme values).

As stated in Sect. 5, TCCON sites already have adjusted pre-amplifier gains and beam intensity to avoid detector saturation, so large detector saturations are typically rare, and normally found and fixed shortly after installation. The flag is implemented both to ensure that these rare saturation events do not get processed, and to help PIs identify the problem quickly. We have added a sentence indicating that the large saturation effects are rare:

> "We have implemented a check early in i2s processing to remove interferograms affected by detector or signal chain saturation, an extreme form of detector non-linearity... We call this "detector saturation" and this causes irreversible detector nonlinearity in which spectral information is permanently lost. **Detector saturation is rarely found in the TCCON spectra, and is straightforward to resolve once it is identified. We do this in two ways: (1)** detectors used for TCCON measurements have reduced pre-amplifier gain settings **and (2)** we limit the number of photons incident on the detector...""

l.243: In such saturation cases, only the very low frequencies are lost. These frequencies are mostly retrieved by the continuum fitting and should not bring information on gases. Do you think that gas information is lost in such configuration?

Detector saturation is an irreversible problem (unlike small detector nonlinearities, discussed later in the manuscript), and the retrievals from spectra affected by detector saturation produce nonsensical results. Detector nonlinearity, if small, is reversible. However, we do expect nonlinearity to affect the retrieved gas columns. The nonlinearity causes a zero level offset, which alters the line depths by lowering the continuum. Sha et al. (2020) identified an example of how the subtle nonlinearity at Sodankyla introduced a bias in the XCO2. We have added a sentence between the paragraph on detector saturation and detector nonlinearity to make clear which type of nonlinearity we are referring to:

> "**There are more subtle detector nonlinearity effects that do not result in detector saturation, but can adversely affect the retrievals.** We now compute and store a detector nonlinearity diagnostic variable ("DIP")..."

**Section 6**

Is there no interference between the instrument continuum fitting and the O2 1.27μm CIA when using polynomial orders greater than 2? The CIA brings much information on O2 amount.

Yes, there is some interference. But the shape of the CIA absorption is fixed by sppec-troscopy, whereas the shape of the continuum is fitted. We also point out that the O2 CIA absorption contribution is fitted separately from the contribution from the discrete O2 lines. The O2 retrieved from the CIA is not used, due to the possibility of interference with continuum.

**Section 7**

Section 7.1: can you confirm that the new ADCF are computed on the Xgas benefiting from the O2 spectroscopic improvements of section 3.3?

    We have added:

> "The second is an in situ-based, or airmass-independent correction (§8.3), which aims to eliminate the mean bias in $X_{gas}$ values arising from incorrect spectroscopic line strengths. **These corrections are calculated from data that includes all improvements discussed in the preceding sections.**"

    l.395: you consider the temperature dependence of the ADCF as "spurious". I think this a hypothesis (like symmetry of Xgas with respect to noon in equation (5)) that requires to be identified as is. Can natural phenomena not also be responsible for such trends? Probably seeing this effect on some but not all windows of a trace gas could enforce the spurious hypothesis.
    We acknowledge that this could be a real change in the atmosphere, though it is highly unlikely because of the differences between windows (as you said) and because the ADCF is constructed to focus on changes in Xgas that are symmetric about solar noon, which natural processes are unlikely to match. We have added this to the end of Sect. 7.1:

> "The final step in selecting ADCFs for GGG2020 was to account for **potentially** spurious temperature dependence in the $X_{gas}$ values. As we saw with $O_2$ in §6.3, incorrect temperature dependence in the line widths introduces a temperature dependence in retrieved $X_{gas}$, which could alias into the airmass dependence. **While we acknowledge that such temperature dependence of the ADCFs could be due to a real change in the atmosphere, we believe this to be unlikely for two reasons. First, the ADCF is constructed to account only for variations in Xgas that are symmetric around solar noon, and generally changes in atmospheric composition are not perfectly symmetric around solar noon. Second, as we show in Fig. 13, different windows for the same gas have different relationships between the ADCF and temperature. A real change in atmospheric composition would be more likely to show up in all windows for a given gas.**"

    In section 3.3 you update the O2 spectroscopic coefficients, using O2 from the pressure sensor as the truth to fit. Here in section 7.1, why do you not do the same optimization exercise for the spectroscopic coefficients of trace gases, rather than a posteriori empirical correction? This would require an external truth which could be the in situ measurements of section 7.3. L.406 mentions this plan for temperature dependence but could be enlarged to SZA dependence.
    We have left the optimization of the CO2, CH4, etc. spectroscopic coefficients for future work because we did not identify a truth metric for the optimization until after GGG2020 had

entered final testing. We will investigate whether optimizing the spectroscopic coefficients using a similar approach as we did for O2, but using the slopes of ADCF versus temperature as our quality metric, can reduce the SZA- and temperature- dependent biases. While the in situ data used later in the paper might be a possible truth metric, we would need to evaluate whether there are enough in situ profiles to reliably correct the spectroscopic errors leading to SZA dependence.

Section 7.1.1: you remove the windows that are the more affected by the temperature dependence. What is your threshold for the decision of correcting or removing?

We did not have a predetermined threshold; our rationale are explained in the text of Sect. 7.1.1 (now §8.1.1). For the wCO2 and CO windows, one window had a clearly larger temperature dependence than the other, and we had reason to suspect stronger water interference in the windows with larger temperature dependence. For the HCl windows, we also tried to identify windows with strong water interference, but as stated in the text, removed two additional windows as special cases.

Section 7.3: I would like to understand the differing results of GGG2014 and GGG2020. In l.455, what are the differences between the in situ dataset used for GGG2014 and that for GGG2020? Do you expect changes / improvements in your AICF estimation with the GGG2020 dataset? Larger variety of in situ instrument to vary the potential biases; larger range in weather conditions to disangle trends?

For GGG2020, we reconstructed our suite of in situ profiles from NOAA ObsPack data, whereas previously we had used individual campaigns' data files. This has several benefits, but means that a profile-by-profile comparison between GGG2014 and GGG2020 is not simple to describe. We do expect improvements to the AICF with the new dataset, primarily from updates to the spectroscopic parameters. However, the primary purpose of the in situ correction remains to tie remotely sensed GHG data to the same measurement scales as the in situ data. We added a paragraph to Sect. 7.3 (now §8.3) explaining this:

> "The use of ObsPack data represents a slight methodological change compared to GGG2014. Most of the in situ aircraft profiles used for the GGG2014 in situ correction are included in the ObsPack, and switching to the ObsPack instead of individual campaigns' data files will allow us to use the same tools to ingest future new profiles added to the ObsPack. This also allows us to benefit from the data curation and quality control efforts of the ObsPack team. With the larger number of profiles now available (especially for $CO_2$), we are able to test for correlations with potential sources or metrics of bias. However, the primary purpose of the in situ comparison remains to tie TCCON (and through TCCON, satellite) GHG data to the same metrological scales as in situ GHG data."

What is the size of the GGG2020 in situ data set? And thus the number of elements in fig 11? OK this is answered in table 2, but should be given in plain text.

Since each gas has a different number of profiles, we prefer to keep this information in tabular form. We have added a reference to table to in the now second to last paragraph before sect. 7.3.1:

> "Due to the relative sparsity of $N_2O$ profiles, GGG2020 TCCON $N_2O$ products were evaluated against surface $N_2O$ data and a different approach, which will be covered in §8.3.3. **The number of usable profiles for each gas is given in Table 2.**"

The appendix C6 is very important for this paper. It would be useful to give orders of magnitudes of the several sources of uncertainties (partially given in table 3).
We have moved the individual error terms from Table 3 to a new table in appendix C and added the two TCCON-related error terms.

Appendix C6 l.1213: Why do you take twice the std and not the std itself?
This to be conservative about the uncertainty from random terms in the overall error. The exact multiple should not matter substantially, as we use two sigma consistently in all parts of the uncertainty calculation. We have added this to the first paragraph of appendix C6:

> "For the TCCON/in situ ratios in §8.3, we considered five sources of uncertainty for the comparisons. **We chose twice the standard deviation as our metric for deriving uncertainty (rather than $1\sigma$ to be conservative), and use that consistently for all random error terms.**"

l.518: I agree the presentation of fig11 is better than the older presentation (fig5 in Wunch et al 2010, fig.8 in Wunch et al. 2015). Please precise that the in situ ratio of the fig11 is equivalent to the inverse of the slope of the best fit of older papers.
The ratios from Fig 11. are *not* the inverse of the slopes from Wunch et al. 2010 and 2015, both the slopes and the ratios are TCCON/in situ. We have added a sentence in the paragraph following the enumerated uncertainty sources:

> "The use of TCCON to in situ ratios to derive the in situ correction is equivalent to the best fit lines forced through the origin used in Wunch et al. (2010), as the best fit line through the origin is essentially the mean TCCON to in situ ratio. **As in Wunch et al. (2010), a ratio (or slope in Wunch et al. (2010)) $> 1$ indicates TCCON $X_{gas}$ values are biased high relative to in situ, and vice versa for ratios $< 1$.**"

l.524: Here for CO2 the ratio is 1.01%, whereas in Wunch et al. 2010 fig5 and in Wunch et al. 2015 fig8 it was 1.1% (1/0.989). Can we conclude that the updates of the GFIT processing and the new ADCF described in this paper have provided such a 10% improvement? If so, please emphasize it. If not, please explain why (the values are comparable since the data set are different).
As noted above, the ratios and slopes are directly comparable, so in Wunch et al. 2015, the retrieved pre-in situ correction XCO2 was 1% low compared to in situ, and here it is about 1% high. We would not expect the ADCF to change the mean this significantly; however full attribution will be done in an ongoing study. We have added a new paragraph in Sect. 7.3.1:

"Additionally, we note the TCCON $X_{CO_2}$ product changed from being 1% low compared to in situ (pre-in situ correction) in Wunch et al. (2015) to 1% high here. We would expect this to be due to changes in spectroscopy, such as an average decrease in $CO_2$ line strengths or increase in $O_2$ line strengths. However, we are in the process of conducting a full attribution study for all the component changes between GGG2014 and GGG2020, and reserve a final conclusion until that is complete."

l.627: I think "small" is a bit under-evaluated, since for 420ppm the order of magnitude is the same as the one of the new XCO2 scale.

Since both changes are of order 0.1 to 0.2 ppm, "small" seems appropriate. However, we have modified this sentence to read:

"For $X_{CO_2}$ values around 400 ppm, **the change is of similar magnitude to the WMO scale change** for $CO_2$ products."

As already mentioned in my "general comments", this section is very interesting in terms of metrology. I would interpret section 7.1 as a correction of intrinsic quality of the detector (removing all artifacts regardless the conditions), and section 7.3 as the absolute calibration of the sensor (more precisely fit to WMO standard). In previous papers, only airmass dependencies were corrected. But now we can see that the list of potential dependencies for intrinsic quality is larger: ADCF in section 7.1, atmospheric temperature dependency in section 7.1, impact of Xgas. [the classical Xgas(TCCON) = f(Xgas(in situ)) (like fig.8 in Wunch et al. 2015) have been discarded in this paper (section 7.3), this implicitly mean that TCCON is linear with Xgas], impact of Xluft as seen by section 7.3 fig.11 (l.555 mentions that it will be a future update). Maybe the paper should be more explicit about this "metrological" process. As a consequence, and depending on the size of the section 7.3 data set, I think other dependencies could be looked for (humidity, AOD, altitude, etc.).

To be clear, the post-hoc corrections are still limited to the airmass dependent correction and the scaling to match in situ data. The only temperature corrections were done by updating the O2 spectroscopic parameters, so we consider that an empirical adjustment to the forward model physics rather than a post hoc correction. Additionally, we are now identifying new sources of uncertainty not because they did not exist before, but because we have reduced other sources of uncertainty that allow us to identify these new ones and because the need for greater precision in remotely sensed GHG products from the scientific community is driving us to identify ever smaller errors. I am not clear why is meant by "Xgas(TCCON) = f(Xgas(in situ))" and "TCCON is linear with Xgas" in the third bullet point. In Wunch et al. 2015, the relationship between in situ and TCCON Xgas values was a linear fit with the intercept forced through 0, which is equivalent to the mean ratio of TCCON/in situ we use here. Both Wunch et al. (2015) and this work assume that bias between TCCON Xgas and in situ Xgas values will be multiplicative because of the nature of biases in retrieved columns arising from errors in the average spectroscopic line strengths, thus a ratio of TCCON/in situ values is the correct way to calculate and correct those biases. We are also working with a colleague at the National Institute of Standards and Technology

who is very interested in improving the metrology of remote sensing GHG products. As the NIST group develops the appropriate framework for such metrology, we are happy to participate in future efforts to more rigorously document the chain of references. Regarding other dependencies, we have looked for other correlates with the TCCON/in situ bias, but have not found strong evidence for drivers other than SZA and temperature. This does not mean such drivers are absent, only that the in situ dataset is not sufficient yet to identify them. We have added a sentence to the beginning of Sect. 7.3:

> "As in GGG2014, the GGG2020 $X_{CO_2}$, $X_{CH_4}$, $X_{N_2O}$, and $X_{H_2O}$ products are tied to standard scales by in situ aircraft, balloon, and/or radiosonde measurements to remove any mean multiplicative bias introduced by error in absorption line intensity. **As the absorption of a gas is the product of its column density and spectroscopic cross section, a bias in the mean line intensity (and therefore the cross section) will by definition lead to a multiplicative bias in the simulated absorption and thus the retrieved column density.**"

**Section 8**

l.758: it is written that surface pressure measurement is used for calculating the total column of air, whereas previously it was said that the O2 absorption band is used for that purpose, this is contradiction.

We have revised this to:

> "The surface pressure measurements we collect as part of our on-site meteorological data are important for **determining the bottom altitude when integrating the total columns.**"

l.784: do you not think that after the works done in section 7.1, it could be possible to add the spectroscopic errors as an uncertainty source?

No, spectroscopic errors will usually manifest as a static bias or one covarying with atmospheric parameters, rather than a random uncertainty source. We remove these errors to the best of our ability with the airmass dependent correction and in situ correction, and spectroscopic uncertainies are considered in the comparison between TCCON and in situ data at the end of section 8. These uncertainties will fall out during those corrections. We have noted this in a new paragraph at the end of the list of error terms considered in Sect. 8, subsection "Other sources of error":

> "This error budget does not include radiometric noise or spectroscopic errors. We omit radiometric noise because Wunch et al. (2011) showed that random noise does not introduce a bias in $X_{CO_2}$ because TCCON spectra have a high signal-to-noise ratio due to the direct-sun viewing geometry and the strength of our target gases' absorption lines. **We omit spectroscopic errors in this section because mean and SZA-dependent spectroscopic errors are removed by the post processing corrections (§8.1, §8.3).**"

l.784: It would be interesting to get an inter-instrument budget beside the single instrument budget. This would be very useful since one of the use of the network is to analysis spatial gradients. For example, error in the retrieval like the choice of the prior will be partially common to all instruments (partially because it may depend on latitude). Some errors (pointing error, FOV error) will be different from an instrument to another.

The challenge is that such an analysis either needs a transfer standard or some external assumptions (e.g. models) to distinguish between inter-site differences due to real differences in atmospheric composition and those due to differences in the instruments. This is something being explored through the use of an EM27 as a travel standard.

l.868: the classical way would be to use the standard deviation, please explain why you use the median absolute deviation here (robustness to outliers?)

Yes, here we prefer MAD because it is less sensitive to outliers and we need a robust estimate of the TCCON vs. in situ difference, and do not need the properties of Gaussian error statistics. Also, this was "mean absolute deviation", not "median". We have corrected that typo and clarified the rationale for choosing MAD as follows:

> "For the in situ uncertainty, we use the median absolute deviation (MAD) of the TCCON $X_{\mathrm{gas}}$ values from the in situ $X_{\mathrm{gas}}$ values after removing the mean bias for each $X_{\mathrm{gas}}$ (i.e. the correction factor in Table 4). **We use MAD over standard deviation because it is less sensitive to outliers.**"

l.878: In your sensitivity study (first part of section 8), you did not include the radiometric noise, which would be the main random error source. Most sources you considered should be quite constant over a day, so the assumption of reduce sources of random error sounds good to me. Be careful however that some sources of your sensitivity study could be slowly variable and therefore mostly seen in the mean bias, not in the median absolute deviation.

We did not include radiometric noise because Wunch et al. (2011) had previously shown that random noise had little effect on XCO2. We have noted this in a new paragraph at the end of the list of error terms considered in Sect. 8, "Other sources of error":

> "**This error budget does not include radiometric noise or spectroscopic errors. We omit radiometric noise because Wunch et al. (2011) showed that random noise does not introduce a bias in $X_{\mathrm{CO_2}}$ because TCCON spectra have a high signal-to-noise ratio due to the direct-sun viewing geometry and the strength of our target gases' absorption lines.** We omit spectroscopic errors in this section because mean and SZA-dependent spectroscopic errors are removed by the post processing corrections (§8.1, §8.3).""

table 3 and l.883: In "Mean abs. dev." there is the contribution of the instrument, of the in situ measurement and of the comparison between both. Do you not think you should compare "Mean abs. dev." with the quadratic summation of "Error budget" and "Epsilon_insitu", rather than "Mean abs. dev." with "error budget"?

Yes, that is a good point. We have updated the table (now Table 5) to include the quadrature sum and modified the text to use that comparison instead.

**Clarity**

**Sect. 1**

This introduction should be expanded, and divided into several sub-sections: (1) It should recall the main uses of the TCCON network (as given in the abstract). (2) It should explain that the scope of this paper is to describe the major changes from GGG2014 to GGG2020, justify why so hard work has undertaken. The expected accuracy for GGG2020 and the current performance of GGG2014 would be the best rationale. (3) Introduction should also reference to Wunch et al 2011 and 2010, since the major parts of the algorithm are described in the 2011 paper. (4) I think the complete window definition should be recalled in a table (or at least referenced) in introduction or in appendix. This will ease the comprehension of section 2 by newcomers (useful also for l126), and also give the current status. (5) Introduction should recall the main steps of the retrieval (Bayesian approach or not?), including the cross-sections computation and the AK definition, so as to make the paper more self-consistent. (6) L 75: maybe the introduction should also mention the systematic quality check done by the central facility? Does it include the filterings listed in section 7.3?

We have implemented some of these suggestions.

1. Common applications of TCCON data were already given in the first paragraph of the introduction

2. We have added a sentence stating the purpose of this paper to the 5th paragraph of the revised introduction, and a new second paragraph describing the motivation for the update.

3. We have added a "see also" reference to Wunch et al. 2010 and 2011 in the 5th paragraph of the revised introduction.

4. The windows have been added as Tables A2 and A3 in the appendices.

5. We now state that the approach in GGG differs slightly from the standard Bayesian approach in the 6th paragraph and provide a reference with more details. The AK computation is not done as part of typical GGG processing (they are precomputed), so we have not included that here.

6. We now state "This data undergoes quality evaluation before release, with all data reviewed by experienced TCCON members from various sites." in the second to last paragraph of the revised introduction. Sect. 7.3 only pertains to the in situ correction; any filtering described there is separate from the standard QC process.

Description of the new merge of several windows of line 30 is redundant with section 7.2 and therefore could not be mentioned in introduction. I was not able to find the way several windows were merged in previous papers.

We prefer to keep this in the introduction as well, since it is part of the data processing sequence being laid out there.

It is not clear in sections 1, 7.1 and 7.2 whether the ADCF correction and the window merging is performed on column densities or on column average dry mole fraction (l.52). Please clarify.

We have changed an earlier sentence to make the order of operations clear:

> "The post processing step includes **the conversion from column densities to column-average mole fractions, followed by** the above window-to-window averaging, an empirical airmass-dependent correction, and a scaling correction to tie TCCON data to the relevant calibration scales."

Please precise that Vgas and VO2 are column densities, and give the physical unit.
We have added the units of molec/cm2

l 45: can you give a reference for the 0.25%?
We've removed the 0.25% here because we are discussing all gases generally, which have different requirements. We have added a paragraph earlier in the introduction that includes citations for the CO2 requirements.

The "scaling factor" or "scaling correction" could already be named AICF, and the "empirical airmass-dependent correction" ADCF.
We prefer to introduce the concepts without the later jargon here.

**Sect. 2**

People knowing the CO2 spectroscopy could wonder why the weak window at 6536cm-1 is not mentioned, maybe you can refer to section 7.1.1 l.415 which brings the explanation. Same for the 4905cm-1 strong CO2 band.

We have added a short paragraph to Sect. 2 with this information:

> "For $wCO_2$, we chose not to use the second weak band around 6500 $cm^{-1}$ for reasons detailed in §8.1. For $lCO_2$, we did not use the strong band around 4900 $cm^{-1}$ because the lines are so strong that the retrieval would be more sensitive to errors in the line shape and zero level offsets in the interferograms."

To what the "l" of "lCO2" refers to? In the OCO-2 mission, such band is called sCO2, and the 6300cm-1 bands to wCO2, which is confusing here.

We have added mnemonics to the first paragraph of Sect 2:

> "We refer to these as "$lCO_2$" (for "lower" $CO_2$) and "$wCO_2$" (for "weak" $CO_2$), respectively."

Maybe the "CO2 window centered at 6220 cm-1" (l.119), which is the first standard window, should be given a short name as it the case for sCO2 and lCO2?

No. The 6220 and 6339 windows make up the standard TCCON CO2 product, so these windows should be referred to as "CO2" with no prefix letter, as was the case in GGG2014, for continuity.

**Sect. 3**

For clarity, I think a chapter named "Improvement of the forward model and the retrieval" should be created to include sections 3, 4, 5 and 6. The following sort would be more obvious : 5, 4, 3, 6.

As it is atypical to include chapter level organization in a paper and these sections do address distinct parts of GGG, we have kept them as individual sections. However, we have adopted the suggested order and added a new figure with a flowchart of the components of GGG and the data delivery (new Fig. 1).

L 114 : give a reference for "Numerous spectroscopic studies"?

We have added references to Tran et al. (2013), Hartmann et al. (2009), and Gordon et al. (2017).

L149 : Xluft is an important notion, but new and never mathematically defined (later it is said "similar to Xair"). Please give the mathematical formula of Xluft.

We have added one as the new Eq. (4).

**Sect. 4**

Please recall (in introduction?) that the absorption cross-sections are pre-computed, using the meteorological profiles. Despite the 3-hourly new product, can you confirm in the plain text that only one profile per day is used?

We do not use one profile per day in GGG2020, we now use the nearest 3-hourly profile in time. We have added a new sentence to Sect. 4.2 to clarify:

> "GGG2020 uses the nearest profile in time, changing every three hours, to better capture changes throughout the day."

**Sect. 5**

section 5.2: I cannot catch the improvement of GGG2020 with respect to 2014 in this section.

This was the reduction in the phase correction threshold, described near the end of the section. We have made the change its own paragraph so it is easier to find.

section 5.3: I was not able to find in literature (Wunch et al. 2011, 2015) that the TCCON interferometer is single—sided. Please mention it (introduction?), and provide the length of the short arm (as well as that of the long arm).

TCCON sites must have a maximum optical path difference of at least 45 cm, but may choose to use a longer max OPD if they wish. Therefore there is not a single arm length to give. The length of the short arm varies considerable between sites, from 0.1 to 5 cm. But this doesn't matter because the current processing discards almost all the short side. We have made explicit in Sect 5.3 that TCCON spectrometers are single passed:

"We now make better use of the entire interferogram collected by the spectrometer in i2s. In typical linear single-passed Fourier transform spectrometers **(such as those used by TCCON)**,"

Note that TCCON interferometers are single-passed and record singled-sided interferograms. EM27/SUNs, on the other hand, often record double-sided interferograms. It is only for double-sided interferograms that we use the short side of the interferogram. We have clarified this:

"I2S now has the capability to process interferograms as single sided (using data only from one side of ZPD, usually the long arm) or double sided (using data from both sides of ZPD, the long and short arms). When processing an interferogram as double sided, the optical path difference (OPD) on either side of ZPD must be the same. This means that for standard TCCON processing, I2S will always choose to process the interferogram as single sided, because the long arm is much longer ($\geq$ 45 cm) than the short arm (typically 0.2 to 5.0 cm). However, for spectrometers such as the EM27/SUNs where the OPD is more symmetrical about ZPD, I2S can process the interferogram as double sided, which avoids discarding useful data from the short arm."

l 309: please explain why it is "more efficient" : is it for a better SNR?
It simply avoids discarding part of the interferogram unnecessarily. We have clarified this:

"I2S can process the interferogram as double sided, **which avoids discarding useful data from the short arm.**"

L 312: "spectral response of the instrument" is ambiguous, may be confused with ILS. I understand you are talking about the instrumental "continuum".
We have added a footnote clarifying the distinction between "spectral response of the instrument" and "ILS":

"Here, by "spectral responses of the instrument," we mean an instrument-specific response which can be characterized as a frequency-dependent vector that multiplies the incoming solar spectra. This is distinct from the ILS, which is instead best considered as an instrument-specific vector that convolves the incoming solar spectra."

L 320: "the discrete Legendre polynomials" is not mentioned in Wunch et al 2015. Do you confirm it? Why Legendre polynomials and not classical polynomials?
Yes, we use Legendre polynomials, as they have useful properties, the most important being that they are orthogonal. In a classical polynomial (i.e. $y = a + bx + cx^2$) the coefficients are not orthogonal. We have noted this in Sect 7 of the revised paper now:

"Higher order **Legendre** polynomials are now used widely in the GGG2020 spectral windows to better account for continuum shape changes between instruments and over time. **(We use Legendre polynomials because they are orthogonal, whereas standard polynomials are not.)** The continuum curvature fitting option..."

Please give the orders used per window (or at least their maximum).
We have added new tables A2 and A3 and reference them in what is now §7.

**Sect. 7**

Section 7.1: sub-sections would be welcome
We have split this into subsections for ADCF approach changes and derivation.

I think that equations (3), (4) and (5) cannot be understood without an explicit reference to Wunch et al 2011 appendix A.e.i (l356 mentions "like GGG2014" but Wunch et al 2015 does not mention it). Please refer to it. Please recall that f is a model for the observed Xgaz diurnal variation, making the important assumption that any symmetrical Xgaz variation around noon is not expected to be true but an artifact.
We have added this reference and a note about the assumption of symmetrical behavior:

"GGG2020, like GGG2014, applies a post hoc correction to the $X_{\text{gas}}$ values to remove airmass dependences. This correction is applied to each $X_{\text{gas}}$ value. **It has a similar form to that in Appendix A of Wunch et al. (2011)...**"

"...where $t$ and $t_{\text{noon}}$ are the measurement time and solar noon time (in day of year), $f_c$ is the polynomial defined in Eq. (5), and $c_{\text{mean}}$, $c_{\text{asym}}$, and $c_{\text{ADCF}}$ are the fitted coefficients. **This equation assumes that symmetrical variation of $X_{\text{gas}}$ values around noon (fit by $f_c$) are due to spectroscopic errors and real variations throughout the day are antisymmetrical and will be fit by the $c_{\text{asym}}$ term....**"

I think an illustration of XCO2=f(t,theta) with several examples would be welcome, and also to show the standard deviation that is aimed at being minimized.
It is not the standard deviation of XCO2 being minimized but the standard deviation of the ADCF. This is already illustrated in Fig. 6 (now Fig. 11).

I note that despite Wunch et al 2011, the sin() function is replaced by a linear function, why?
The sin function was just missed in entering the equation, it has been fixed. Thanks for catching that.

l.383: please precise how you get these uncertainties.
We have added a short explanation:

"The coefficients and their errors are calculated with a weighted least squares fit using the individual windows' $X_{gas}$ uncertainties **(calculated from the spectral residuals of the target gas and $O_2$)** as the weights."

l 395: As far as I understand by comparison with previous papers, this temperature dependence correction is new in GGG2020. Please emphasize it.

We have added two bullet points at the beginning of the section to highlight this:

"In GGG2014, only data from 3 TCCON sites (Park Falls, Lamont, and Darwin) were used to compute the ADCFs. For GGG2020, we use 18 sites' data....In GGG2014, we did not examine the ADCF for temperature dependence. We do in GGG2020 and attempt to account for that in how we select the final ADCF values."

l 399: The use of the theta notation for potential temperature is source of confusion since theta is use earlier for SZA. Maybe you should change it.

We now use "SZA" instead of theta for solar zenith angle

l.402: please detail the exact operation: division by the ACDF at theta_mid=310K? linear correction requesting the knowledge of theta_mid?

There is no operation, the value of the fit at 310 K *is* the ADCF. We have done our best to clarify:

"For each window, we use the value of the fit to this data at $\theta_{mid} = 310$ K as the final ADCF value."

L.419: in this paper, as well as in previous papers, HCl was never mentioned as an atmospheric gas measured by TCCON (see table 3 of Wunch et al 2015), but only as for the gas cell for interferometer calibration. The mentioned windows should therefore be explained.

We have added:

"Lastly, we also removed a number of HCl windows. **TCCON instruments use HCl lines to assess instrument alignment with an HCl cell that can be illuminated by the solar beam or an internal lamp.** TCCON used 16 windows to measure HCl in GGG2014..."

Section 7.2: after reading the Wunch et al. 2015, 2011 and 2010, I was not able to find the way windows were merged in GGG2014 and older. I do not fully understand the iterative process described here. Please detail it.

GGG is open source software. Users interested in this level of detail are encouraged to download the software and examine the algorithm themselves. Because we do not use the iteration in GGG2020, we do not wish to distract from the main point of this section.

l 432: you mention the "retrieval error", previous papers seems not to define it. I think it should be given in section 1.

Changed "retrieval error" to "spectral residuals".

l.472: please give the formula of the FVSI index (or give a reference). Is it computed for a single scan duration? What is the duration of a single scan? (it can be mentioned in section 1)

We have added:

> "This is the standard deviation of solar intensity divided by the average solar intensity during the $\sim 80$ s long scan, and filters out observations impacted by intermittent clouds."

l.481 also requires to mention the duration of a single TCCON duration, so as to understand whether 30 TCCON is a large part of a 2h window or not.

We have added:

> "For each in situ profile, we require **at least** 30 TCCON observations **(each $\sim 80$ s)** passing these quality checks""

l.495: I guess ai is the averaging kernel, please clarify it.

We have added the symbols used for each term in parenthesis within the text:

> "$\delta\mathbf{x}$ is the difference between the in situ $(x_{\text{insitu},i})$ and TCCON posterior $(x_{a,i})$ profiles, modified by the TCCON averaging kernel $(a_i)$: $\delta x_i = a_i(x_{\text{insitu},i} - \gamma x_{a,i})$"

l.514 and fig.11 legend: please explicitly precise that the uncertainty bars are given by appendix C6.

We have added:

> "The calculation of each term and how they are combined for the error bars in Fig. 16 is detailed in Appendix C6."

l.564: can you tell which way was used for each dataset?

We have added a footnote that points to the NOAA release notes describing this:

> "The ObsPack release notes at `https://gml.noaa.gov/ccgg/obspack/release_notes.html#obspack_co2_1_GLOBALVIEWplus_v7.0_2021-08-18` provide information on how to determine which data was fully recalibrated."

l.582: for clarity reason, I would suggest to start a new sub-section dedicated to O2 decrease, as this is a different correction source (even if applied simultaneously with new XCO2 standard).

Since both changes are applied to the same data, we prefer to keep this as one section.

l.634: It is not clear in this paragraph whether the new product includes the variable O2 variable fraction or not. The answer is given later in l.692, with some redundancy, therefore I would discard the l.634 paragraph.

This paragraph starts with "$X_{CO_2}$, $X_{wCO_2}$, and $X_{lCO_2}$ on the X2019 scale *and accounting for the variable $O_2$ mole fraction...*"; it explicitly says that both changes are incorporated in the product.

**Sect. 8**

This chapter is very important and must be kept. But it is a long part in an already long paper, and a bit different from the remaining of the paper which explains the updates of GGG2020. I would suggest several solutions: (a) To place it in a dedicated companion paper. This can be merged with appendix B (which is small) or (b) (preferred) to move the text between l.704 and 782 in appendix B. This part is largely an update of similar works by Wunch et al. 2015 (section 8), 2011 (appendix B).

Given the emphasis being placed on pushing uncertainties in and precision of remotely sensed GHG products to smaller and smaller values, especially in the next generation of space-based missions, understanding what the dominate sources of random uncertainty are is critical to guide future work. Therefore we prefer to leave this section where it is.

Text between l.704 and 782 at least deserves its own sub-section

This text effectively is in its own subsection, since it is in the leading part of the overall section.

Please refer to Wunch et al. 2011 and Wunch et al. 2015 for section 8. In particular l.776 & 777 can refer to Wunch et al. 2015 for more details on ME.

We added:

> "Here we model two cases: a "shear" misalignment, where the modulation efficiency of the spectrometer increases linearly to 1.05 as a function of optical path difference, and an "angular" misalignment, where the modulation efficiency drops linearly to 0.95 as a function of optical path difference. **(See section 8 of Wunch et al. (2015) for more details on the mathematical forms for these misalignments.)**"

I would change the numeration of 8.x for x in 2,..,10 to 8.1.x.

Since these sections expand on the general comments in 8.1, we prefer to keep them as-is.

l.854: Please define this scale factors for HCl and how to use them to assess the ILS (or give a reference)

We added

"...largely independent of surface pressure or other atmospheric adjustments. **Therefore, deviations of the HCl scale factors from 1 indicates a drift in ILS.** To assess the..."

**Sect. 9**

I think it would be more consistent to place this section at the beginning of document, either after 2 of after section 6.
Agreed, we have moved this after sect. 2.

**Typing corrections**

Section 6.1 has no 6.x follower
That is intended.

Section 7.1.1 has no 7.1.x follower.
That is intended.

figures 8, 9, 11 and their font should be enlarged
We have increased the font size and rearranged the panels of fig 11 to allow each panel to be larger

l.1595: the link seems to be dead.
Corrected.

**References**

Gordon, I. E., Rothman, L. S., Hill, C., Kochanov, R. V., Tan, Y., Bernath, P. F., Birk, M., Boudon, V., Campargue, A., Chance, K., et al.: The HITRAN2016 molecular spectroscopic database, J. Quant. Spectrosc. Ra., 203, 3–69, 2017.

Hartmann, J.-M., Tran, H., and Toon, G.: Influence of line mixing on the retrievals of atmospheric CO 2 from spectra in the 1.6 and 2.1 $\mu$m regions, Atmos. Chem. Phys., 9, 7303–7312, 2009.

Roche, S.: Measurements of Greenhouse Gases from Near-infrared Solar Absorption Spectra, Ph.D. thesis, University of Toronto, URL `http://hdl.handle.net/1807/108784`, 2021.

Rodgers, C. D.: Inverse Methods for Atmospheric Sounding Theory and Practice, World Scientific Publishing Co. Pte. Ltd., 2000.

Tran, H., Ngo, N. H., and Hartmann, J.-M.: Efficient computation of some speed-dependent isolated line profiles, J. Quant. Spectrosc. Ra., 129, 199–203, 2013.

Wunch, D., Toon, G. C., Wennberg, P. O., Wofsy, S. C., Stephens, B. B., Fischer, M. L., Uchino, O., Abshire, J. B., Bernath, P., Biraud, S. C., Blavier, J.-F. L., Boone, C., Bowman, K. P., Browell, E. V., Campos, T., Connor, B. J., Daube, B. C., Deutscher, N. M., Diao, M., Elkins, J. W., Gerbig, C., Gottlieb, E., Griffith, D. W. T., Hurst, D. F., Jiménez, R., Keppel-Aleks, G., Kort, E. A., Macatangay, R., Machida, T., Matsueda, H., Moore, F., Morino, I., Park, S., Robinson, J., Roehl, C. M., Sawa, Y., Sherlock, V., Sweeney, C., Tanaka, T., and Zondlo, M. A.: Calibration of the Total Carbon Column Observing Network using aircraft profile data, Atmos. Meas. Tech., 3, 1351–1362, https://doi.org/10.5194/amt-3-1351-2010, 2010.

Wunch, D., Toon, G. C., Blavier, J.-F. L., Washenfelder, R. A., Notholt, J., Connor, B. J., Griffith, D. W., Sherlock, V., and Wennberg, P. O.: The Total Carbon Column Observing Network, Philosophical Transactions of the Royal Society A: Mathematical, Physical and Engineering Sciences, 369, 2087–2112, https://doi.org/10.1098/rsta.2010.0240, 2011.

Wunch, D., Toon, G. C., Sherlock, V., Deutscher, N. M., Liu, C., Feist, D. G., and Wennberg, P. O.: Documentation for the 2014 TCCON Data Release, https://doi.org/10.14291/TCCON.GGG2014.DOCUMENTATION.R0/1221662, 2015.

---

## Author Comment (AC2)

**The Total Carbon Column Observing Network's GGG2020 Data Version**

**Response to Anonymous Referee #2**

Joshua L. Laughner on behalf of all coauthors

February 26, 2024

Thank you to the reviewer for taking the time to evaluate our manuscript, and for their positive endorsement of its value. We have done our best to address the comments made. Below, comments from the reviewer are in red, our responses are in blue, and quotes of the manuscript are in black. Unless otherwise stated, line, figure, table, and section numbers used in comments and responses refer to the *original* manuscript while such cross-references in quotes of the new version of the paper refer to the *revised* version.

One note on the organization. The processing chain is complex and there are many steps. The first section of the paper gives a bit of an overview, and then notes the subprograms and where they are discussed in the paper. I do agree with another review that suggests the paper sections should be ordered to reflect the processing chain more directly. Start with the section on interferograms (now 5), and then move to gsetup (now section 4), and then the spectroscopy and continuum fitting (now sections 3 and 6). Finish with post processing, as the paper now has.

Agreed, we have adopted the suggested order as well as moved the section on miscellaneous changes to the data format to follow sect. 2. We have also added a new figure with a flowchart of the components of GGG and the data delivery (new Fig. 1).

The introduction section steps through the process of gathering measurement data and transforming into the desired gas columns. I was expecting it to set the stage for the subprogram discussion that starts at about line 60. I would suggest a modification in the paragraph that starts on line 23 with "TCCON instruments measure solar spectra in the near-infrared (NIR) wavelengths;". I suggest you say TCCON instruments measure interferograms with direct sun measurements in the near-infrared (NIR wavelengths). There are transformed into spectra. . . . Or something to that effect, so the idea of interferograms is introduced here. The transformation to spectra and the issues such as detector non-linearity and phase correction are important and improvements there are having a significant positive impact on the dataset.

Changed this sentence to:

> "TCCON instruments **record interferograms of direct-sun measurements** in the near-infrared (NIR) wavelengths. **These interferograms are transformed into spectra from which** the final column average mole fractions

[Figure]

Figure R1: The new figure added to compare the TCCON and OCO-2/3 CO2 windows.

> (henceforth denoted as "$X_{\text{gas}}$", e.g. "$X_{\text{CO}_2}$") are derived using the retrieval software GGG."

Line 55 - I would replace "for all retrievals except those listed in §7.3.2" with "for the discussion of changing O2 mole fraction in §7.3.2"
We changed this to:

> "GGG2020 assumes that $f_{\text{O}_2} = 0.2095$ for **all** $X_{\text{gas}}$ **products** except those listed in §8.3.2 **with a variable O$_2$ mole fraction implemented.**"

Lines 82 and 83 - how do the windows discussed here relate to the measurement windows of OCO-2 and other satellite instruments. Seems that if they are the same, this could provide some insights into how the intercomparions of TCCON and satellite data are influenced by window selection.
We have added a new figure that shows this visually and a new sentence referencing it:

> "Figure 2 shows how these two windows (plus the windows for the standard TCCON $X_{\text{CO}_2}$ product) align with the strong and weak CO$_2$ windows used by OCO -2 and -3.""

Line 111 - perhaps add a pointer to the section where the solar continuum is addressed (section 6)?
We have added a final line to this section:

> "The solar continuum is handled separately from the linelist in GGG. This is discussed in §7."

Line 117 in section 3.2 - what are the implications of the choice of lineshape? Now HITRAN is not a database that includes the info you need? Is that a change of strategy?

Correct, HITRAN (or at least the standard 160 character product) does not contain all the data we need. However, this is not a full-fledged change in strategy, as we have always developed a custom linelist that draws from multiple sources. We have added a paragraph to the non-Voigt section that says this:

> "This does mean that the standard 160-character wide HITRAN linelist product does not include all of the parameters required for these gases. It has always been true that GGG has used a customized version of the HITRAN linelist. Therefore, this need for additional parameters represents an increase in the complexity of our linelist strategy, but a continuation of the same approach to use the best spectroscopic information from various sources."

Line 198 - Are the changes you have made to the line intensities similar to the uncertainties reported by spectroscopists? Do experts in this area think this is a reasonable scaling?

HITRAN uncertainties are conservative, so if they're asking if our changes are within the uncertainties reported by the HITRAN codes, then we believe so. Uncertainties reported by individual groups tend to be more optimistic, so we expect our changes are within the group-to-group differences, but perhaps outside the individual groups' reported uncertainties.

Line 275 - is igram meant to be interferogram?

Yes, changed.

Line 300 - I'm curious to know what the level of disagreement is that remains? Is it hard to summarize with a single statistic, and that is what you have just stated "does not completely resolve the problem", or can you be more quantitative?

The challenging part here is that the underlying problem—inconsistencies in the phase correction of the interferogram—can be present without causing the more readily observable bias in XCO2 between the forward and reverse scans. In our development, lowering the PCT eliminated that forward/reverse bias in XCO2, but did so by moving the challenging part of the phase correction to a part of the interferogram where it is much less likely to impact the Xgas values. So by "does not completely resolve the problem," we meant that the impact on the Xgas values (that the user cares about) has been mitigated, but the underlying physical problem remains and could have other impacts not yet identified. We have modified this sentence to say this more plainly:

> "...which improves the consistency between forward and reverse scans. **This eliminates the observed bias in $X_{CO_2}$ between forward and reverse scans, but is not a fully general solution to the underlying problem.** We hope to develop a future version..."

Line 344 - is there information about the order of the polynomials in the final data product? How does a user know of this has been set properly or if channel fringing is an issue in a particular instrument?

This information is stored in the detailed netCDF files delivered by TCCON partners to Caltech. We review diagnostics for channel fringing as part of our routine quality control

process before releasing the data, and can check the polynomial order if we have reason to suspect that these are incorrect. Users of the public data do not have access to the detailed information (the sheer volume of variables in the private files make them much more difficult to use); if they have concerns about the data, they are encouraged to contact us with questions. We have added a paragraph to §6.1:

> "Diagnostics to detect channel fringes are reviewed as part the quality control process before TCCON data is made public. Any channel fringes detected will be removed by adjusting the fitting before the data is released to the public archive, though this is extremely uncommon."

Line 516 - figure 11 presents data as a function of Xluft. Did you examine other variables, like time or modulation efficiency, and decide that Xluft was the best variable to use?

We did look at other variables, but have not yet found a better predictor of bias. Xluft is particularly useful because it acts as a metric for several error terms (e.g. ILS, nonlinearity, pointing), so for Fig. 11 it is the best variable for the x-axis.

---

## Author Comment (AC3)

**The Total Carbon Column Observing Network's GGG2020 Data Version**

**Response to Referee #1 (Gretchen Keppel-Aleks)**

Joshua L. Laughner on behalf of all coauthors

February 26, 2024

Thank you to Dr. Keppel-Aleks for her positive review of this manuscript's importance and her careful consideration of how to improve its value as a reference for our data product. We have done our best to address the comments made. Below, comments from the reviewer are in red, our responses are in blue, and quotes of the manuscript are in black. Unless otherwise stated, line, figure, table, and section numbers used in comments and responses refer to the *original* manuscript while such cross-references in quotes of the new version of the paper refer to the *revised* version.

The paper does presuppose knowledge of the network's goals and past versions of the data. For example, line 46 casually states that there is a 0.25% accuracy needed for greenhouse gas data, but this is not motivated in the next nor cited from another source. It could be nice to provide a brief retrospective of the defined needs and past performance for TCCON's precision/accuracy, noting that the precision needs for validation of satellite instruments and for carbon cycle science could be different.

We have added two new paragraphs to the introduction:

> "The need for updates to the retrieval algorithm used by TCCON has been largely driven by the need for increasingly high accuracy and precision of total column greenhouse gas (GHG) data for carbon cycle science and satellite validation. GHG measurements require quite high precision to distinguish signals from anthropogenic, terrestrial, or oceanic processes from the background mixing ratios. The 2018 National Academies decadal strategy recommends random and systematic errors for $CO_2$ be less than 1 and 0.2 ppm ($\sim 0.25\%$ and $\sim 0.05\%$), respectively and likewise less than 6 and 2.5 ppb ($\sim 0.3\%$ and $\sim 0.1\%$), respectively for $CH_4$ (National Academies of Sciences, Engineering, and Medicine, 2018, Table B.1, question C-3, p. 601). Future space-based $CO_2$ observing missions are striving for even greater precision; for example, CO2M has a stated goal of 0.7 ppm precision and $< 0.5$ ppm systematic error in $X_{CO_2}$ (ESA, 2020). The increasingly stringent precision requirements for carbon cycle science and satellite validation demands that ground based networks, such as TCCON, continue to refine their data to support these requirements.

A second factor driving improvements in the retrieval is the emergence of portable, low resolution solar-viewing FTIR instruments such as EM27/SUNs. These instruments can be deployed to areas that cannot support a full TCCON site, and are also affordable enough to be deployed in greater density around locations of interest (e.g. cities). This capability complements the higher precision and accuracy data produced by TCCON. To facilitate comparisons between TCCON and EM27/SUN data, it is beneficial to use the same retrieval for both. Improvements to handling of EM27/SUN interferograms (§4.3) have been added."

It might be helpful to provide a table of contents at the end of the introduction, to let the reader know what the main sections of the paper are (e.g., 2 = new Xgas, 3 = Updated spectroscopy). Since many readers will not read the paper end-to-end, this could make the manuscript more effective as a reference.

Done; this is the new Table 1. We have also added a new figure with a flowchart of the components of GGG and the data delivery (new Fig. 1).

While reading the paper, I thought it could be helpful for the authors to tabulate the remaining/known issues with the GGG2020 data that the next version of GGG will attempt to address. For example, the XCO2 vs Xluft dependence or the T-dependence of N2O retrievals. A summary section is included at the end of the paper, however, so this might be unnecessary, but it would be helpful to note that this section is coming in the table of contents so the reader knows it is coming. On the other hand, it could be helpful, to summarize the known issues in a compact format (table, or a call-out box rather than just another section of the paper) if the authors think it could help the user community avoid author-anticipated pitfalls when interpreting the data.

We have added Table 6 in the conclusion.

Somewhat similarly, when I read the error budget section, I noted that a table would be helpful, and there was a table at the end. It is great that the paper is so thorough, but adding a bit of organization to let the reader know that something is coming a bit later on would be very helpful.

We added a sentence at the end of the Sect. 8 introduction that points to this table:

"Each source of uncertainty included in our error budget is described below. **Table 5 towards the end of this section summarizes the error budget for the primary TCCON products.**"

Line 111: I am curious the extent to which the linelists have been empirically adjusted. For example, how many weak lines have been added based on the empirical identification process described in this paragraph? It would also be interesting if the paper noted how/whether this information gained from the high resolution TCCON spectra flows back to other groups in Earth/sun remote sensing.

It is difficult to summarize the extent of the changes to the linelists, since these changes involve adopting new parts of the HITRAN linelists, adding/removing/shifting lines, etc.

We've instead tried to summarize the improvement in $CO_2$ at least with statistics of the mean spectral residuals and bias calculated from lab spectra with known $CO_2$ amounts. This is table 3 in the revised paper. In addition to this table and a paragraph explaining it, we have added a short paragraph on how these improvements in spectroscopy are communicated to the HITRAN group:

> "Improvements to the telluric linelists are communicated to the HITRAN group through spectroscopic evaluations, posted to `https://mark4sun.jpl.nasa.gov/presentation.html` (last access 31 Jan 2024). Such evaluations are also performed on candidate linelists developed by the HITRAN group to provide feedback on the performance of those linelists before they are adopted."

Line 118: refers to Mendonca 2016 and Devi 2007ab, which leads me to believe that these are the lineshapes that are being used in the 6220 and 6339 cm-1 CO2 bands but this is not explicitly stated.

We have added a table to the appendix that lists all the GGG2020 windows and this includes whether gases in that window include non-Voigt data. We point to this at the end of the first paragraph of Sect. 3.2:

> "...quadratic speed-dependent Voigt (qSDV) with line mixing (LM) code from Tran et al. (2013) was implemented into forward model of GGG (Toon, 2022). **Tables A2 and A3 list the frequency windows used in GGG2020, and contain columns identifying which windows include speed-dependent and line mixing lineshape information**"

Line 122: What is the metric the authors used to determine that the new spectroscopy and line shape improved "the quality of XCO2 retrievals in this spectral region" ?

We have clarified that this means spectral RMS, as mentioned earlier in the paragraph:

> "...the spectroscopic parameters from Benner et al. (2016) are used with the qSDV and first order LM to calculate absorption coefficients. This resulted in improving the quality of $XCO_2$ retrievals **(i.e. reducing the spectral fit RMS)** from this spectral region. New spectroscopic studies aimed at improving $CO_2$ absorption coefficient calculations..."

Section 3.3: The empirical process to optimize the O2 line widths was very detailed, and I had to read it a few times. I am not sure how the text could be more clear given the multiple interdependent constraints. The figures here were very helpful in communicating the approach. One minor comment is that the allusion to T700 as a metric for synoptic scale meteorology doesn't seem to fit (since a relationship between non-oxygen gases and T700 represents real variability whereas with O2 it is a spurious relationship). It might be more clear to say that T700 was used as a temperature metric because this value is already saved with the observed spectra since it has also been applied for downwind scientific analyses.

T700 isn't saved in the spectra for GGG2020, but was calculated from the a priori profiles for this. We have rewritten this sentence as follows:

"Our rationale was that the temperature dependence of $X_{\text{luft}}$ was the most important error to eliminate, thus minimizing its magnitude took priority. T700 is taken from the a priori meteorology data, **and was chosen on the assumption that this is a reasonable metric for temperature variations in the free troposphere containing the majority ($\sim$ 60%, 800 to 200 hPa) of the $O_2$ column.**"

Line 178: The sentence about minimizing the variance in Xluft rather than the average value was confusing to me (perhaps because the sentence is pre-empting an argument that didn't occur to me – I'm not sure why it would be useful to minimize the average value).
We've added a sentence to clarify:

"...rather than the pressure and temperature effects on line width adjusted in this initial experiment. **Therefore, while we ideally want $X_{\text{luft}} = 1$, this first step was not optimizing the spectroscopic parameters that can achieve that. We do adjust the $O_2$ line strengths separately, as noted at the end of this section.**"

Section 4.3: The reference to trace gas profile updates here was not alluded to in the introduction; perhaps mention ginput in the introductory section
We didn't include any list of updates to the algorithm in the introduction. We believe that now the table of contents serves that purpose; it includes a mention of the meteorology and trace gas profile updates.

Line 251: Clarify whether the DIP metric is reported for users to consider, or is an active filter employed by the TCCON team prior to reporting data. Upon the first reading, it seemed like it was a diagnostic, but I later realized (Line 940) that it is actively used in the filtering process.
We have edited the second-to-last paragraph in this section to make this clear:

"**We now use the DIP diagnostic during the quality control step to identify data affected similarly by nonlinearity. Once such data is identified,** the correction process **described in the previous paragraph** is applied to **the afflicted data.**"

Line 275 "igram" should be "interferogram", I think.
Fixed.

Line 307: The need to phase correct the interferogram is clear, but it is not clear why computing a spectrum using both the long and short arm of the interferogram is "more efficient" than only using the long arm. Please add a brief sentence to clarify.
This just avoids discarding useful data, but this is only used when processing double-sided interferograms. Since the main use case with double-sided interferograms is EM27 data, we have changed this paragraph to be clearer:

[Figure]

Figure R1: The updated Fig. 5 (now Fig. 10) showing the efficacy of the airmass correction.

"I2S now has the capability to process interferograms as single sided (using data only from one side of ZPD, usually the long arm) or double sided (using data from both sides of ZPD, the long and short arms). When processing an interferogram as double sided, the optical path difference (OPD) on either side of ZPD must be the same. This means that for standard TCCON processing, I2S will always choose to process the interferogram as single sided, because the long arm is much longer ($\geq$ 45 cm) than the short arm (typically 0.2 to 5.0 cm). However, for spectrometers such as the EM27/SUNs where the OPD is more symmetrical about ZPD, I2S can process the interferogram as double sided, which avoids discarding useful data from the short arm."

For section 7.1 on the airmass correction, the description of the ACDF was a bit confusing. There were quite a few variables introduced, and some might be redundant. For example, my understanding is that alpha and ACDF are the same? And that c3 is closely related to these (just that it has a daily fitted value whereas alpha is the mean of all c3's), so perhaps there is some notation that could be used to show this more clearly (e.g., calling it alpha_i, or alpha_daily)?

We replaced alpha with "ADCF" directly in the equations and renamed the $c$'s to $c_{\mathrm{mean}}$, $c_{\mathrm{asym}}$, and $c_{\mathrm{ADCF}}$ to more directly communicate their meaning.

Figure 5: Would it be possible to show panels with the corrected data (side-by-side panels or overlaid if that isn't too messy) to show the result of the procedure?

Done. We have also added background bar plots to indicate the number of data points contributing to each bin, so as to allow readers to understand that some of the low and high SZA behavior is due to having less data available.

Figure 6: It appears that the value selected for g in the 6339 cm-1 window is 45 deg,

which is at the margin of the allowable parameter space. This suggests the optimization hasn't converged, and may reflect that the functional form chosen is not appropriate. Can the authors comment on this?

While it may not have fully converged, we can see from the background color (representing the quantity being minimized) that the final solution is in the middle of a part of the parameter space where the metric to be minimized is already near 0. In fact, because both panels of this figure use the same color map, we can see that the 6339 cm-1 window's standard deviation is overall lower than the 6220 cm-1 window. That suggests instead to us that the 6339 cm-1 window can be well corrected with a variety of values for g and p. That is borne out by the newly added panels for the previous figure (now fig. 6) which show that the $CO_2$ airmass correction is very effective, leaving only a very minor variation in XCO2 with airmass for both windows.

Figure 8 and 9: Please increase the font size on these figures.
Done.

Section 7.2: perhaps clarify whether the sj are the same for all sites and times.
We have added:

"Thus, while GGG2020 retains the capability to compute the $s_j$ values on-the-fly, the $s_j$ values are prescribed for standard TCCON processing, **and all sites use the same $s_j$ values.**"

Line 450: The vertical scale has been changed from 70 uniform levels to one in which the level separation increase away from the surface. Can the authors comment on how this change impacts the error associated with the stratospheric extrapolation?

We have previously shown (Laughner et al., 2023) that the a priori profiles used for this extrapolation have very good agreement with AirCore profiles in the lower stratosphere. Therefore, any small loss in accuracy due to the larger spacing should be more than offset by the improvement in the quality of the profiles themselves. We have edited this bullet point to say:

"extend the profiles' tops to 70 km altitude using the standard GGG2020 priors **(shown in Laughner et al. (2023) to have good agreement with in situ profiles in the stratosphere)** and to the surface by extrapolation or use of surface data,""

Line 475: how frequently is a negative Xgas retrieved, and are there any general conditions under which this occurs?

Depending on the gas, this is between 0 and 1% of the spectra in the time windows around the in situ profiles (0.7% for CO2, 0.1% for CH4, 0% for CO and H2O). Because this happens so infrequently, we consider these to be outliers that should be removed. With so few data points to investigate, it is difficult to determine if there is any consistent reason this occurs; however, it probably happens when a cloud interrupts an already-started interferogram.

Figure 11: Could this be bigger and could the yellow color be replaced with a more saturated hue?

Done. We reorganized the panels to allow them to be larger as well.

Figure 12: Does the orange line (X2019+VarO2 – X2019) imply that there is a 0.2 ppm draft in the X2007 retrievals that do not account for variable oxygen, or is this partially corrected for by the slope of the in situ correction since XCO2? Could a residual error be estimated due to neglecting the trend in O2?

Yes, the orange line does imply a drift, and it is not corrected for by the in situ scaling, as that can only correct for the mean error in O2 mole fraction among the retrievals used in the in situ comparisons. That line is the residual error that comes from neglecting the O2. Since we now provide data fields in the public files that include the variation in O2 mole fraction, however, we would recommend users update to these new fields rather than trying to estimate the effect of the O2 mole fraction.

Equation 10: This is another equation where the key parameter is alpha, but with different meaning. Perhaps a different variable could be chosen so that the variables used in the paper have a unique definition.

Since we changed alpha to ADCF in sect. 7.1, this should be unique now.

The error budget was very thorough, but I do not believe Fig. 17 was referenced in the text. This was the figure I found least informative, so perhaps it could be removed completely.

This is referenced in the "Instrument Line Shape" section describing how the sensitivity tests were performed, so we prefer to keep this figure.

**References**

Benner, D. C., Devi, V. M., Sung, K., Brown, L. R., Miller, C. E., Payne, V. H., Drouin, B. J., Yu, S., Crawford, T. J., Mantz, A. W., et al.: Line parameters including temperature dependences of air-and self-broadened line shapes of $^{12}C^{16}O_2$: 2.06-$\mu$m region, Journal of Molecular Spectroscopy, 326, 21–47, 2016.

ESA: Copernicus $CO_2$ Monitoring Mission Requirements Document, URL `https://esamultimedia.esa.int/docs/EarthObservation/CO2M_MRD_v3.0_20201001_Issued.pdf`, last accessed 11 Jan 2024, 2020.

Laughner, J. L., Roche, S., Kiel, M., Toon, G. C., Wunch, D., Baier, B. C., Biraud, S., Chen, H., Kivi, R., Laemmel, T., McKain, K., Quéhé, P.-Y., Rousogenous, C., Stephens, B. B., Walker, K., and Wennberg, P. O.: A new algorithm to generate a priori trace gas profiles for the GGG2020 retrieval algorithm, Atmos. Meas. Tech., 16, 1121–1146, https://doi.org/10.5194/amt-16-1121-2023, 2023.

National Academies of Sciences, Engineering, and Medicine: Thriving on Our Changing Planet: A Decadal Strategy for Earth Observation from Space, The National Academies Press, Washington, DC, https://doi.org/10.17226/24938, 2018.

Toon, G.: Atmospheric Non-Voigt Line List for the TCCON 2020 Data Release (GGG2020.R0) [Data set]. CaltechDATA, https://doi.org/10.14291/TCCON.GGG2020.ATMNV.R0, 2022.

Tran, H., Ngo, N. H., and Hartmann, J.-M.: Efficient computation of some speed-dependent isolated line profiles, J. Quant. Spectrosc. Ra., 129, 199–203, 2013.